# Directionality of developing skeletal muscles is set by mechanical forces

Kazunori Sunadome[1,14], Alek G. Erickson[1,14], Delf Kah [2], Ben Fabry [2], Csaba Adori[3,4], Polina Kameneva[5], Louis Faure [5], Shigeaki Kanatani [6], Marketa Kaucka [7], Ivar Dehnisch Ellström[8], Marketa Tesarova [9], Tomas Zikmund [9], Jozef Kaiser [9], Steven Edwards[10], Koichiro Maki[11], Taiji Adachi [11], Takuya Yamamoto[12,13], Kaj Fried [3] ✉ & Igor Adameyko [1,5] ✉

Formation of oriented myofibrils is a key event in musculoskeletal development. However, the mechanisms that drive myocyte orientation and fusion to control muscle directionality in adults remain enigmatic. Here, we demonstrate that the developing skeleton instructs the directional outgrowth of skeletal muscle and other soft tissues during limb and facial morphogenesis in zebrafish and mouse. Time-lapse live imaging reveals that during early craniofacial development, myoblasts condense into round clusters corresponding to future muscle groups. These clusters undergo oriented stretch and alignment during embryonic growth. Genetic perturbation of cartilage patterning or size disrupts the directionality and number of myofibrils in vivo. Laser ablation of musculoskeletal attachment points reveals tension imposed by cartilage expansion on the forming myofibers. Application of continuous tension using artificial attachment points, or stretchable membrane substrates, is sufficient to drive polarization of myocyte populations in vitro. Overall, this work outlines a biomechanical guidance mechanism that is potentially useful for engineering functional skeletal muscle.

Striated muscles are composed of oriented and aligned multi-nucleated myofibrils that are generated from individual myocyte fusion events[1,2]. After splitting from mesodermal progenitors, individual myocytes migrate to the future muscle-forming sites and eventually fuse. A wealth of data exists about the various steps in the early and late muscle differentiation program[3]. However, we lack knowledge about how pre-fusion myocytes become polarized within the body to form aligned multinucleated myofibrils with specific orientation.

Striated muscles do not form in isolation. They connect to skeletal elements or other structures starting from early developmental stages. Cartilage development precedes the bone[4], and muscles initially attach to the cartilaginous templates of future skeletal elements[5]. Attachment

[1]Department of Physiology and Pharmacology, Karolinska Institutet, 17177 Stockholm, Sweden. [2]Department of Physics, University of Erlangen-Nuremberg, 91052 Erlangen, Germany. [3]Department of Neuroscience, Karolinska Institutet, 17177 Stockholm, Sweden. [4]Department of Molecular Biosciences, Wenner Gren Institute, Stockholm University, 10691 Stockholm, Sweden. [5]Department of Neuroimmunology, Center for Brain Research, Medical University Vienna, 1090 Vienna, Austria. [6]Department of Medical Biochemistry and Biophysics, Division of Molecular Neurobiology, Karolinska Institutet, 17177 Stockholm, Sweden. [7]Max Planck Institute for Evolutionary Biology, August-Thienemann-Str.2, 24306 Plön, Germany. [8]Spinalis Foundation, 169 70 Solna, Sweden. [9]Central European Institute of Technology, Brno University of Technology, Brno, Czech Republic. [10]KTH Royal Institute of Technology, SE-100 44 Stockholm, Sweden. [11]Laboratory of Biomechanics, Institute for Life and Medical Sciences, Kyoto University, Kyoto 606-8507, Japan. [12]Institute for the Advanced Study of Human Biology (ASHBi), Kyoto University, Kyoto 606-8501, Japan. [13]Center for iPS Cell Research and Application (CiRA), Kyoto University, Kyoto 606-8507, Japan. [14]These authors contributed equally: Kazunori Sunadome, Alek G Erickson. ✉e-mail: kaj.fried@ki.se; igor.adameyko@ki.se

of embryonic muscles to skeletal elements is mediated by specific cells, tenocytes, that express a set of markers that include Scleraxis (Scx) and Xirp2[6–8]. Tenocytes diverge from common Sox9+ chondrogenic progenitors within immature mesenchymal condensations[9]. The particular positions of tenocytes might indeed have crucial importance in orienting the arriving myocytes, which will eventually form the muscle[6]. However, it is still not clear how tenocytes or emerging cartilages interact with forming muscles, and to what extent their interactions affect muscle size and directionality.

Current opinion holds that oriented growth of different body parts, including limbs and face, is mediated by directional cell behavior in response to molecular signals (such as non-canonical WNT ligands) emanating from polarizing centers. For instance, when it comes to the orientation of muscles, Gros and co-authors demonstrated that neural tube-derived WNT11 influences the orientation of early trunk myocytes[10]. Similarly, knockout of *Wnt5a* and its receptors leads to disturbances in limb and facial outgrowth[11,12]. Pronounced phenotypes are caused by knockouts of noncanonical WNT proteins, and members of the WNT-associated planar cell polarity (or PCP) pathway, but the molecular logic behind residual polarization and oriented cell behavior in such mutants remains to be explained.

Mechanical forces have also been shown to influence the orientation of muscle and other cell types in vitro[13,14]. In general, mechanosensory signaling is mediated by specific ion channels[15], as well as integrins connected to the intracellular actin networks[16,17]. However, recent studies have suggested that non-canonical WNT signaling could be integrated with mechanosensory pathways by orienting actomyosin fibrils[18] and that some PCP proteins (i.e. PKD1) are in fact mechanosensory[19]. These experiments leave the possibility that the action of WNT11 on myocytes and myofibrils might be indirect, and could be contingent on mechanical influence from muscle attachment points. Alternatively, principles of attachment and polarization of facial and limb muscles might be fundamentally different from those operating in trunk skeletal muscles.

Tissues and organs are constantly exposed to extrinsic physical forces that play important roles in guiding morphogenesis in a wide range of systems during development and homeostasis[20–24]. Moreover, there is growing evidence that mechanical tension, for example resulting from actomyosin contractility, influences cell polarization to regulate tissue organization[21,25]. Anisotropic tissue growth can, in turn, generate new forces that act on surrounding tissues. Indeed, skeletal structures (such as the notochord, cartilage, and bone) not only provide stiffness and stability to the vertebrate embryo, but also undergo oriented growth that might influence nearby structures. Diseases stemming from insufficient cartilage growth often cause general underdevelopment and hypoplasia of entire body parts, as seen in different cases of chondrodysplasia, achondroplasia and dwarfism[26–28]. Because skeletal elements provide anchoring points for early myocytes, cartilage growth might generate tension on attached and nearby cells, representing a possible mechanism regulating muscle tissue architecture in the face and limbs.

To understand how muscles connect to cartilage and develop spatial orientation, we investigated the mechanisms that integrate cartilage, muscle attachment points, and myocytes. We addressed the hypothesis that early cartilage expansion results in continuous mechanical stretch at muscle attachment points, providing primary orientation cues to the attached myocytes. We found that during embryonic development, emerging cartilage immediately interacts with individual myocytes by orienting them through adhesion and mechanically coupled processes. In a controlled ex vivo environment, tension was sufficient to polarize individual myocytes, independent of fusion events or specific cell type-mediated muscle attachment components. Altogether, knowledge of a mechanical basis for myofiber polarization should benefit manufacturing efforts towards functional ex vivo-grown striated muscles and muscle organoids.

## Results

### Muscle and cartilage development are synchronized in zebrafish

To reveal the timing of myocyte fusion and orientation priming in relation to skeletal development, we used zebrafish embryos as a model system because of their amenability to live imaging and gene manipulation. Between the stages of 45–89hpf, cartilages and muscles emerge in the zebrafish facial region[29], with most ventral musculoskeletal elements becoming apparent between 50–60hpf (Fig. 1a top left). By 72hpf, type II collagen (Col2)-positive cartilage and myosin heavy chain (MyHC)-positive facial muscles already establish a mature interconnected pattern (Supplementary Movie 1, and Fig. 1a bottom left) that consolidates as the fish larva continues to grow (Fig. 1a right).

To track chondrocyte and myocyte cell behavior during facial development, we used transgenic fish that express mCherry under the promoter of *type II collagen* (*col2:mCherry*), and *Cre* recombinase under the promoter of *tbx1* (*tbx1:Cre*), which was created to label muscle progenitor cells and early myocytes (tools detailed in Supplementary Data 1). As expected, the *tbx1* gene was activated in cranial mesoderm (Supplementary Fig. 1a), and the cell labeling by the controlled expression of *Cre* under the *tbx1* promoter led to the tracing of blood vessels, heart, and facial muscles (Supplementary Fig. 1b).

We injected Cre-inducible GFP cassette into Tg (*col2:mCherry, tbx1:Cre*) fish embryos, and subjected them to live imaging with light-sheet microscopy. This experiment revealed that clusters of cartilaginous cells and muscle cells simultaneously emerge in the face within a close distance by 48–50 hpf (Fig. 1b, inferior oblique muscle shown at top, intermandibularis posterior muscle shown in bottom left, adductor mandibulae muscle shown in bottom right). Individual myocytes attach to clusters of coalescing chondrocytes, and grow alongside the cartilage elements as they achieve a roughly final shape (Fig. 1b, muscle-cartilage attachment point displacement represented by cyan arrowheads). Thus, the initiation and further development of the cranial muscles is synchronized with the appearance of the first cartilage and corresponding attachment points (Fig. 1b and Supplementary Movie 2), consistent with earlier reports on the development of the musculoskeletal system[29].

### Cartilage development promotes oriented muscle growth

The coordinated co-initiation of cartilage and associated individual myocytes suggested the role of chondrocytes and tenocytes in control of muscle size and orientation. Therefore, to directly examine the role of cartilage for muscle patterning, we performed loss-of-function experiments focusing on *sox9* and *runx2*, which are essential for cartilage development[30,31]. *Runx2* is expressed exclusively in the developing zebrafish cartilage during the investigated stages of head development, and, thus, provides a perfect target to manipulate cranial cartilage in fish. We generated gRNA against *sox9a* and *sox9b*, two zebrafish paralogues of the *Sox9* gene, for CRISPR/Cas9-mediated gene editing, verified by Sanger sequencing (Supplementary Information). The *runx2* genes have several splicing variants, making it challenging to design target gRNA. Therefore, we used morpholino antisense oligonucleotide for *runx2b*, which was reported to effectively abrogate cartilage development[31].

*Sox9-* and *runx2b-*deficient embryos had the expected craniofacial phenotype (Fig. 1c), including short jaws resulting from growth defects in Meckel's and palatoquadrate cartilages (Fig. 1d). In our experimental setting, the functional inactivation of *runx2b* produced more severe effects on cartilage than combined knockdown of *sox9a* and *sox9b*, judging from the reduced expression of Col2 (Fig. 1e). Facial muscles of both *sox9-*deficient and *runx2b-*deficient embryos showed abnormal morphology with misoriented and misaligned muscles, with *runx2b* downregulation causing more severe effects than *sox9* (Supplementary Movies 3–5). This can be explained by differential impacts on cartilage development, which in turn might be explained by previous reports that morpholino knockdowns are generally more potent than

## Time-lapse IHC and live imaging reveals coordinated facial muscle/cartilage development

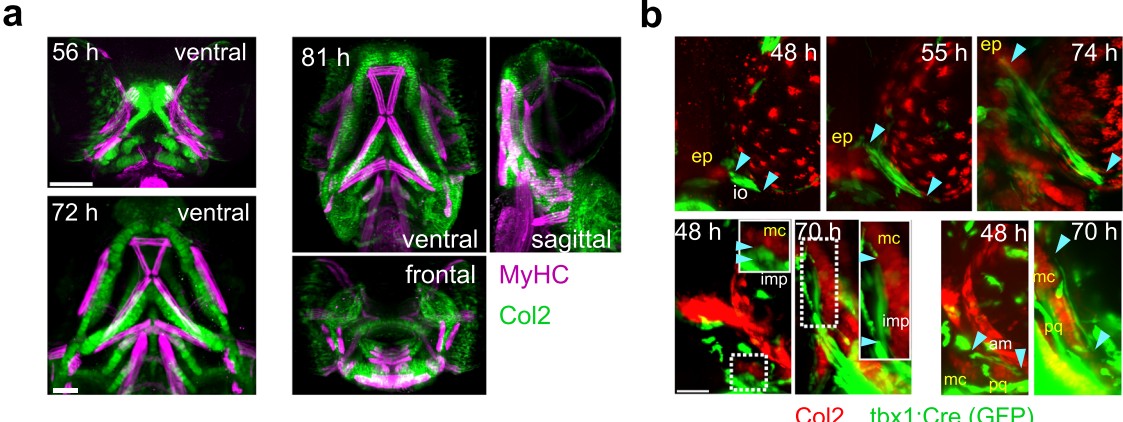

## Impaired outgrowth of jaw in cartilage-deficient zebrafish (81hpf)

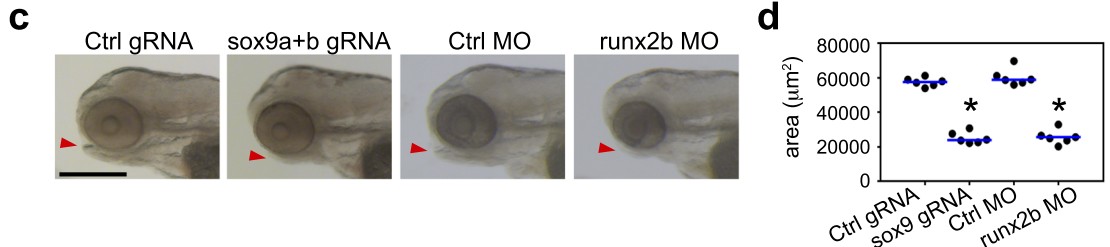

## Effects of cartilage perturbations on muscle length and myofiber morphology (81hpf)

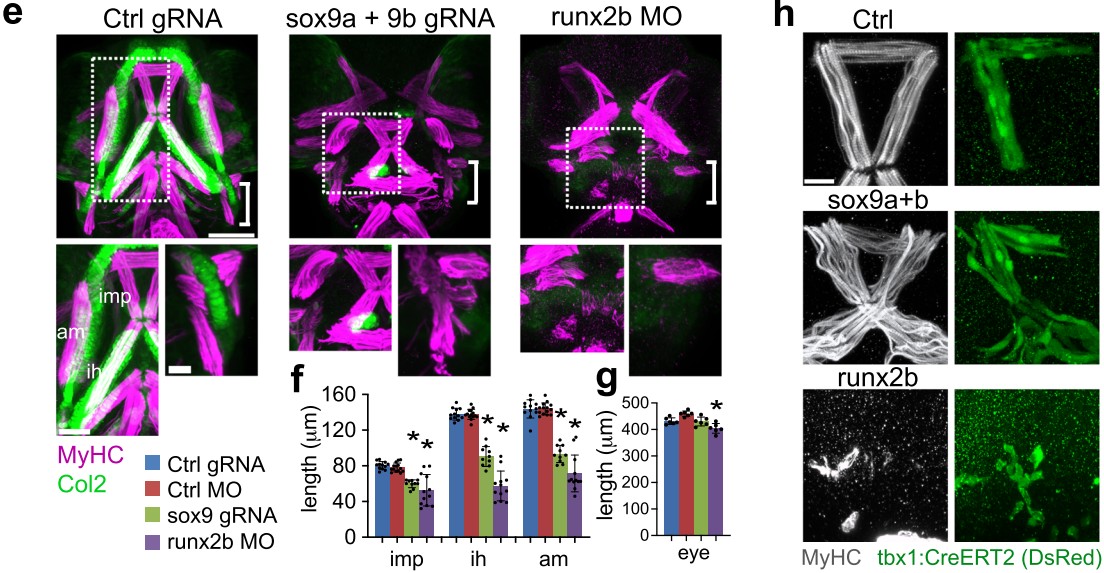

CRISPR editing[32]. *Sox9*-deficient embryos developed misoriented shorter and wider muscles in the mandible, whereas *runx2b*-deficient embryos demonstrated a particularly severe general reduction and misorientation of MyHC+ differentiated myocytes even at 81 hpf (Figs. 1e, f). However, they displayed only marginal changes in eye size (Fig. 1g), suggesting a normal progression of general body development.

Next, we utilized Tg (*tbx1:CreERT2*) to label sparse cells after 4-hydroxytamoxifen (4OHT) treatment. Individual myofibrils of *sox9*-deficient embryos appeared short and curved, and some of them did not reach the end of the muscle (Fig. 1h). The muscle and cartilage phenotype in Sox9-deficient embryos persisted until at least 120 hpf

(Supplementary Fig. 1c), lowering the likelihood of experimental artifacts driven by nonspecific developmental delays. Myofibers of *runx2b*-deficient embryos exhibited more profound abnormalities, being extremely short and completely unpolarized (Fig. 1h). Although we cannot completely rule out that developmental delays contribute to the phenotype caused by *runx2b* deficiency, these results suggest that correct morphogenesis of some zebrafish facial muscles depends on correct development of the skeletal elements to which they attach.

### Myofiber cytoskeletal polarity depends on cartilage growth
Next we examined myofiber development in the adductor mandibulae (am) to understand the cellular basis of muscle disorganization in the

**Fig. 1 | Oriented growth of cartilage is essential for correct muscle patterning and directionality in zebrafish. a** Embryo of Tg(*col2:mCherry*) was stained with anti-DsRed (Col2) and anti-MyHC antibody at 56, 72, and 81hpf, the latter for which views from front, side and top are shown. **b** An embryo of Tg(*col2:mCherry, tbx1:Cre*) injected with actb2:loxp-mTagBFPcaax-loxp-EGFP and subjected to live imaging (left panels, see also Supplementary Movie 2, *n* = 1 embryo). Arrowheads indicate end points of muscle. **c** Brightfield images of facial profiles for control zebrafish embryos and the indicated cartilage-less zebrafish embryos at 81hpf (quantified in **d**). **d** The area of the mandibular region was measured for control embryos and the indicated cartilage-less embryos at 81 hpf. *P < 0.0001, unpaired two-tailed Student's *t*-test (*n* = 6 biologically independent embryos per condition). **e** The embryos of Tg(*col2:mCherry*) were injected with the indicated gRNA or MO, and stained with anti-DsRed (Col2) and anti-MyHC antibody. Magnified images of the internal region of lower jaw (dashed box) and the region surrounding hyosymplectic cartilage (box bracket) are shown in the lower left and right panel, respectively. Although maximum intensity projections are shown here,

Supplementary Movies 3, 4, and 5 show these Z-stacks at single myofiber resolution. **f** The graph shows mean length±SD of each muscle, for *n* = 12 control gRNA-injected embryos, 14 control MO-injected embryos, 10 *sox9* crispant embryos (*P < 0.0001, unpaired two-tailed Student's *t* test), and 12 *runx2b* morphant embryos (*P < 0.0001, unpaired two-tailed Student's *t* test). **g** Diameter of eye is shown. Values are means of 6 embryos ±SD. *P < 0.0001, unpaired two-tailed Student's *t* test. **h** Embryos from the outcross between Tg(*tbx1:CreERT2*) and Tg(*actb2:loxp:sstop:loxp-DsRed*) were treated with 4OHT as described in Methods section. Myofibrils (anti-MyHC staining) and individual muscle cells (anti-DsRed staining) are shown in the imp, ima, and am muscles of control (control gRNA-injected) and cartilage-reduced embryos at 81hpf. Scale bar; (**a**) and (**e**), upper panel, 100 μm. (**b**), (**e**) lower left, 50 μm. (**b**), (**e**) lower right, (**h**), 20 μm. (**c**), 500 μm. ep epiphysis. mc Meckel's cartilage, pq palatoquadrate. io inferior oblique, imp inermandibularis posterior, ima intermandibularis anterior, am adductor mandibulae, ih interhyoideus.

cartilage-deficient zebrafish (Fig. 2a). Analysis of myofiber length in 81 hpf embryos revealed significantly shorter myofibers in *runx2b*-morphants and *sox9a/b*-crispant embryos compared to controls (Fig. 2b). *Sox9*-crispant muscle contained significantly more thin myofibrils compared to controls (Fig. 2c), so we stained nuclei of myofibril-labeled embryos to assess cell-cell fusion (Fig. 2d). Indeed, both *sox9* and *runx2b* perturbation resulted in fewer numbers of nuclei per myofiber (Fig. 2e), suggesting myocyte fusion was partly inhibited in these models.

Additionally, we found that myofibrils in both types of cartilage-reduced embryos showed a distorted sarcomere structure (Fig. 2f). Sarcomere assembly is regulated by passive tension between muscle and tendon in *Drosophila*[33]. In vertebrates, the tension at the forming muscle-cartilage interface might promote myofibril development. Given that tensile strain can influence cell shape via actin fibers, we tested whether cartilage disruptions could cause alterations in actin cytoskeletal organization in early-developing myocytes. In control embryos between 48 and 81hpf, phalloidin staining revealed accumulation of actin fibers at the interface between muscle and cartilage. Here, filamentous actin fibers start and run along the longitudinal direction of the elongating muscle (Fig. 2g). Polarized fibers were abrogated in cartilage-reduced embryos at 52hpf (Fig. 2h), consistent with a model in which these embryonic structures failed to transmit force to differentiating myoblasts.

## Mature tendons are not required for myocyte polarization
Tendons produce the anchoring point that attaches muscle to cartilage in adults, so we assessed the role of tendon cells during muscle polarization. We explored the expression of *xirp2a*, a marker of zebrafish tenocytes[8]. *Xirp2a* mRNA accumulates in the tip of muscles (*xirp2a*+, MyHC+) and tendons (*xirp2a*+, MyHC-) near cartilage (Fig. 2i) at 72hpf. *Sox9*-crispant embryos showed diffuse and disorganized expression of *xirp2a*, although the overall pattern across the face was preserved. Also, the *xirp2a*+/MyHC+ muscle tips detached from *xirp2a*+/MyHC- tendon cells, corresponding to the reduced cartilage growth in this crispant. A similar result of muscle and tenocyte disorganization was revealed in the expression pattern of *col2a1*, *tnmd* and MyHC at 72hpf in a stable mutant line *sox9a^{tw37/tw37}* (Supplementary Fig. 1d). *Runx2b*-morphant embryos exhibited no *xirp2a*+ tendon cells with scattered muscle tips (Fig. 2i), and the shortage in tendon formation was confirmed by a loss of the *tnmd* expression pattern at 72hpf (Supplementary Fig. 1e). Thus, the amount of cartilage disruption correlates with phenotype severity of both myocytes and tenocytes, but a hierarchy of how each cell group contributed to the phenotype remained unclear.

It was previously reported that cranial tenocytes and myocytes are specified independently in zebrafish but that muscles are required for tendon maintenance and maturation[8]. Time-course experiments

revealed that *tnmd* expression was visible by 60hpf but not at 48hpf, suggesting muscles elongate significantly before tenocytes mature (Supplementary Fig. 2a). We performed CRISPR/Cas9-based genetic screening with gRNA pools targeted at the tenocyte matrix, and found minimal effect on musculoskeletal morphologies (Supplementary Fig. 2b and 2c, deleterious mutations were verified by Sanger sequencing shown in Supplementary Information). This suggests that establishment of myofibril polarization depends primarily on cartilage growth, and may not require mature tenocytes, or some components of the tendon-associated ECM (extracellular matrix).

We examined whether tenocyte precursors (expressing *scxa*, a transcriptional regulator of tenocyte cell identity) were present during facial muscle polarization. *Scxa* was absent at 44hpf (Supplementary Fig. 3a) and visible at 56hpf (Supplementary Fig. 3b) when using traditional whole mount in situ hybridization. Using RNA hybridization chain reaction technology (HCR, Supplementary Fig. 3c) *scxa* was found in facial mesenchyme as early as 48hpf (Supplementary Fig. 3d), after the initial attachment between myocytes and cartilage condensations. Similarly, in mouse embryonic lower jaw we found *MyoD* expression (visible at E11.5) preceded *Scx* expression (visible by E12.5) when using standard HCR in whole mount (Supplementary Fig 3e-j). When increasing the HCR probe concentration, Scx expression domains were obvious in the E11.5 ribs and limb but still not in jaw (Supplementary Fig. 3k–s). However, HCR of unfixed, fresh frozen sections revealed low levels of *Scx* expression in the E11.5 facial mesenchyme (Supplementary Fig 3t–v). Thus, tenocyte precursor cells might be present during muscle polarization in zebrafish and mouse, though *Scx* is apparently not exclusive to this cell type at early time points[34].

We mapped the developmental timeline of tendons near the AM muscle by assessing gene expression of *scxa*, *tnmd*, and *col1a2* (Supplementary Fig. 4a) and which also suggests that tenocyte precursors, but not mature tenocytes, could be present during AM muscle polarization. Labeling tenocytes in cartilage-less embryos revealed that both *tnmd* and *scxa* expression patterns were disrupted alongside cartilage growth defects (Supplementary Fig. 4b). It remains unknown to what degree the positioning of tenocyte precursors influences myoblast attachment to chondrogenic condensations, and this open question should be pursued in future work.

Taken together, these results reveal that cartilage growth regulates facial muscular development. Depending on its severity, cartilage growth defects cause disruptions in myoblast cytoskeletal organization and polarity, myofiber elongation and fusion, and the resulting muscle morphology.

## Cartilage promotes anisotropic muscle growth in mouse
The observation that cartilage development promotes muscle tissue architecture in zebrafish warranted validation in mammals.

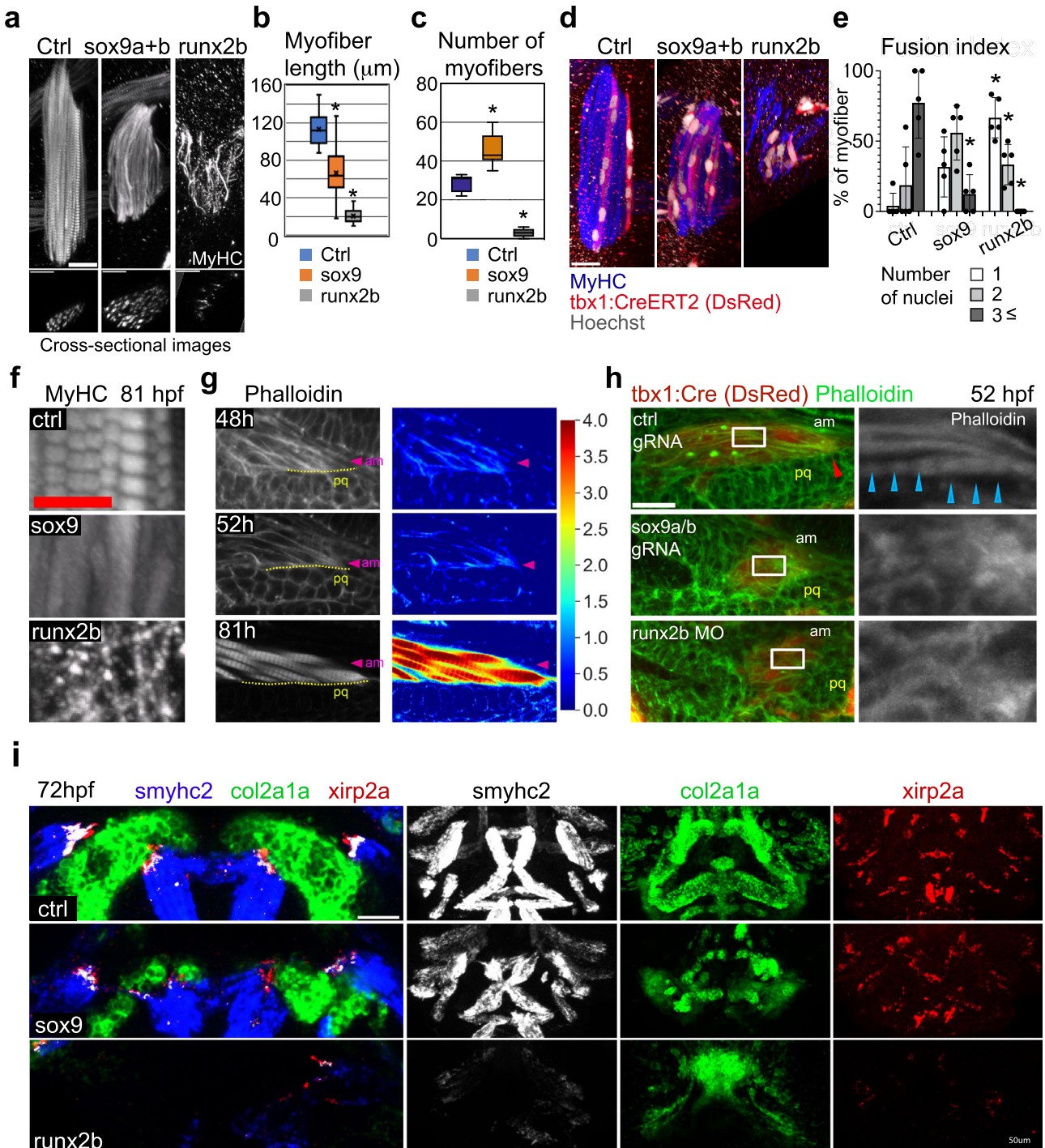

**Fig. 2 | Cartilage ablation disrupts myofiber development in zebrafish.**
**a** Myofibrils with cross-sectional images of am muscle at 81 hpf are shown. **b** The length of individual muscle cells at 81hpf are shown in the box plot (median bounded by IQR, whiskers=1.5x IQR). *P < 0.0001, unpaired two-tailed Student's t test (n = 27, 31, and 24 for ctrl, sox9, and runx2b, respectively). **c** Number of myofibers per muscle were quantified from cross-sectional images of am muscle at 81hpf and shown in the box plot (median bounded by IQR, whiskers =1.5x IQR). *P < 0.0002, unpaired two-tailed Student's t test (n = 7, 10, and 7 for ctrl, sox9, and runx2b, respectively). **d** Individual muscle cells at 81hpf are shown with internal nuclei. **e** Muscle cells at 81hpf were classified according to the number of nuclei and the percentage of muscle cells is shown. On average, 5 different myofibers were counted for nuclei per embryo. Values are means of n = 5 embryos ±SD. *P < .0001, two-sided Fisher's exact test. **f** The sarcomere structures of control and perturbed embryos are visualized with anti-MyHC staining at 81hpf. **g** The actin organization is

visualized in am muscle at different times in a control embryo. The same staining is shown as a heatmap on the right, in which 16-bit intensity value was normalized to that of cortical action in pq cartilage and log-transformed (base: 2). The boundary between am muscle and pq cartilage is shown with a dashed line. The pink arrowhead indicates the tip of the am muscle. **h** The actin organization is visualized in am muscle at 52hpf in different perturbation conditions. The box area is magnified and shown below. The boundary between am muscle and pq cartilage is shown with dashed line. Red arrowhead shows actin at the cartilage-muscle interface. Blue arrowheads show the actin structure of myofibrils. Images are 10 μm oblique slices. **i** The expression patterns of the indicated genes in control and perturbed embryos are visualized with in situ hybridization at 72hpf. Scale bar; (**a**), (**d**), (**h**), and (i, left inset) 20 μm. **f** 5 μm. (**i**, right panels) 50 μm. pq palatoquadrate, am adductor mandibulae, IQR interquartile range.

We visualized mouse embryonic muscle and cartilage formation in whole mount using HCR[35,36]. Similar to the facial development of fish, mouse myoblasts (*MyoD*+) and chondrogenic cells (*Col2a1*+) emerge nearly at the same time within a close distance at E11.5, where the ends of muscle groups appeared in contact with induced cartilage (Fig. 3a, arrow heads, and Supplementary Movie 6). *MyoD*+ cells in these contact points expressed MyHC, a terminal differentiation marker, indicating the more advanced state of these cells in differentiation (Fig. 3b). Moreover, in the intercostal region, the alignment of MyHC+ cells was correlated with their locations relative to cartilage: muscle cells were aligned between the stripes of intercostal cartilage whereas they remained unaligned in the ventral region where cartilage was not yet developed (Fig. 3c). At E12.5, muscle and cartilage produced a developed interconnected pattern (Supplementary Movie 7) and exhibited a correlated growth pattern at later embryonic stages (Fig. 3d–g, see below). These results suggest that cartilage plays a role in myofiber alignment from the earliest stages of muscle development in mouse.

To challenge this idea, we examined the consequences of cartilage loss on muscle growth. To specifically ablate developing cartilage, we used a transgenic mouse strain that produces diphtheria toxin subunit A (DTA) after *Cre* recombination under the control of the *Col2* promoter in the presence of tamoxifen (*Col2a1:CreERT2/R26DTA*)[26]. We injected 3 mg of tamoxifen at E12.5 and E13.5, a stage with rapid musculoskeletal development and growth. We then subjected the fixed and PTA-contrasted E15.5 embryos to micro-CT 3D imaging. Limb musculoskeletal tissues were smaller in DTA mutants compared to controls (Fig. 3d), evidenced by decreased cartilage volume (Fig. 3e). Muscle volumes were slightly but not significantly reduced (Fig. 3e), however muscles were significantly shorter (Fig. 3f), indicating specific effects on growth anisotropy rather than a general growth defect. Additionally, the limb muscles in DTA mutants showed abnormal morphology and made contact with each other (Fig. 3g).

In addition to the limb, outgrowth of the frontal face was markedly inhibited (Fig. 3h). Here, tongue muscle is integrated with muscles running along the dorsoventral axis, which connect it to Meckel's cartilage, hyoid cartilage, and cartilage in the ear area at E17.5 (Fig. 3i left, white and yellow arrowheads). At E11.5, parts of these muscle develop along with the lateral axis, being connected to nascent cartilage (Fig. 3a light and dark blue arrow heads). This suggests that these muscles are stretched along the dorsoventral axis by the growing cartilage from E11.5 to E17.5, and change their directionality and shape. The tongue muscles of the DTA mutant displayed an abnormal structure and appeared shrunken (Fig. 3i middle column). Specifically, the muscle attached to Meckel's cartilage in front showed a remarkably severe phenotype (see yellow arrow and highlighted part in Fig. 3i). This might be related to late development of this part after E11.5, corresponding to the timing of tamoxifen treatment which reduces facial cartilage after that stage. Taken together, these experiments reveal that cartilage development influences growth anisotropy of skeletal muscle in both mouse and zebrafish.

### Cartilage attachment points drive myocyte polarization

To elucidate the cellular mechanism that aligns myofibrils, the timing and location of myocyte polarization was assessed using live imaging in zebrafish. For this, we focused on the AM muscle, which connects to palatoquadrate and Meckel's cartilage and, thus, is easy to image in zebrafish. We used *fli1:GFP* to track cartilaginous cells, where we could detect differentiating and mature chondrocytes via their characteristic morphologies. Our live imaging data revealed that the primordium of the AM muscle starts to interact with the presumptive chondrogenic condensations of the palatoquadrate between 42 and 45 hpf (Fig. 4a, b, and Supplementary Movies 8 and 9), and this interaction continues throughout the time frame of cartilage and muscle differentiation.

After the apparent adhesion of individual myocytes to emerging cartilage, the myocytes condense into a cluster, as seen in the reconstructed sequence of events (Fig. 4a). Subsequently, individual muscle cells align 90 to 60 degrees towards the presumptive chondrocytes. The polarization of individual myocytes starts from a contact point with developing cartilages, then propagates to the whole cluster of myocytes until every myocyte is oriented, before stretching and fusion events commence (Supplementary Movies 8 and 9). Next, the cluster of individual myocytes becomes elongated along the direction of the expanding cartilage (Fig. 4a, from 49hpf onward). This suggests that the cluster of myocytes anchored to the Meckel's and palatoquadrate cartilages become stretched by the displacement of their attachment points during cartilage growth. These events appear to determine muscle orientation (Supplementary Movie 8).

Cartilage ablation, as predicted, had a dramatic impact on myoblast cell behavior during muscle morphogenesis (Fig. 4b–d). In *sox9*-deficient embryos, the initial polarization was delayed for more than 8 h later than controls (Fig. 4b middle row). The polarized muscle cells in *sox9*-crispants were not able to maintain a stable contact with the reduced cartilage, continuously moved up and down (Supplementary Movie 9), and failed to elongate afterwards. Myocytes from fish with downregulated *runx2b* and correspondingly missing cartilage never acquired any polarization and failed to extend (Fig. 4b bottom panels and Supplementary Movie 9). Quantifications of aspect ratio (Fig. 4c) and cell orientations (Fig. 4d) reflect the failure of myocytes to polarize and elongate properly in cartilage-deficient conditions. Nuclei counts of the AM muscle across all conditions confirmed that rates of cell proliferation were not affected by these conditions, suggesting the loss of cartilage affected myocyte cell shape (Supplementary Fig. 5). In line with this, myoblasts located near the forming cartilage condensations in wild-type embryos always appear polarized and stretched, whereas more distant myocytes remain unpolarized and un-stretched (Fig. 4e, f). In addition, myoblast nuclei near cartilages showed a flattened shape resembling nuclei in mature muscle, whereas nuclei located distantly from cartilages remain spherical (Supplementary Movie 10).

### Muscle attachment to growing cartilage produces tension

Our live imaging data indicated that stretched myocytes are exposed to mechanical forces generated by the expanding and elongating cartilage. To test this possibility, we characterized the mechanical tension in the stretched muscle primordia. One way to reveal the existence of tension in viscoelastic biological systems is through laser ablation, which removes all external forces and allows the system to elastically recoil to its tension-free equilibrium length. Hence, if the material recoils from the ablation site, it can be reasonably concluded that it was under tension.

We ablated the attachment point between myocytes and cartilage (Fig. 4g, the region encompassed with bracket) with 2-photon laser, and observed the response with live-imaging. After cutting, the tip of myocyte appears to detach and drift away from the ablated sites along the axis of the muscle tissue (Fig. 4g, Supplementary Movie 11). We measured the distance between the tip of the myocyte and the ablated site, and found that the recoil velocity decreased over time (Fig. 4h), indicating a viscoelastic response that depends on the tension and friction-like resistance in muscle primordia. Importantly, this model assumes no active cell migration, but rather that external forces cause the cells to move after they have been detached from the cartilage by laser ablation.

Thus, either loss of cartilage growth, or loss of primary attachment points might explain polarization defects caused by genetic perturbation of cartilage development. Addressing this, we ablated nascent cartilage with 2-photon laser just before myoblast polarization at 42hpf in the developing zebrafish head. As a consequence, the unpolarized cells reorganized cell shape and orientation immediately

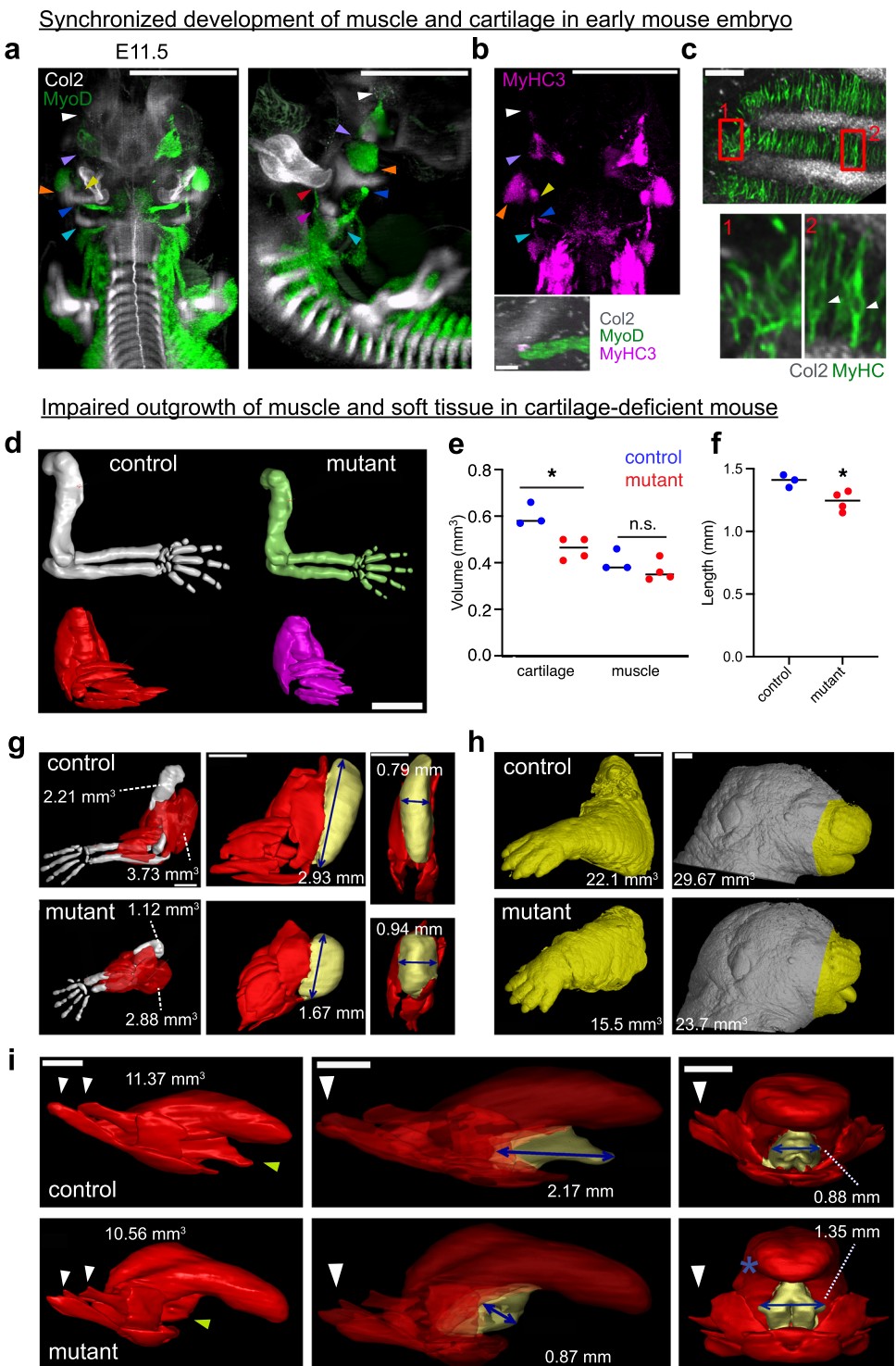

Fig. 3 | **Cartilage development coincides with muscle and controls directional outgrowth in the mouse embryo. a**, **b** Localizations of *Col2*, *MyoD*, and *Myh3* transcripts are visualized with whole mount RNA HCR in E11.5 wild-type embryo. Front and side views of 3D rendering images are shown. Arrowheads indicate the contact points between cartilage and muscle. The lower panel in (**b**) corresponds to the region surrounding the dark blue arrowhead in the upper panel. **c** Localizations of the *Col2* transcript and MyHC protein in the intercostal region are shown in an E11.5 embryo. Images are 2 μm optical sections. The window 1 and window 2 correspond to the regions of undeveloped and developed cartilage, respectively. The arrowheads indicate nuclei, which lack the MyHC signal. **d** MicroCT scan of E15.5 control and cartilage-ablated DTA embryos. Cartilage (upper panel) and muscle (lower panel) in arm are shown. **e**, **f** The plots show the total volume of cartilage and muscle in the arm and muscle length in the control versus DTA embryos. Significant differences in muscle length (*p = 0.0265), cartilage volume (*p = 0.0111), and muscle volume (p = 0.284), between the control and DTA embryos were determined by an unpaired two-tailed Student's *t*-test (*n* = 3 control embryos from 3 separate litters, 4 DTA embryos from one litter). **g–i** Control and DTA embryos of E17.5 were subjected to microCT scanning. Cartilage and muscle of the arm (**g**), the overall structures of face and arm (**h**), and tongue muscle (**i**) are shown (*n* = 1 tongue, face, and whole arm segmented each for control and DTA). Some parts of muscle are highlighted with yellow for comparison. Attachment points of tongue muscle to cartilage are indicated by arrowheads. Scale bar; (**a**), (**b**) upper panel, (**d**), (**g**), (**h**) and (**i**), 1 mm. (**b**) lower panel, (**c**), 100 μm.

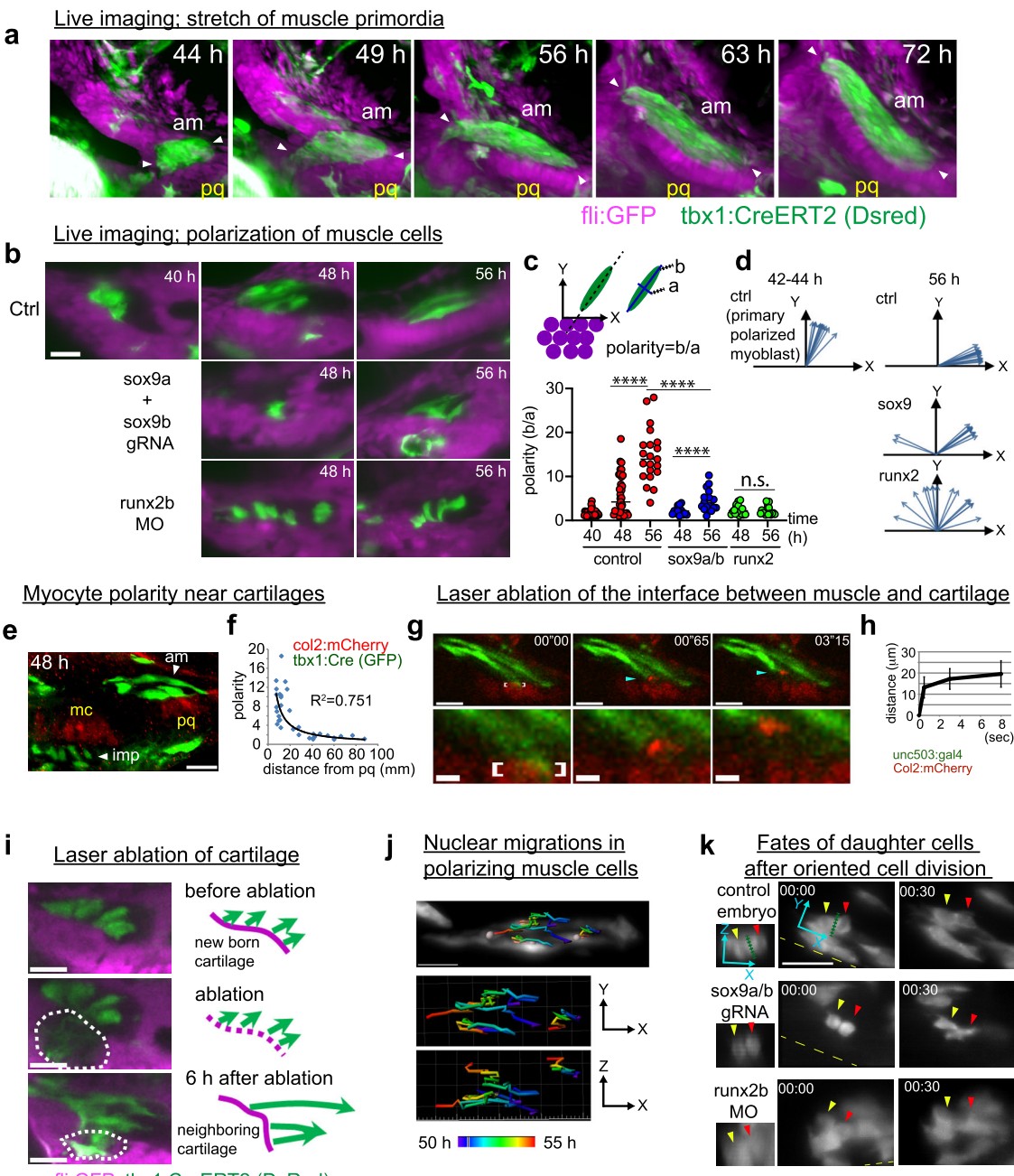

**Fig. 4 | Time-lapse imaging reveals that physical attachment to cartilage influences polarization of myocytes. a** Embryos from the outcross between Tg(*actb2:loxp:sstop:loxp-DsRed*) and Tg(*fli:GFP, tbx1:Cre*) were subjected to live imaging. See also Supplementary Movie 8. **b** Differentiating muscle (*tbx1:CreERT2*) and condensing cartilage (*fli:GFP*) cells were tracked as in (**a**) in control and perturbed embryos. The plots show the extent of polarization in cell shape in (**c**) and the orientation of muscle cells (**d**) relative to pq cartilage. Primary polarized myoblast corresponds to polarized myoblasts adjacent to pq cartilage in the early stage. See also Supplementary Movie 9. Control 40hpf, n = 28 cells. Control 48hpf, n = 33 cells. Control 56hpf, n = 21 cells. *Sox9*-crispant 48hpf, n = 15 cells. *Sox9*-crispant 56hpf, n = 18 cells. *Runx2b*-morphant 48hpf, n = 16 cells. *Runx2b*-morphant 56hpf, n = 18 cells. Statistical significance in (**c**) was determined by two-way ANOVA (****P < 0.0001) **e, f** The relations between polarity of cells and the distance from pq cartilage in control embryo. Images are 10 μm oblique slices (**e**). Graphs shows scatter plot with best fit power trendline (**f**). 33 cells were measured. The distance was defined as length between the center of pq cartilage and the closest point of cells to it or center of nuclei. See also Supplementary Movie 10. **g** The interface between am muscle and pq cartilage was ablated by laser pulse in the embryo from an outcross of Tg(*unc503:gal4VP16*) and Tg(*col2:mCherry, UAS:GFP*) around 54hpf. The area of ablation is shown with brackets. The time when the laser-pulse was started is set to 00″00. **h** The distance from the tip of the muscle cell (light blue arrowhead on (**g**)) from the ablated spot was measured and shown. Values are means of 10 embryos ±SD. See also Supplementary Movie 11. **i** The Pq pre-cartilage condensation of a 42hpf embryo was laser-ablated, and the polarization of muscle precursor cells were examined with live imaging. The area encompassed by the dashed line was the ablated region. See also Supplementary Movie 12. **j** The trajectory of nuclei in differentiating muscle cells in a control embryo. See also Supplementary Movie 13. **k** Cell division and the fate of daughter cells (arrowheads) 30 min after cell division were shown in control and cartilage-less embryos. The longitudinal axis of myofibril close to the dividing cells was set as X axis (yellow dash). Images are 3 μm oblique slices. Green dotted line in control shows the division plane. See also Supplementary Movie 14. Scale bar; 20 μm except in magnified part of (**g**) (5 μm).

after the laser ablation of a previous attachment and re-polarized to the cartilage next to the ablated region (Fig. 4i, Supplementary Movie 12). We did not assess downstream alterations in the nearby ECM or signaling environment following laser ablation. Nonetheless, these live imaging experiments suggest that cartilage growth promotes myoblast polarization via tension at attachment sites, and that myoblasts can adaptively discover new attachments to cartilage if primary attachment sites are lost.

## Cartilage helps to control directional myocyte fusion

In the next phase after initial myocyte shaping and stretching, the developing myocytes displayed many features of polarized cells, such as nuclear migration, oriented cell division, and cell fusion along the anteroposterior axis of the AM muscle. Rapid back-and-forth migration of the nuclei was observed in these myocytes (Fig. 4j, and Supplementary Movie 13) similarly to the interkinetic nuclear migrations in radial glial cells of the developing central nervous system[37]. Additionally, muscle precursor cells divide along the longitudinal direction of nascent adjacent myotubes (Fig. 4k and Supplementary Movie 14). After cell division, two daughter cells remain in contact while keeping their alignment, and then fuse (Fig. 4k upper panels). This bears similarity to the directional cell division of adult muscle satellite cells, which is regulated by the basal lamina of adjacent myofibrils[38]. We tested whether the orientation of cell division and cell fusion is regulated during cartilage-dependent polarization (Fig. 4k and Supplementary Movie 14). Cartilage-reduced embryos also showed myocytes dividing along the direction of the proximodistal axis of the AM muscle similar to control (based on four samples each from *sox9a*-crispants and *runx2b*-morphants), but the following tandem mode of fusion between daughter cells was dysregulated (Fig. 4k lower panels and Supplementary Movie 14). Thus, despite evidence that cartilage is required for initial myocyte cytoskeletal polarization, directional elongation, and fusion, we cannot rule out an alternative, cartilage-independent mechanism promoting oriented mitosis in facial myocytes.

## Cytoskeletal contractility promotes myofiber elongation

We next examined the mechanism of how cartilage regulates polarized myofiber growth by decoupling passenger from driver effects. Given that cytoskeletal organization is a key mediator of cell polarity in general, our observation of disrupted actin fibers in cartilage-deficient embryos (Fig. 2h) suggested that cytoskeletal function mediates the myocyte response to cartilage growth. We blocked the contractility of actin fibers by exposing control embryos to blebbistatin, which is a potent myosin II inhibitor. We treated embryos with blebbistatin from 36 to 52hpf (Fig. 5a), when AM muscle attaches to palatoquadrate and Meckel's cartilage, and individual myocytes are stretched in the cluster (Supplementary Movies 8 and 9). Blebbistatin-treated embryos showed misoriented (Fig. 5a, middle) or unpolarized (Fig. 5a, right) muscle cells. Then we treated embryos from 55 to 72hpf, when AM muscle dynamically changes the directionality with the translocation of Meckel's cartilage (Supplementary Movie 8). This late stage of drug treatment led to milder phenotypes, with misaligned myofibrils and spread end points of AM (Fig. 5b) and eye muscles (Fig. 5c, left). The internal muscles of the lower jaw exhibited more severe defects in muscle cell orientation (Fig. 5c, right), which is probably due to the late onset of cell polarization in these muscles (Supplementary Movie 2). Treatment with the inhibitor BTS[39], which is highly specific for skeletal muscle myosin II, resulted in milder disorganization of lower jaw muscles (Supplementary Fig. 6a, b), reflecting a possible involvement of non-muscle myosin in the phenotype of blebbistatin-treated embryos. Treatment with a less potent myosin inhibitor BDM[40] also caused a mild muscular phenotype (Supplementary Fig. 6c, d). To confirm drug specificity, active (-) blebbistatin was compared to its inactive (+) form (Supplementary Fig. 6e–l). Treatment from 36–52hpf

with either form of blebbistatin did not significantly affect the nuclei count in AM muscles (Supplementary Fig. 6h), while the active (but not inactive) blebbistatin resulted in more numerous myofibers per AM muscle (Supplementary Fig. 6l). These results suggest that myosin II-driven contractility might complement cartilage growth to generate tension during myofibril polarization.

Another cytoskeletal feature disturbed in cartilage-deficient embryos was the sarcomere. *Titin*, a giant muscle-specific cytoskeletal regulatory protein that stabilizes the sarcomere structure, is proposed to act as a force and strain sensor[41]. We found that some filamentous actin in differentiating muscle has a periodical structure (Fig. 2g, see arrowheads), suggesting titin-dependent myofibril assembly, even at early stages. We used CRISPR/Cas9 to edit *titin* (*ttn.1* plus *ttn.2*, editing verification in Supplementary Information), and found that crispant embryos harbored bubble-like myofibrils (Fig. 5d) that were significantly shorter in length (Fig. 5e). Individual myocytes that attached to cartilage did not fully extend, and halted halfway in the muscle group (Fig. 5f, arrow heads). Therefore, Titin is apparently important for establishing the sarcomere to ensure correct myofibril morphology and elongation, but might not be a key protein for polarization.

Importantly, both blebbistatin-treated and *ttn*-crispant embryos had fewer nuclei per myofiber (Fig. 5g). For *ttn*-crispants the low nuclei per myofiber could be explained by a proliferation defect as we observed fewer nuclei per entire muscle (Supplementary Fig. 6m–p) without changes in myofibril number (Supplementary Fig. 6q–t). On the other hand, blebbistatin-treated embryos had significantly more myofibers per muscle (Supplementary Fig. 6t) and comparable number of nuclei per entire muscle versus controls (Supplementary Fig. 6p), overall suggesting a fusion defect.

The measurements indicate that these molecules influence myocyte fusion, either directly or indirectly via defects in cell proliferation/migration. Taken together, it seems that tensile forces generated by attachment to the growing cartilage cooperate with the myocyte cytoskeleton to regulate myofiber assembly.

## Cell fusion is not required for myocyte polarization

Because cell fusion was disrupted in all conditions leading to reduced myofiber length, we also downregulated *jamb*, a gene encoding a transmembrane protein required for muscle cell fusion in zebrafish[42], with a morpholino. No structural abnormalities were observed in morphant embryos (Fig. 5h) and the muscles were correctly polarized (Fig. 5i). Myofibers in the morphant were slightly but significantly less elongated compared to controls (Fig. 5j, $p = 0.027$, unpaired two-tailed Student's *t* test) despite an obvious fusion deficit (Fig. 5k). This result confirms that cell fusion itself is not required for defining the primary direction of muscle cells. This is also supported by the finding that unfused myocytes are oriented correctly in the early muscle development of mouse embryos (Fig. 3c, white arrow heads indicate nuclei). Hence, cartilage growth promotes myocyte fusion, which in turn promotes further myofiber growth to some degree.

## Cartilage growth mediates Wnt/PCP effects on muscle shape

A previous study showed that WNT11 expressed by the neural tube polarizes the trunk muscle through the non-canonical PCP pathway[10]. To examine the involvement of the WNT family in cartilage-regulated oriented events of muscle, we silenced the function of *wnt11*, *wnt5b*, and *wnt5a* in zebrafish using CRISPR/Cas9 and verified gene-specific editing by Sanger sequencing (Supplementary Information). *Wnt*-crispants recapitulated known *wnt* phenotypes such as short body length in *wnt5b*-crispants (ref.[43] Supplementary Fig. 7a) and short distance between the eyes of *wnt11*-crispants (ref.[44] Supplementary Fig. 7b). Downregulation of *wnt11* or *wnt5b*, but not *wnt5a*, caused the deformation of the facial structure including cartilage, showing their general requirement for the skeletal shape formation (Supplementary Fig. 7c). In these mutants, muscle formed according to the misshaped

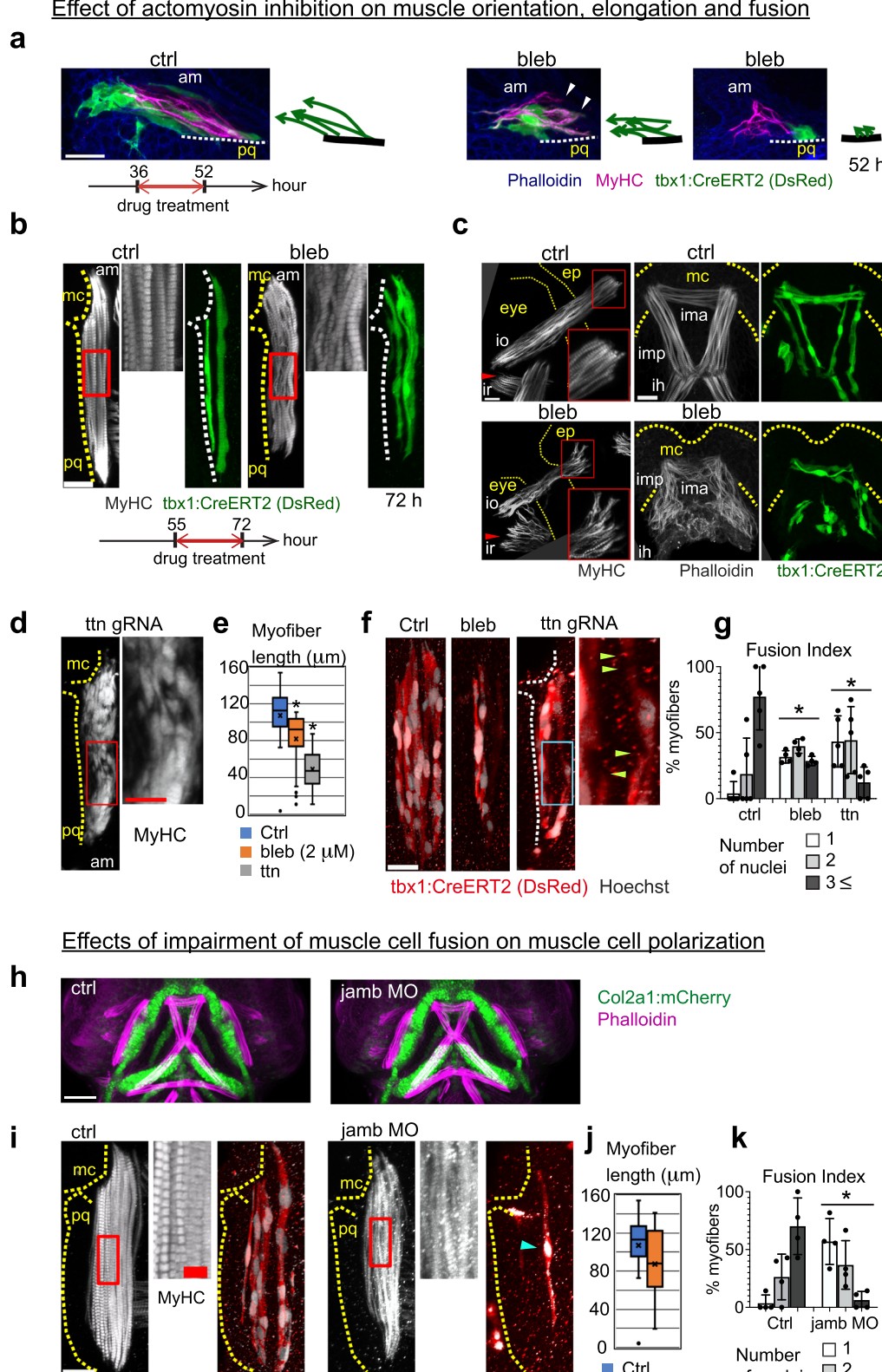

Effect of actomyosin inhibition on muscle orientation, elongation and fusion

Effects of impairment of muscle cell fusion on muscle cell polarization

cartilage, and exhibited an abnormal shape. However, individual myofibrils showed correct alignment and were perfectly polarized in jaw (Supplementary Fig. 7d) and AM muscles (Supplementary Fig. 7e). Furthermore, muscle lineage-specific expression of a *dvl* mutant, which dysregulates the WNT/PCP pathway, did not affect the muscle orientation (Supplementary Fig. 7f). We also examined the effects of the WNT gradients on the polarity of C2C12 mouse myoblasts in vitro,

both by co-culture with WNT-expressing cells (Supplementary Fig. 7g) and by the use of commercial microfluidic gradient-generating culture models (Supplementary Fig. 7h) but did not find support of direct polarization by WNTs.

We cannot fully rule out a direct role of WNT/PCP in controlling facial of myocyte polarization, partly because WNT regulation of muscle formation differs along the body axis[45], and different WNTs

**Fig. 5 | Cytoskeletal contractility is important for oriented muscle cell growth and fusion. a** Myofibrils (MyHC) and individual muscle cells (DsRed) are visualized after the treatment of blebbistatin (10 μM) for the indicated period. The orientation of muscle cells (DsRed) to cartilage is shown with drawing. Arrowheads indicate the end points of muscle cells that are detached from pq cartilage. Images are 6 μm oblique slices. **b, c** Embryos were treated with blebbistatin (10 μM) for the indicated period and the orientation of myofibrils and myocytes in the am muscle and eye/jaw muscles were examined as in (**a**). The boxed areas are magnified. The cartilage and eye region are outlined with dashed yellow lines. Note that blebbistatin treatment of late stage led to misalignment of myofibrils (**b**, see magnified image) and errant convergence of myofibrils at end points (**c**, see red box and arrowhead). **d–g** Myofibrils (MyHC, shown in **d**) and individual muscle cells (DsRed) with internal nuclei (**f**) are shown in control (control gRNA-injected) and crispant (ttn.1 gRNA plus ttn.2 gRNA-injected) embryos at 81hpf. The boxed areas are magnified and shown adjacent. Arrowheads indicate end points of muscle cells left in the cluster. The length of individual muscle cells is quantified and shown in box plot (median bounded by IQR, whiskers =1.5x IQR) in (**e**). *P < 0.0001, unpaired two-tailed Student's t test (n = 27, 30, and 21 for control, bleb, and ttn, respectively). **g** Muscle cells were classified according to the number of nuclei and the percentage of muscle cells is shown. Values are means of n = 4 embryos ±SD. *P < 0.004, two-sided Fisher's exact test (n = 5 embryos in the case of ttn). Blebbistatin (2 μM) was administered from 48 to 81hpf for this assay. **h** Col2a1:mCherry and phalloidin staining for control and jamb-morphant embryo. **i–k** Myofibrils (MyHC) and individual muscle cells (DsRed) with internal nuclei are shown in control and jamb MO-injected embryos at 81hpf. The regions of red boxes are magnified and shown in the next panel. Cartilage (pq and am) is outlined with dashed yellow lines. **j** The length of individual muscle cells is quantified and shown in box plot (median bounded by IQR, whiskers = 1.5x IQR). P = 0.027, unpaired two-tailed Student's t test (n = 35 and 27 for ctrl and morphant embryo, respectively). **k** Muscle cells were classified according to the number of nuclei and the percentage of muscle cells is shown. Values are means of 4 embryos ±SD. *P < 0.0001, two-sided Fisher's exact test. The blue arrowhead indicates a cell containing single nuclei. Non-treated embryos served as control. Scale bar; 20 μm except magnified image of (**d**) (10 μm), (**h**) (70 μm), and magnified image of (**i**) (5 μm).

may be involved. However, our results did not support such a direct role, and rather support an indirect role via effects on skeletal growth. Future experiments using tissue-specific and inducible transgenes will be needed to test this idea.

### Attachment sites drive myofiber polarization in vitro

To directly examine the effects of mechanical stretch on muscle cell morphology, we used PDMS (polydimethysiloxane) membranes (Fig. 6a) on which cells can be physically stretched up to 20% (Fig. 6b). C2C12 cells seeded on laminin-coated PDMS membranes were incrementally stretched as soon as differentiation was induced, and subjected to live-imaging. Hours after the onset of stretch, C2C12 cells started migrating in the direction of stretch, and polarized with their cell shape oriented along the stretching axis (Supplementary Movie 15). We also found that the direction of cell division was aligned with the stretch direction (Fig. 6c). In a second experiment, C2C12 cells were also stretched starting at 20 h after differentiation, just before cells started to show spindle-shaped morphology. While unstretched myotubes (Fig. 6d) showed no particular orientation (Fig. 6e) we found that stretched myotubes (Fig. 6f) were aligned within 30 degrees oblique to the direction of stretch (Fig. 6g). In unstretched controls, we observed branched myotubes, which are caused by end-to-lateral fusion, whereas myotubes in the stretched condition were straight and more elongated (Fig. 6d, f, insets). We treated the stretched C2C12 cells with $Gd^{3+}$ (gadolinium), a $Ca^{2+}$ channel blocker that inhibits the function of stretch-induced ion channels[46–48], and found that myocytes were aligned along the stretched axis but cell fusion was blocked and the size of myotubes was decreased (Supplementary Fig. 8a). Also, the $Gd^{3+}$ treated zebrafish embryos exhibited myofibrils with reduced number of nuclei (Supplementary Fig. 8b–d). Therefore, stretch-induced ion channels are not required for stretch-induced polarization, but might play a role in muscle growth through fusion.

Furthermore, 3D microtissue experiments suggested that two artificial attachment points are sufficient to produce static tension in individual muscle cells and effectively orient myofibrils, an effect that can be potentiated by stretch (see below). To explore this in detail, we generated C2C12-derived muscle microtissues in a 3D environment to investigate the effects of continuous mild stretch on myofibril orientation, differentiation, and growth (Fig. 6h). We utilized a custom-build system of two PDMS pillars as described previously[49,50]: C2C12 cells per tissue were mixed with a collagen-based extracellular matrix and seeded around the pillars. Within 24 h, the cells compacted the matrix and self-organized into highly oriented, dog bone-shaped muscle tissues anchored by the pillars at both ends (Fig. 6i). After 24 h, differentiation was initialized.

In addition to being artificial attachment points, such pillars also serve as force sensors, since their deflection (as a result of tissue

contraction) can be directly measured and converted into contractile tension. We measured high levels of static contractile tension in the formed tissues (Fig. 6j, 705 ± 58 Pa, n = 28), which we hypothesized to determine the orientation of individual cells. When continuous stretch was applied to the tissue, static tension significantly increased (1062 ± 99 Pa, n = 21), and the muscle tissue was more polarized (Fig. 6k).

Similar to the in vivo situation with natural muscle attachment points, the pillar positions dictated the directionality of coalescing muscle tissue. We compared artificial tissues seeded with single (Fig. 6h), dual (Fig. 6i), and multiple attachment points (Fig. 6l). When we provided two attachment points, the individual myocytes were aligned along the pillar-to-pillar axis, and they formed a polarized and highly aligned muscle tissue, expressing α-actinin, a muscle-specific structural protein (Fig. 6m, boxes 1 and 2). However, cell polarity was more uniform along the edges than in the very center, suggesting that the tissue edge might also influence cell polarity (Fig. 6m, compare boxes 2 and 3). When myoblasts were seeded without a second attachment point, cells attached to the single pillar also expressed α-actinin (Fig. 6n upper), suggesting that attachment to a stiff object can induce differentiation. However, individual myotubes did not show directional consistency, and the overall tissue structure ended up in un-polarized aggregates (Fig. 6n lower). This result confirms that mechanical tension in a muscle cluster between two attachment points is sufficient to guide cell orientation.

Next, we looked into the cases of 3 or 4 attachment points (Fig. 6o, p). In these cases, myotubes near the pillar align following the axis pointing to the nearest pillar (Fig. 6o see boxes 1 and 3, and Fig. 6p, see box 1). Where most cells in the 2-pillar condition are either near a pillar or the tissue edge, 3-pillar and 4-pillar conditions feature large central regions of cells, distant from any directional input. Many cells positioned in these central areas of the tissue displayed random orientation, suggesting that increasing distance from attachment points, or distance from the tissue edges, reduces the effectiveness of directional cues for coherent cell orientation (Fig. 6o, p, inset windows 2). In these artificial 3D muscle tissues, polarization and orientation of individual muscle cells is entirely self-organized and only guided by the attachment points and the ensuing cell-generated mechanical tension that appears to primarily build along the tissue edges. These results show that mechanical tension is sufficient to drive directionality, elongation, and controlled fusion of individual muscle cells, but that this can be potentiated by muscle tissue shape.

### ECM promotes muscle attachment and directionality

The importance of physical cues for the directional growth of muscles led us to characterize the adhesion proteins that allow myocytes to attach to cartilage, and possibly transmit force. Cells can sense

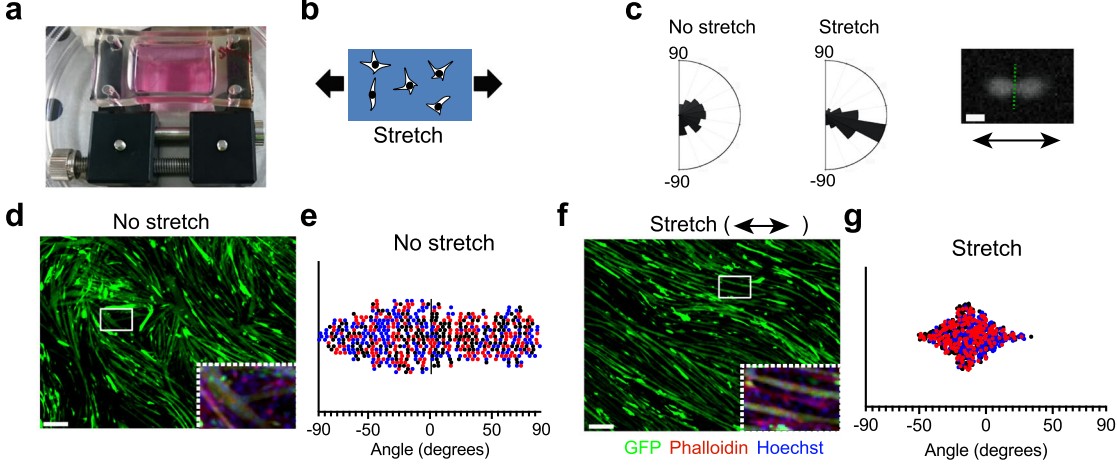

Effects of mechanical stretch on muscle cell orientation in 2D culture

Effects of mechanical forces on cell orientation in 3D environments

mechanical tension through members of the integrin family, among which integrin β1 plays a central role. In the developing zebrafish embryo, integrins are broadly expressed in facial cartilage, muscle, and connective tissues[51]. Specifically, *itgb1a* and *itgb1b* transcripts can be detected in both chondrocytes and myocytes at 56 hpf (Supplementary Fig. 9a). We performed CRISPR/Cas9 editing of *itgb1a* and *itgb1b*, two zebrafish paralogues of the *Itgb1* gene, and verified editing by

Sanger sequencing (Supplementary Information). The crispant embryos showed profound abnormalities in muscle, affecting polarization (Supplementary Fig. 9b), nuclear positioning (Supplementary Fig. 9c), myofiber length, (Supplementary Fig. 9d) and fusion (Supplementary Fig. 9e). The extent of abnormalities of muscle and cartilage in *itgb1*-crispant embryos appeared similarly severe to that of *runx2b*-deficient embryos, in which cartilage development is severely

**Fig. 6 | The attachment points and directional forces are sufficient for muscle orientation and alignment in vitro. a** The 2D cell-stretcher device. **b** Cartoon describing the in vitro stretching experiment. **c** C2C12 cells were seeded on laminin-coated PDMS membrane, and transfected with GFP. Cells were induced to differentiated and then immediately stretched 20%. Differentiating cells were subjected to live imaging (See also Supplementary Movie 15). The plot shows the distribution of the angle of cell division. 0 degree corresponds to the axis of stretch. About 201 and 291 cell divisions were analyzed for the non-stretch and stretch conditions, respectively. **d–g** GFP-transfected cells were either unstretched or stretched 20% from 20 h of differentiation. Cells were fixed at 3.5 days after differentiation and analyzed for cell orientation. Angles of control (**d**, **e**) and stretched (**f**, **g**) myotubes were measured relative to the direction of stretch (set to 0 degrees). Plots show 200 myotubes per condition, per experiment, from three independent experiments (signified by black, red, and blue dots). The dashed box area is magnified and shown with phalloidin and Hoechst staining. **h** C2C12 cells were seeded with the mixture of Collagen-R, Collagen-G, and Geltrex in 3D chamber containing PDMS-coated pillars. After 24 h, cells were induced to differentiate. **i–k** Cells surrounding 2 pillars (non-stretch and stretch) were visualized with bright-field microscopy 8 days after differentiation. Static tension (**j**, *P = 0.002) and tissue width (**k**, *P < 0.0001) were measured and shown in box plots (defined as median bounded by IQR, whiskers =1.5x IQR). Significance was determined by an unpaired two-tailed Student's *t* test (n = 28 non-stretched and 21 stretched samples across two independent experiments.) **l** The 3-pillar and 4-pillar devices. **m** For cell orientation analyses, tissues were stained with anti-α-actinin antibody, and the orientation of individual myotubes was analyzed. Because of the fragility of stretched tissues, non-stretched tissues were used for the assay of 2 pillars. The axis connecting two pillars was set as 0 degree. Myotubes from 6 different tissues were analyzed for the orientation, and shown in the histogram. Box 1 represents the area close to the pillar, whereas box 2 represents the middle of the tissue edge, and box 3 represents the very center of the tissue. 213, 204, and 231 myotubes were analyzed for the regions represented by boxes 1, 2, and 3, respectively. **n** Cell orientation analysis of tissues with only one attachment point, stained for α-actinin. 201 myotubes from 3 different tissues were analyzed for the orientation, and shown in the histogram. **o, p** The orientation of individual muscle cells were analyzed after α-actinin staining in the case of 3 and 4 pillars. The angle of cells near the indicated pillar (box 1 or 3) or at the center of the tissue (box 2) was measured. For the 3-pillar setup, 222 (box 1), 232 (box 3), and 273 (box 2) myotubes from 3 different tissues were counted. For the 4-pillar setup, 268 (box 1) and 273 myotubes (box 2) were counted. The horizontal axis connecting two pillars was set to 0 degrees. Scale bar; (**d**), 200 μm. (**c**), 20 μm. **m** through (**p**), white, 500 μm. (**n**) through (**p**), red, 50 μm.

impaired. Although we cannot rule out whether inactivation of *itgb1* might cause muscle-autonomous effects, these findings are consistent with a model in which mechanical sensing via integrins guides oriented muscle development in vivo. This conclusion is further supported by the fact that the eyes developed normally in the *itgb1* mutant, while the eye-contacting muscles lost their orientation and structure (Supplementary Fig. 9f).

We reasoned that cell adhesion to the extracellular matrix is likely to affect muscle attachment to cartilage. We first examined the localization of laminin, an extracellular matrix component known to provide integrity of muscle both in development and postnatal life[52]. In zebrafish, laminin is reported to accumulate in myosepta, which form the somite boundaries and separate individual muscles[53]. Laminin was detected in AM muscle at 48hpf, when the muscle makes contact with cartilage and individual myocytes start to rearrange their morphology (Supplementary Fig. 10a). Then, at 62hpf, when Meckel's cartilage rapidly translocates together with the connected AM muscle, laminin accumulates in the muscle attachment point on the cartilage (Supplementary Fig. 10b, left). Importantly, this regional-specific pattern of laminin localization was dysregulated both in *sox9*- and *runx2b*-deficient embryos (Supplementary Fig. 10b, middle and right).

In order to determine the composition of the laminin matrix, we examined the gene expression pattern of *laminin α2* (*lama2*), *laminin β 1a* (*lamb1a*), *laminin β 1b* (*lamb1b*), and *laminin g1* (*lamc1*), which are reported to be required for the attachment and integrity of muscle in the somite of zebrafish[54,55]. At 56hpf, we found specific and strong expression of *lamc1* in AM muscle, *lamb1a* in Meckel's and palato-quadrate cartilage, and *lamb1b* both in the muscle and cartilage (Supplementary Fig. 10c). We inhibited *laminin* function using CRISPR/Cas9-mediated mutagenesis and verified editing by Sanger sequencing (Supplementary Information). *Lamc1*-crispant embryos had short body length, as expected in cases of *lamc1* loss of function (Supplementary Fig. 10d). Myofibrils of *laminin* crispant embryos appeared misaligned (although generally oriented towards incorrectly positioned cartilage) (Supplementary Fig. 10e). Live-imaging analysis shows individual myoblasts of the *lamc1*-mutant embryo were able to contact the cartilage but subsequently failed to elongate (Supplementary Movie 16 and Supplementary Fig. 10f). Additional loss of function assays on extracellular matrices highly expressed in cartilage or tendon either affected the organization and orientation of facial muscles (such as type II collagen, aggrecan, and comp, primary structural proteins of cartilage, see ECM in Supplementary Fig. 10e), or did not cause any obvious abnormality in the pattern of muscle structure (such as *Tsp4b*, a gene with functions at the muscle-tendon junction[53], Supplementary Fig. 2b). Together, these data suggest that many different matrix molecules are involved in establishing and maintaining the physical contact between muscle and cartilage. Although we cannot rule out that some of the laminin perturbation in these experiments may have directly interfered with correct muscle formation, together the results highlight the importance of cartilage extracellular matrices in muscle directionality and patterning.

### Involvement of the JNK pathway in muscle cell orientation

The JNK (c-jun N-terminal kinase) pathway is known to regulate cytoskeleton remodeling and thus cell and tissue morphology[56]. Myotube formation (Fig. 7a) was disrupted when differentiating myoblasts on 10 kPa substrates were treated with SP600125, a potent JNK inhibitor (Fig. 7b).

We also treated in vitro-grown 3D muscle tissues created between 2 pillars with SP600125 and found that muscle cells were misaligned and misshapen compared to control cells (Fig. 7c, d). When we instead blocked tensile force with blebbistatin, muscle cells similarly showed complete loss of polarization (Fig. 7e). Measuring static tension generated between pillars revealed a significant reduction of tensile force in the treated tissues (Fig. 7f). These results suggest that the JNK pathway plays a role in the mechanical force-regulated reorganization of muscle cell orientation and shape in vitro. We then inhibited the JNK activity in zebrafish embryos. Compared to control (Fig. 7g), embryos treated with SP600125 just before muscle polarization showed dysregulated actin fibers and polarization defects at later developmental stages (Fig. 7h). This may be caused by systemic factors rather than cell-intrinsic mechanisms. However, when we observed cellular behavior with live imaging just after the onset of drug treatment, we found that myoblasts failed to polarize in the direction of the differentiating cartilaginous cells as expected during the mesenchymal condensation phase (Fig. 7i and Supplementary Movie 17). Taken together with the results of 3D muscle tissue in vitro, the JNK pathway is required for autonomous muscle-cell polarization.

### Transcriptomic analysis of myofiber assembly in vitro

To further explore the role of the mechanical environment during muscle cell functionalization, we used an in vitro experimental paradigm where C2C12 cells are induced to differentiate while being cultured on soft (0.5 kPa) or hard (100 kPa) substrates. The difference in mechanical environment causes obvious differences in cell morphology: hard-cultured cells become elongated, form fibrils, and polarize, whereas the soft-cultured cells do not (Fig. 8a). Hard-cultured cells display significantly more cell fusion (Fig. 8b) and significantly more

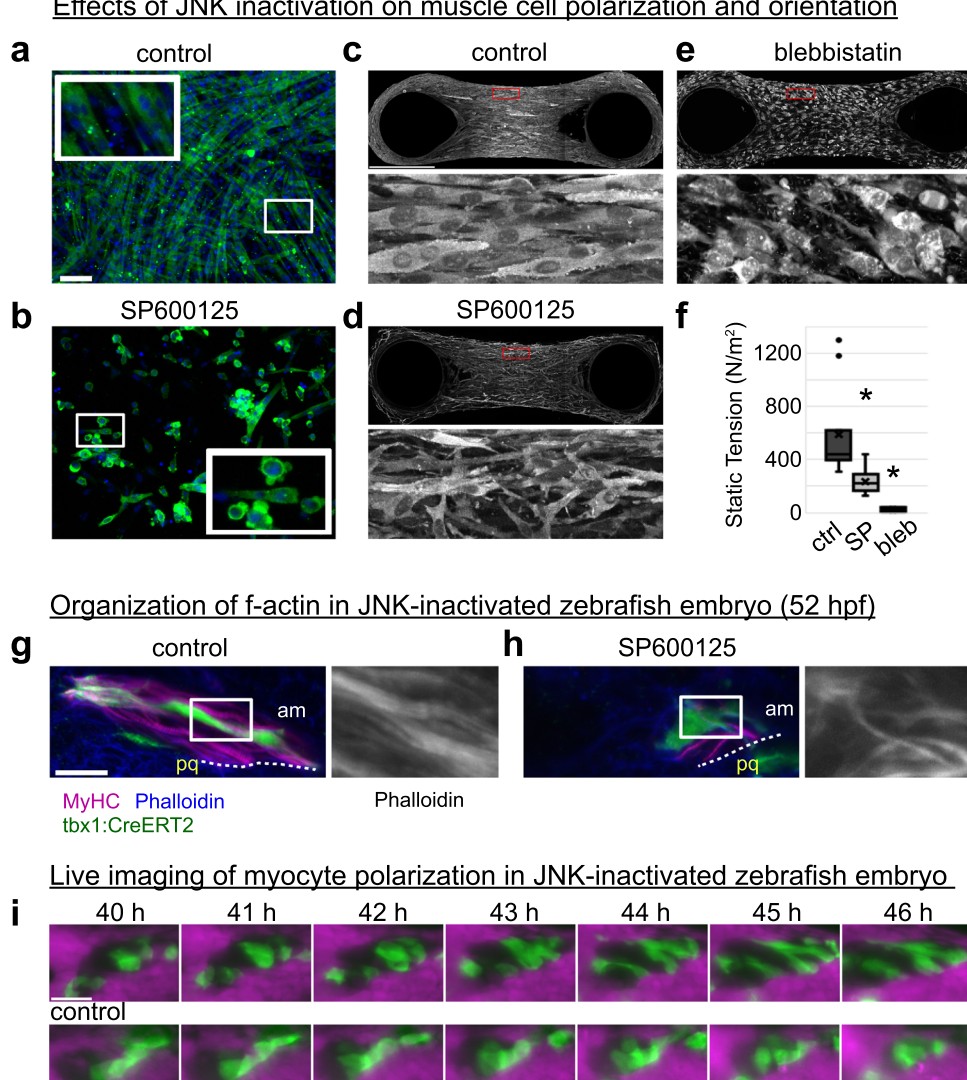

**Fig. 7 | JNK signaling regulates myocyte polarization in vivo and in vitro.**
**a**, **b** C2C12 cells were cultured in 2D and induced to differentiate with ITS, in the presence of either DMSO vehicle control or 20 µM SP600125. Cell morphology was assessed using phalloidin staining, and nuclei were stained with Hoechst ($n = 2$ independent experiments). The white boxes are shown as insets. **c**–**f** C2C12 cells were seeded with the mixture of Collagen-R, Collagen-G, and Geltrex in 3D chamber containing PDMS-coated pillars. After 24 h, cells were induced to differentiate with ITS, in the presence of either DMSO (**c**), 20 µM SP600125 (**d**), or 20 µM blebbistatin (**e**). Tissues were stained with anti-α-actinin antibody. The areas of dashed red boxes are magnified and shown below each overview. **f** The static tension was measured and shown in box plot (defined as median bounded by interquartile range (IQR), whiskers=1.5x IQR). $^*P = 0.0004$, unpaired two-tailed Student's *t* test ($n = 11, 16$, and 7 for control, SP, and bleb, respectively). **g**, **h** Zebrafish embryos were either untreated or treated with 20 µM SP600125 from 39hpf. The actin organization in am muscle at 52hpf is shown. The white boxed area is magnified and shown beside the overview. The boundary between am muscle and pq cartilage is shown with dashed line. $N = 2$ biologically independent control embryos and 4 biologically independent treated embryos. **i** Muscle cell polarization in control and SP600125-treated embryos was examined with time lapse imaging (See also Supplementary Movie 17, $n = 1$ fish per condition). Control; non-treated. Scale bar; (**a**), 100 µm. (**c**), 500 µm. (**g**) and (**i**), 20 µm.

hard-cultured cells are labeled via MyHC immunostaining (Fig. 8c), compared to soft-cultured cells. This inspired us to check if myogenic progenitors require a well-polarized state as a condition to finish muscle differentiation. For this, we performed single-cell transcriptomic analysis of cells collected daily from soft and hard culture conditions (Fig. 8d). The samples of 500 cells collected each day were pooled, and cDNA libraries were prepared and sequenced, resulting in 2000 cells from each condition analyzed via the 10X Genomics platform. Co-embedding and clustering of both conditions revealed nine distinct cell groups, corresponding to major stages in skeletal muscle cell differentiation (Fig. 8e, Supplementary Fig. 11 and 12). The clusters included mesenchymal progenitor cells (C1, light green), *Id1/3*⁺ mononuclear myoprogenitor cells (C4, orange), *Des*⁺ myoblasts (C5,

purple), and *Pdgfra/b*⁺ myofibroblasts (C7, cyan), separate clusters for cycling counterparts of those clusters (C2, C3, and C6, respectively yellow, blue, and grey), *MyoD1*⁺ differentiating myocytes (C8, pink), and *Myh1/3*⁺ mature myocytes (C9, red). *Scx* and *Acta2* were expressed in C9, hinting the cultured myotubes might be immature compared to those observed in vivo, or may retain some properties of smooth muscle and/or general connective tissue (Supplementary Fig. 12a).

Surprisingly, the contributions of hard/soft cultured cells to the mature cell subpopulations C8 and C9 were largely similar (Fig. 8f left). Cells from both conditions evenly dispersed among many clusters in the embedding, with a few exceptions (Fig. 8f middle). Cycling myofibroblasts (C6) were dominated by hard-cultured, whereas mononuclear cells (C4), which are also distinguished by cell adhesion and

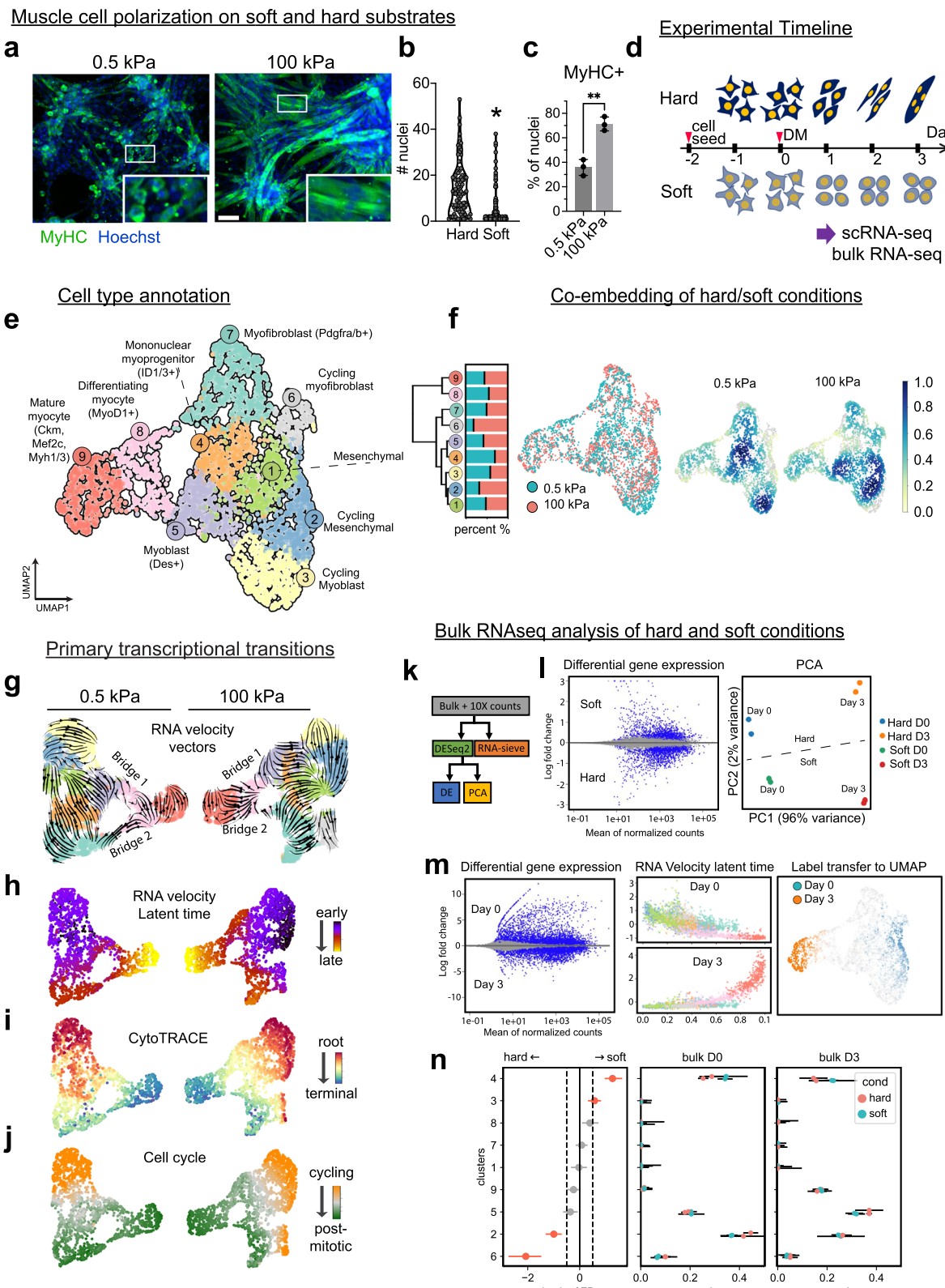

**Muscle cell polarization on soft and hard substrates**

**a** 0.5 kPa   100 kPa   **b** # nuclei   **c** MyHC+   **d** Experimental Timeline

**Cell type annotation**

**Co-embedding of hard/soft conditions**

**Primary transcriptional transitions**

**Bulk RNAseq analysis of hard and soft conditions**

**g** 0.5 kPa   100 kPa — RNA velocity vectors

**h** RNA velocity Latent time

**i** CytoTRACE

**j** Cell cycle

**k** Bulk + 10X counts → DESeq2 / RNA-sieve → DE / PCA

**l** Differential gene expression — PCA

**m** Differential gene expression — RNA Velocity latent time — Label transfer to UMAP

**n** hard ← → soft — bulk D0 — bulk D3

extracellular matrix genes (including *Ecm1*, *Timp3*, *Itgb3*), were enriched with soft-cultured cells (Fig. 8f). These and other local compositional fluctuations can be observed by the sample enrichment analysis (Fig. 8f, right), and all cell markers can be found in Supplementary Fig. 9 and 10. This hints that rather than having a unique transcriptomic signature, myotube-forming myocytes might instead be distinguished from non-myotube forming myocytes at a post-

transcriptional level, possibly via cytoskeletal organization or other localized protein activities.

Next we assessed the main transcriptional trajectories by combining the results from RNA velocity[57], an unbiased approach that leverages the distinction of spliced and un-spliced RNA transcripts from the aligned sequences, with CytoTRACE[58], which uses the number of expressed genes per cell as a measure for differentiation state

**Fig. 8 | Single-cell transcriptomics analysis reveals myocyte differentiation trajectories on hard versus soft substrates. a** Fluorescence microscopy visualization of C2C12 cells cultured under hard and soft conditions for three days, and stained for MyHC. Inset fields of view represent differences in cell morphology and polarization. **b** Quantification of nuclei per myotube or fused cell cluster. Myotubes were analyzed from 3 samples per condition, with a total of 94 myotubes analyzed from the hard condition and 119 myotubes analyzed from the soft condition. Significance was determined by two-tailed Student's $t$-test ($^*p = 0.02$). **c** Quantification of MyHC positive nuclei in each sample ($^{**}p = 0.002$, statistical significance determined by two-tailed Student's $t$-test, $n = 3$ experiments). **d** Schematic representation of the experimental timeline. DM=differentiation media. **e** UMAP-based embedding of the single-cell transcriptomes from hard and soft conditions, pooled from all collection time points. Each color represents a cluster of cells comprising a unique transcriptional identity. **f** Hierarchical clustering of all identified cell subpopulations based on the cluster-specific expression of over-dispersed gene markers, quantifying representation of hard/soft cells in each cluster (left). Co-embedding of both hard and soft datasets (middle), with heatmaps (right) showing the combined dataset with regional levels of enrichment of cells from hard versus soft conditions. **g** Analysis of transcriptional trajectories in the separate hard/soft embeddings. Arrows are representative of the predicted transcriptional flow based on RNA velocity analysis of spliced versus un-spliced transcripts. Predicted

transcriptional transitions are labeled as Bridge 1 and Bridge 2. **h** Heat map showing analysis of latent splice time on soft (left) versus hard (right) conditions, based on splicing kinetics derived from the *scvelo* model. Dot color represents each cell's level of progression through the transcriptional transitions defined by RNA velocity, starting at 0 and ending at 1. **i** Heat map showing analysis of transcriptional diversity of each cell in the soft (left) versus hard (right) datasets. Hot colors indicate more mRNA species observed per cell, while cooler colors indicate fewer unique mRNA species. **j** Bottom, color-coded representation of cell cycle distribution in the soft (left) and hard (right) datasets based on expression of cell cycle-related genes. **k** Bioinformatics pipeline for comparison of single-cell and bulk RNA sequencing data. **l** MA plots showing differentially expressed genes in hard versus soft cultured cells. Statistical significance was determined by a Wald test (via DESeq2 package). Right, PCA (principal component analysis) of variability across samples. **m** MA plots showing differentially expressed genes in cells at day 0 versus day 3 of the culture period (left). Mapping day-associated genes with cell clusters (middle). The strength of the day 0 versus day 3 gene expression signature in the single-cell transcriptomics dataset (right). **n** Results of RNA-sieve ($n = 2$ biological replicates from independent experiments, bars represent 95% confidence interval) showing the representation of cluster-specific gene modules in hard versus soft samples (left), and showing the proportion of cluster-specific gene modules present in early versus late samples (right). Scale bar; (**a**), 100 μm.

(Fig. 8g–i). As expected, gene expression vector maps flowed from regions of less differentiated (C1, C2, C3) to more differentiated (C6-9) subpopulations, with myocytes and myofibroblasts representing somewhat terminal states. As cells activate their differentiation program, intermediate populations (C4-5) linking progenitor cells with either mature myocytes or myofibroblasts became apparent. Latent time, inferred from the dynamical model of *scvelo* (Fig. 8h), the CytoTRACE score (Fig. 8i) as well as the pagoda2 biological aspect related to cell division (Fig. 8j) all indicate a common directionality from cycling progenitors to either post-mitotic differentiated myocytes or myofibroblasts[59]. The distribution of cell cycle was similar across both hard and soft conditions (Fig. 8j), except for C6 which was mostly composed of hard-cultured cells (Fig. 8f) undergoing cell division (Fig. 8j).

Mature myocytes (C9) were linked to the rest of the UMAP by two thin bridge-like cell populations that, according to latent time, might represent differentiation trajectories (Fig. 8h, i). The UMAP embedding presents a population connecting *Des*⁺ myoblasts (C5) to myocytes (C9), labeled as bridge 1 (Fig. 8g). Also, a link between these two ostensibly terminal fates (what we labeled bridge 2, between C7 and C9), hints at a possible fate convergence by distinct transcriptional mechanisms. However, given that RNA velocity vectors point towards both attractors from a trajectory midpoint, these differentiation routes may be slow and inefficient. A more precise look at the phase portrait (spliced versus un-spliced, Supplementary Fig. 13) of the high-velocity genes on bridge 1 (the C5 to C9 link), show clear connections to the process of muscle formation (*Myl1, Srl, Mymk, Ryr1*). Bridge 2 (the C7 to C9 link) is formed by considerably fewer cells and lacks such well-defined phase portraits (Supplementary Fig. 14). It nevertheless suggests a possibility of a transition between myocytes and myofibroblasts. In the soft condition, we detected a strong bias away from myocyte towards a myofibroblast state, suggesting that mature myocyte might not be the default terminal differentiation state in soft culture conditions, which is consistent with previous experimental results. Interestingly, genes driving the velocity and differentiation of intermediate populations belonging to two alternative differentiation routes appeared distinct in hard versus soft conditions (Supplementary Fig. 13 and 14).

Because hard-cultured cells form myotubes that may hinder creating single-cell suspensions for transcriptomics, we also sequenced bulk RNA from hard-cultured versus soft-cultured cells after zero or three days of induced differentiation and compared it to our single-cell transcriptomics data. To this end we used two different

approaches: DESeq, for understanding the major causes of variability across the four types of samples (D0/D3 x Hard/Soft), and RNA-sieve, a tool for finding representation of single-cell clusters within bulk RNA datasets[60]. DESeq analysis revealed that many genes differed when comparing hard versus soft conditions (Fig. 8l, Supplementary Source Data), including 1267 and 1516 genes upregulated in soft and hard in bulk, respectively. Of these bulk differentially expressed genes, 548 and 475 of these genes were also differentially regulated specifically in clusters overrepresented by soft and hard cultured cells, respectively. However, despite these measurable differences in gene expression, hard/soft culture conditions accounted for less than 2% of variability across samples, whereas 96% of variability could be explained by time in culture. In other words, samples at day 3 were transcriptionally very similar, regardless of whether they were cultured on hard versus soft substrates (despite radically different cell morphology). We analyzed genes correlated with time in culture (Fig. 8m, Supplementary Source Data) of which the significantly differentially expressed genes could be used to create aggregate gene modules associated with initial or final timepoints. When cells from the single-cell dataset were plotted, comparing RNA velocity latent time against the activity of time-associated gene modules, we could confirm the general prediction that over time, myocytes were in fact differentiating irrespective of culture condition (Fig. 8m). Further, RNA-sieve deconvolution of hard/soft cluster representation in bulk RNA basically matched that of our single-cell cluster composition analysis and showed similarly increased myocyte (C9) associated gene expression over time in both hard/soft samples, arguing against artefactual dropouts of differentiated cells in our single-cell datasets (Fig. 8n). This result agrees with other recent transcriptomics studies of muscle cells[61–63].

Finally, we used GO categories to define further local variations driven by differences in enrichment between hard and soft populations, which uncovered differential regulation of cytoskeleton behavior via expression of genes involved in Rac signaling (*Rac1*), Rho signaling (*Rock1, Rock2*), cell migration (*Itgb1, Itgb3, Limk1*), and epithelial-mesenchymal transition (for examples see Supplementary Fig. 15). Overall, the observed changes induced by culturing cells on soft versus hard matrix point to genes that may be involved in mechano-sensation as well as transduction of mechanosensory inputs, including JNK-dependent mechanisms. These predictions are in agreement with our observations of aligned actin fibers during polarization (Fig. 2g). However, we did not find support for the hypothesis that polarization in myogenic progenitors would be required for myocyte differentiation. Thus, hard versus soft environments may be

sufficient to drive polarization without greatly affecting gene expression. Although it is common sense that skeletal tissue is stiffer than muscle when fully differentiated, whether such differences in material properties emerge prior to muscle polarization is not obvious. We compared the stiffness of tissue sections containing regions of condensing chondroblasts and myoblasts in mouse craniofacial tissue using atomic force microscopy (AFM). From this experiment, we found pre-cartilage regions to be stiffer than pre-muscle (Supplementary Fig. 16). In support of our transcriptomics analyses, expression of *MyoD* was found with similar intensity in the myoblasts near cartilage condensations, versus the myoblasts far away from pre-skeletal regions (Supplementary Fig. 16).

Taken together, our results reveal how facial muscles become properly oriented in the body. We demonstrated the importance of coordination between facial muscle and cartilage development, as localization of cartilage elements will define the primary polarization of individual myocytes and forming myofibrils. In the developing embryo, oriented cartilage growth imposes force on the local mechanical environment, stimulating co-directional outgrowth of other soft tissues. Specifically, mechanical forces produced at the musculoskeletal attachment point guides the orientation and stretching of maturing myocytes. Moreover, contact between myocytes and facial cartilage further promotes fusion events, assembly of the sarcomere structure, and muscle growth.

## Discussion

The classical role of embryonic cartilage has been mostly studied in the context of skeletal growth and bone formation during development[64]. Previously, we discovered principles explaining the oriented growth and shaping of cartilaginous structures[4,26]. However, tissue interactions involving cartilage might also explain how some body parts expand during morphogenesis. Here, we report that embryonic cartilage is essential for promoting the polarized growth of nearby muscle and soft tissue. These interactions between cartilage and early muscles include mechanical forces that are essential for directional outgrowth of extending myocytes. Overall, in our animal models with genetically controlled cartilage ablations, limbs and jaws are shorter, and nonskeletal connective and muscle tissues develop with proportions determined by the skeleton. This phenotype is reminiscent of the clinical picture of dwarfism resulting from the premature closure of the cartilaginous growth plate, where the limbs and muscles of the affected person are disproportionally short[28,65].

Our results offer a resolution to a long-standing puzzle of how forming muscles acquire specific directionality and length. During embryonic development, muscle myofibrils attach to the cartilage before the bone forms[66], which suggests the initiation of a functional interaction between cartilage and muscles at some point. However, it was not previously known whether the cartilage directs the development of muscle orientation. Consequently, the important biological question about how primary polarization of myocytes and final direction of myofibrils are established during development needed to be addressed. To answer this, we used live imaging and functional experiments in a zebrafish model system. We wanted to test the dependency of muscle orientation and other parameters on the position of progressively differentiating cartilaginous elements within the fish head. SOX9 and RUNX2 are key transcription factors directing cartilage differentiation[30,31,67], and corresponding loss-of-function experiments led to severe cartilage fragmentation or disappearance. As a result, we observed misoriented and disorganized myofibrils in cranial muscles. Experimentally induced changes of the position of cartilage fragments in the head caused reorientation of the corresponding muscles. This led us to the conclusion that muscle size and directionality depend on the pattern of cartilaginous elements that myocytes can attach to. It remains an interesting question how muscles adaptively sense new attachment points after laser-induced

cartilage destruction, as it might utilize mechanisms distinct from stereotypical attachment. Also noteworthy is that cartilage-dependent muscle polarization appears to be unique to craniofacial structures, as trunk myosepta develop without attaching to skeletal elements (Supplementary Movie 18). However, a basic principle of gradual displacement of attachment points, tension, and resulting cytoskeletal anisotropy driving cell shape changes is expected to apply in trunk muscle development as well. Whether other muscles (such as trunk myosepta) utilize such principles to set myocyte polarity will require further investigation.

How do myocytes develop polarity in relation to the geometric distribution of cartilage attachment points? This might be achieved, for instance, via soluble or matrix-embedded molecular guiding cues or, as we demonstrate here, via the influence of mechanical forces. To investigate this question in a system where influences from extrinsic molecular signals can be ruled out, we analyzed the development of oriented multinucleated myofibrils derived from C2C12 cells in 2D and 3D in vitro environments. Unexpectedly, a mild continuous stretch of up to 20% over three days (mimicking expanding cartilage) turned out to be sufficient to induce strong polarization of C2C12 myocytes in the direction of stretch. Such continuous stretch also enhanced the formation of multinucleated myofibrils. When we introduced artificial attachment points into a 3D-environment seeded with myogenic cells, myogenic tissue self-organized into oriented myofibrils connecting attachment points and generated static tension. When we blocked the formation of unidirectional static forces during tissue formation by providing only one anchoring point, the muscle did not develop directionality. This result complies with previous studies, which established that primary myocytes and myogenic cell lines respond to stretch with proliferation and differentiation[68–70]. Notably, pulsating mechanical stress could reorient cultured myoblasts perpendicular to its direction[71]. Although primary polarization stimulates myogenic differentiation and formation of myofibrils according to our experimental results and published data, the single-cell transcriptomics of differentiating C2C12 cells showed that a progression towards a transcriptionally mature myocyte can proceed in an unpolarized condition as well. However, transcriptional maturation does not necessarily reflect the development of a proper myofibril phenotype.

The in vitro-generated results here concur with data obtained from live imaging of developing zebrafish muscles. These experiments revealed that non-polarized myocytes firstly adhere to a nucleated cartilaginous element, where they form a non-oriented multicellular cluster consisting of individual non-fused cells. Subsequently, the myocyte cluster extends cell processes to find another attachment point at another nearby cartilage, and then stretches in a specific direction following the vector of expanding distance between neighboring cartilages. When one potential attachment point is ablated with a laser, the processes of myocytes will navigate to an alternative attachment site, changing the overall orientation accordingly. This occurs concomitantly with the formation of oriented stress fibers within myocytes. During this extension, the myocytes experience elastic stretch, as we showed by laser-mediated ablation of one of the attachment points.

Previously, it was shown that Wnt11, operating via the PCP pathway, acts as a polarizing factor in myocytes during early chick myogenesis[10]. In our case, C2C12 cells placed in a gradient with WNT5A and WNT11 proteins did not develop any specific orientation. Similarly, the knockout of *wnt5a* and *wnt11* in vivo did not lead to misoriented and misaligned myofibrils in fish cranial muscles. The overexpression of *dvl* in fish embryos showed that, although orientation of some fibers were perturbed, overall, the myofibrils still efficiently connect to the cartilaginous elements in an oriented way. The inconsistency with the existing literature might be explained by the fact that multiple cues can operate during myocyte orientation. Furthermore, different mechanisms can dominate at various times and

locations, especially at later timepoints when muscles align with the skeletal elements.

After the initial polarization stage, we found that stretched myocytes acquire a spindle-like shape with the central bulge containing a nucleus that moves constantly back and forth. This is similar to the interkinetic nuclear migrations (INM) in radial glial cells of the developing central nervous system[37]. In fact, INM was previously described in a number of epithelial cell types in different model organisms (reviewed by ref. 72). Current suggestions include the role of INM in tissue compaction and phases of a cell cycle[37,72]. The cells with INM have a long, stretched shape, and the migration of nuclei can assist the distribution of processed RNA, which can be beneficial for integrating signaling pathways with gene expression activity. Support for this idea is provided by the process of nuclei distribution within multinucleated myofibrils[73], which seems to be necessary for the proper metabolic support of every region of the myofibril. After polarization and onset of INM, the single nuclei myocytes stretch and extend while being connected to the attachment points before they start oriented fusions. According to our functional experiments, the initial polarization events occur before and independently of the fusion events and are correlated with the directional expansion of cartilage.

Stretch could affect cell signaling through mechanosensitive ion channels[47]. Such ion channels, for instance, PIEZO1/2 or TRP channels can respond to mechanical stimulation with induction of $Ca^{2+}$ currents and other downstream signals[15,74]. A universal $Ca^{2+}$ channel blocker, $Gd^{3+}$ (Gadolinium) is often used to test the function of different mechanosensitive ion channels[46,48]. Application of $Gd^{3+}$ to developing fish larvae or to C2C12 cells grown on stretchable membrane did not affect muscle orientation per se, but rather influenced the number of nuclei within the resulting myofibrils. This suggests that $Gd^{3+}$-sensitive ion channels participate in myocyte fusion events, but not polarization.

Cytoskeletal rearrangements can also serve as a stretch-sensing system[16,17,75,76], where the formation of oriented actin fibers corresponds to cell polarization[77]. Andersen and coauthors showed that myogenic cells can perceive and interpret mechanical forces via integrin-mediated mechanotransduction, where focal adhesion kinase (FAK) plays a key mediating role[71]. This is consistent with the fact that myocytes adhere to cartilages in a non-polarized manner and soon develop oriented actin fibers following the direction of cartilage growth. Experimental blockade of cartilage development correlated with the loss of actin fiber orientation in myocytes and resulted in disorganized muscle. Consistently, the use of blebbistatin to inhibit myosin II-driven actin fiber contraction[75,78] resulted in underdeveloped and insufficiently polarized myofibrils. Consistently, C2C12 cells differentiating towards myocytes when grown on hard and soft matrices (that allow or do not allow efficient polarization), showed some differences in cytoskeletal genes and cytoskeleton-associated signal transduction genes (Supplementary Fig. 15) according to single-cell transcriptomics analysis. This coincided with our result showing that the JNK pathway is a key for the development of correct myocyte orientation during zebrafish development, as JNK signaling is known to be activated by mechanotransduction via cytoskeleton[16,79]. Following this logic, blockade of specific components of the extracellular matrix components should result in failure of force transmission via the cytoskeleton, giving rise to unpolarized myocytes and myofibrils. Indeed, the knockouts of specific laminin subunits resulted in unpolarized muscles (Supplementary Fig. 10e). In the absence of particular laminins, the myocytes were unable to polarize correctly in vivo and in vitro. Live imaging did not demonstrate any abnormality in initial adherence and formation of myocyte clusters corresponding to relevant muscle groups, but revealed that individual myocytes eventually failed to polarize in the specific direction (Supplementary Fig. 10f).

Taken together, directional expansion of cartilage induces concomitant oriented expansion of nearby forming muscles and soft connective tissues. Mechanical stretch, produced at attachment points drifting apart due to cartilage growth, is sufficient to polarize adhering myocytes, which is supported by 2D and 3D in vitro experiments with stretch and artificial attachment points. Overall, this mechanism represents a previously unanticipated role of mechanical forces in determining the polarity of myocytes and establishing the morphology of the craniofacial musculoskeletal apparatus.

## Methods

### Zebrafish maintenance and embryonic culture

All experimental work with animals was permitted by the Ethical Committee on Animal Experiments (Stockholm North committee) and conducted according to The Swedish Animal Agency's Provisions and Guidelines for Animal Experimentation recommendations. All zebrafish lines were husbanded at Karolinska Institutet Zebrafish Core Facility. Embryos were raised in E3 medium (5 mM NaCl, 0.17 mM KCl, 0.33 mM $CaCl_2$, 0.33 mM $MgSO_4$, and 2 mM HEPES) at 28.5 °C. To prevent pigmentation, embryos 24hpf were transferred to E3 medium containing 0.003% 1-phenyl-2-thiourea (PTU) (P2629, Sigma). For the activation of CreERT2, embryos were treated with 10 µM 4-hydroxytamoxifen (4-OHT; Sigma, H7904) from 8 to 48hpf, with a refreshment of the drug at 24hpf. Blebbistatin (B0560), SP600125 (S5576), BTS (203895), BDM (B0753), and Gadolinium(III) chloride hexahydrate (203289) were purchased from Sigma.

### Transgenic zebrafish lines

Tg(*col2:mCherry*), Tg(*fli:GFP*), and Tg(*actb2:loxp:sstop:loxp-DsRed*) are described previously (Lawson and Weinstein, 2002; Mitchell et al., 2013: Bertrand et al., 2010). Tg(*tbx1:Cre*) and Tg(*tbx1:CreERT2*) were generated with the *Tol2* transposon/transposase system[80,81] as follows: the region between −5717 bp and −7 bp from the start codon of *tbx1* gene was cloned to p5E vector with Gateway BP Clonase II (Thermo Fisher Scientific). Also, Cre recombinase of pCS2-Cre.zf1 (Addgene plasmid #61391) was transferred to pME vector by BP reaction. The expression construct was generated by recombining p5E-tbx1, pME-Cre (or CreERT2, Addgene plasmid #27321), p3E-polyA (tol2 kit #302) and pDestTol2CG2 (tol2 kit #395) with Gateway LR Clonase II (Thermo Fisher Scientific). Similarly, Tg(*ubi:3905NLS*) was generated by recombining p5E-ubi, pME-3905NLS, p3E-poly and pDestTol2pA2. pME-3905NLS was created by transferring the insert of pT2A-3905NLS[82] to the pME vector. Transposase RNA was prepared using the pCS2FA-transposase as a template as described previously[81]. Briefly, in vitro transcription was performed using mMessage mMachine SP6 kit (Thermo Fisher Scientific), and RNA was purified with RNeasy Mini Kit (Qiagen). 24 ng expression construct was injected with 20 ng Transposase RNA to AB line, and embryos were sorted according to fluorescence. Then obtained founder fishes were outcrossed with AB line. For transient expression of Cre-inducible GFP cassette, actb2:loxp-mTagBFPcaax-loxp-EGFP was created as follows: the 10,175 bp promoter of *Actb2* (Bertrand et al., 2010) was cloned to the p5E vector by BP reaction. Then loxp-mTagBFPcaax-loxp sequence was inserted after *Actb2* promoter from pENTR5'_ubi:loxP-EGFP-loxP (Addgene plasmid #27322) and pME-mTagBFPcaax (Addgene plasmid #75149) with restriction enzyme reactions. The expression vector was then created by the recombination of p5E-actb2:loxp-mTagBFPcaax-loxp, pME-EGFP (tol2 kit #383), p3E-polyA, and pDestTol2CG2. For the generation of myog:DVLΔDEP-P2A-EGFP, the promoter region of zebrafish *myogenin* (−956 bp – 0 bp) was cloned to p5E vector, and human *DVL3* lacking DEP region was cloned to pME-p2A-EGFP vector, and then they were recombined to pDestTol2CG2. For the generation of unc503:Gal4VP16, p5E_unc503 (Addgene plasmid # 64020), pME-Gal4VP16 (tol2 kit, #387), and p3E-polyA were recombined to pDEST-tol2pACrymCherry (Addgene plasmid # 64023). Primer sequences used for the promoter cloning are shown in Supplementary Data 1. Stable mutant lines for gene knockout, Sox9a[tw37](item #348),

Runx2b[sal4504] (item#14978), Sox9b[sa40193] (item#36119), and Itgba1[sa8902] (item#37693) were purchased from the European Zebrafish Resource Center.

## Whole mount immunostaining

Whole mount immunostaining was performed according to the protocol described previously[83]). In brief, embryos were fixed by 4% paraformaldehyde at 4 °C overnight and dehydrated in methanol at −20 °C for 24 h. After rehydration, embryos were digested with 30 μg/ml (for 81hpf embryos) or 15 μg/ml (for 62 hpf embryos) proteinase K for 40 minutes at room temperature, incubated with 1 M glycine (pH 8.0) for 30 min, and post-fixed by 4% paraformaldehyde for 20 min. After blocking with blocking solution (0.5% Triton-X PBS, 10% donkey serum, and 1% DMSO), embryos were incubated with primary antibody in blocking solution for 2 days, and then secondary antibody and Hoechst (H1399, Thermo Fisher Scientific) for 1 day at 4 °C. Antibodies used in this study are as follows: anti-MyHC antibody (1:40, A4.1025, DSHB), anti-MyHC antibody (1:200, F59, DSHB), anti-DsRed antibody (1:200, 632496, Clontech), anti-Laminin antibody (1:100, L9393, Sigma), anti-GFP antibody (1:200, ab6662, Abcam), Alexa Fluor 647 donkey anti-mouse IgG (H + L) (1:500, A31571, Thermo Fisher Scientific), Alexa Fluor 555 donkey anti-rabbit IgG (H + L) (1:500, A31572, Thermo Fisher Scientific). Phalloidin staining was performed as described previously[84]. Alexa Fluor 488 was purchased from Thermo Fisher Scientific.

## Whole mount fluorescent in situ hybridization in zebrafish

The procedure was performed according to the protocol described in detail by Talbot, J.C. on ZFIN (http://zfin.org). In brief, embryos were fixed by 4% paraformaldehyde at room temperature for 3 h and dehydrated in methanol at −20 °C for 24 h. After rehydration, embryos were digested with 10 μg/ml proteinase K for 30 min at room temperature. Then, embryos were hybridized with Fluor-, DIG-, and DNP-labeled RNA probes (300 ng each) in 400 μl of Pre-Hyb solution (50% formamide, 5X SSC, 100 μg/ml yeast RNA, 50 μg/ml Heparin, 0.25% tween-20, 0.01 M Citric acid, pH 6.0–6.5) overnight at 68 °C, followed by stringent washes. Embryos were incubated with anti-Fluor-POD (Sigma), anti-DIG-POD (Sigma), or anti-DNP-POD (Perkin Elmer) overnight at 4 °C (1:5000 Fluor, 1:2000 DIG, and 1:500 DNP), and then incubated with FITC-, Cy3-, or Cy5-tyramide (Perkin Elmer). RNA probes were generated using gene-specific sequences cloned into pCR-Blunt (Thermo Fisher Scientific), where SP6 promoter was inserted between MluI and KpnI sites. Linearized templates were subjected to in vitro transcription with DIG-, Fluor-, or DNP-conjugated UTP (Sigma and Perkin Elmer) using T7 or SP6 RNA polymerase (Thermo Fisher Scientific), and then purified with RNeasy Mini Kit (Qiagen). For concurrent in situ hybridization of tenomodulin and col2a1, HCR was used (described below). Primer sequences used for cloning of probe templates are shown in Supplementary Data 1.

## Loss-of-function assays

CRISPR/Cas9-mediated gene-silencing was performed as described previously[85]. with minor modifications. Target sequences were determined using web tools Chopchop (CHOPCHOP v2: a web tool for the next generation of CRISPR genome engineering) and ZiFiT (an updated zinc finger engineering tool). The Template of sgRNA was generated with the PCR reaction (PrimeSTAR GXL Polymerase, Clontech) using scaffold oligo (5′-GATCCGCACCGACTCGGTGCCACTTTTTCAAGTTG ATAACGGACTAGCCTTATTTTAACTTGCTATTTCTAGCTCTAAAAC-3′) and gene-specific oligo (5′-AATTAATACGACTCACTATA (N20) GTTT TAGAGCTAGAAATAGC-3′, where N20 refers to the 20 nucleotides of the sgRNA that bind to the target region of the genome). In vitro transcription was performed with T7 RNA polymerase (Thermo Fisher Scientific) using the PCR product as a template, and obtained sgRNA was purified with phenol-chloroform extraction. Total 200 pg or 400 pg (in the case that target genes are more than two) of sgRNA was injected with 660 pg of Cas9 protein (PNA BIO INC), 10% phenol red, and 0.13 M KCl. Target sequences for each gene are shown in Supplementary Data 1. This method depends on gRNA-targeted double-strand DNA breaks and subsequent non-homologous end joining, which often introduces insertions or deletions that can disrupt gene functions. Using Sanger sequencing and HCR of crispant embryos, we found that all of the gRNAs used for this analysis are effective (Supplementary Information and Supplementary Fig. 17); all the sequenced embryos were shown to have heavy mutations on the target region, where we could not find a stretch of the wild type sequence. Morpholino antisense oligonucleotide-based knockdown for runx2b and jamb was performed as described previously[31,42]. Target sequences for each gene are shown in Supplementary Data 1. 150 μM of runx2b MO or 400 μM of jamb MO were injected with 15% phenol red and 0.2 M KCl. In both cases of the CRISPR/Cas9 and MO method, 100 embryos in total were injected at the one-cell stage for each gene target on at least three different experimental occasions. Embryos showed consistent morphological phenotypes in each biological group. Embryos were randomly selected and imaged, and representative images were shown in the figures.

## Image capture with confocal microscopy and data analyses

Stained zebrafish embryos were cleared by incrementally increasing glycerol in PBS (25%, 50%, 75%, 95%), and observed in 95% glycerol in PBS or 50% glycerol in PBS mixed with 1% agarose gel. Images were taken with LSM710, LSM800, or LSM880 (Zeiss, Zen Blue software), or FV3000 (Olympus) with ×20, ×25 or ×40 objective. Data were analyzed with Imaris (v.8.2.1 or 9.2.1). For the visualization of nuclei in muscle cells and xirp2a + /smyhc2+ positive regions, the colocalization function in Imaris was used. All the data analyses including counting nuclei and measuring length, distance or angle between objects, were performed by going through the optical sections of 3D images in Imaris.

## Live imaging of zebrafish embryos

Live imaging of zebrafish was performed using a Zeiss Z.1 light-sheet microscope. Embryos were anesthetized with tricaine (MS222, 160 μg/ml), and then embedded in 1% agarose gel (Sigma, A9414) with fluorescent microspheres (1:2000 Merck, FZ-050). During experiments, embryos were kept at 28.5 °C. Embryos were illuminated from both sides, and images were acquired with the ×20 objective (W-Plan-APOCHROMAT-1.0NA) with 3, 5, 15, or 20 min interval according to the specific experiment (see figure legend). In Supplementary Movie 2, sample illumination and image acquisition were performed from the lateral position and 45-degree oblique by rotating the sample. Images were fused and deconvolved using the Multiview Reconstruction Fiji plugin (https://www.nature.com/articles/nmeth0610-418 and https://www.nature.com/articles/nmeth.2929). Data from single view experiments were converted to.ims files and analysed with Imaris (v.9.2.1.). According to the experiment, 3D-rendered images or optical sections were exported from Imaris as tiff files, and movies were produced with Fiji.

## Laser ablation in zebrafish embryos

Laser ablation was performed with a Zeiss two-photon microscope (LSM510NLO Meta, Zeiss) equipped with Zeiss Zen Black software. Embryos were anesthetized with tricaine (MS222, 160 μg/ml), and then embedded in 1% agarose gel. In Fig. 4e, the nascent cartilage of 660 μm² was ablated. In Figs. 4d, 14.5 μm x 2.5 μm of the region between cartilage and muscle was ablated. In both cases, the ablation was performed by a short pulse of photon whose wavelength was 800 nm with ×20 objective (W-Plan-APOCHROMAT-1.0NA) and time-lapse image was obtained subsequently. For a more detailed explanation, a description of the protocol can be found in a previous publication[86].

### In vitro cell stretching experiments in 2D and 3D

C2C12 muscle precursor cells were purchased from ATCC. Cells are maintained in 10% fetal bovine serum (FBS) (A3160401, Thermo Fisher Scientific) and 1% Penicillin-Streptomycin (15140-122, Thermo Fisher Scientific) in Dulbecco's Modified Eagle's Medium-high glucose (DMEM) (D5796, Sigma). To induce cell differentiation, the medium was switched to 3% horse serum (HS) (26050-070, Thermo Fisher Scientific), 1% Penicillin-Streptomycin, and 1 μM insulin (I0516, Sigma) in DMEM. In 2D experiments, the stretch chamber harboring the PDMS membrane and the stretcher were purchased from STREX Inc. (STB-CH-04 and STB-100-04, respectively). On the day of seeding cells, PDMS membranes were coated with Laminin by incubating the membrane with 1 ml PBS containing the stock solution of Laminin (L2020, 1-2 mg/ml in PBS, Sigma) for 4 h at 37 °C. After a brief wash with PBS, cells were seeded at 90,000 cells/membrane in antibiotics-free growth medium, incubated for 7 h, and then transfected with pCAG-GFP in refreshed medium with Lipofectamine 3000 (Thermo Fisher Scientific) to highlight cells. After 16 h, cells were induced to differentiate, and stretched by 20% immediately or 20 h later. Live cell imaging was performed using a Cell Observer (Zeiss). Cells were kept at 5% CO2 and 37 °C during the experiments, and images were taken at four fields of view with a ×10 objective at 15 min intervals. Images were stitched with Zen software, converted to .ims file format, and analyzed with Imaris (v.9.1.2.) and ImageJ. Phalloidin staining was performed following the manufacturer's protocol on the stretch chamber. Images were taken with a LSM710 laser scanning confocal microscope (Zeiss) and analyzed with Imaris (v.9.1.2.) and ImageJ. For 3D experiments, 5000 C2C12 cells per tissue were mixed with an extracellular matrix consisting of 0.2 mg/ml Collagen-R, 0.4 mg/ml Collagen-G (both from Matrix Bioscience), and 1.5 mg/ml Geltrex (Invitrogen, Carlsbad, CA) and seeded around the pillars. After 24 h, differentiation was induced by switching the medium to 3% horse serum and 1 μM insulin. Stretch was induced using a stepper motor-based uniaxial stretcher as described previously for 2D substrates[87]. To simulate the mechanical cues of early muscle development, C2C12 tissues were exposed to increased static stretch up to 20% over the course of 3 days starting 24 h after the original cell seeding. Static tension was quantified by evaluating pillar deflection and tissue width using the image analysis software ClickPoints[88]. Cell orientation was analyzed with ImageJ.

### Preparation of polyacrylamide gels

Polyacrylamide for cell culture was prepared according to previous reports[89]. 12 mm Glass-bottom dishes (150680, Thermo Fisher Scientific) were pretreated with bind-silane (M6514, Sigma):acetic acid:ethanol (1:1:14). The dishes were washed twice with ethanol and air-dried. For 100 kPa (0.5 kPa) gels, a 1000 μl stock solution containing HEPES 10 mM, 300 μl (75 μl) acrylamide 40% (1610140, Bio-Rad), 125 μl (30 μl) bisacrylamide 2% (1610142, Bio-Rad), 5 μl (5 μl) 10% ammonium persulfate diluted in water (A3678, Sigma), 0.5 μl (0.5 μl) N, N, N', N'-tetramethylethylenediamine (TEMED) (T22500, Sigma), 565 μl (880 μl) distilled water was prepared. 120 μl of the solution was added to the center of the glass-bottom dishes and the solution was covered by hydrophobically coated coverslip. After 1 h of polymerization, the coverslip was removed, and 80 μl of Sulfo-SANPAH (22589, Thermo Fisher Scientific) was placed on the top of the polyacrylamide gel. Sulfo-SANPAH was diluted with distilled water to the final concentration of 2 mg/ml from the original stock of 50 mg/ml just before use. The surface was activated by UV light for 5 min. Polyacrylamide gels were washed twice with distilled water and once with PBS for 5 min. The gels were incubated with 0.1 mg/ml fibronectin (33010018) overnight at 4 °C and then washed with PBS. To achieve the same level of cell confluency on the day of the induction of differentiation, C2C12 cells were seeded at $1.3 \times 10^4$ and $3.9 \times 10^4$ cells/cm$^2$ on 100 kPa and 0.5 kPa gels, respectively. Two days later, the medium was switched to differentiation medium.

### Mouse strains and animal information

All mouse work was approved and permitted by the Local Ethical Committee on Animal Experiments (North Stockholm Animal Ethics Committee) and conducted according to The Swedish Animal Agency's Provisions and Guidelines for Animal Experimentation recommendations. *Col2a1-CreERT2* strains[90] were obtained from the laboratory of S. Mackem, NIH. *DTA* strain[91] (*B6.129P2-Gt(ROSA) 26Sortm1(DTA)Lky/J*, The Jackson Laboratory) was coupled to *Col2a1-CreERT2*. Mice were mated overnight, and noon of the day of the plug was considered as E0.5. To induce genetic recombination, pregnant females were injected intraperitoneally with tamoxifen (T5648, Sigma) dissolved in corn oil (C8267, Sigma). Mice were sacrificed with isoflurane (Baxter KDG9623) overdose or cervical dislocation. Embryos were dissected in ice-cold PBS, and subjected to staining for microCT scanning as described below.

### Hybridization chain reaction (HCR) in mouse and zebrafish embryos

Hybridization chain reaction (ver.3) was performed according to the manufacturer's protocol (Molecular Instruments, Inc. https://www.molecularinstruments.com/). Briefly, embryos were fixed by 4% paraformaldehyde at 4 °C for 24 h and dehydrated in methanol at −20 °C for 24 h. For optimization of *Scx* visualization on slides, embryos were embedded in OCT and cryosectioned without fixation, and HCR was performed directly on-slide. After rehydration, samples were prehybridized in 500 μl of 30% probe hybridization buffer for minutes at 37 °C, and then hybridized to 4 pmol of probes (2 pmol of odd and even probe) for each target at 37 °C for 16 h. Probe concentration was doubled for experiments where a high probe concentration was tested. After washing with 30% probe wash buffer at 37 °C and 5X SSCT at room temperature, samples were incubated with 30 pmol of each fluorescently labelled hairpin in 500 μl of amplification buffer at room temperature for 16 h, and then washed with 5X SSCT. Probes and buffers were purchased from Molecular Instruments, Inc. Protein staining was performed after HCR reaction according to the specific experiment. Zebrafish embryos were cleared in 50% glycerol in PBS and imaged with confocal microscopy. *Scx*-hybridized tissue sections were imaged with confocal microscopy without clearing. Tissue clearing of mouse embryos was performed with iDISCO+ (https://idisco.info/), as described[92]. Briefly, samples were dehydrated with graded methanol, incubated with 66 % dichloromethane (DCM) (270997, Sigma) / 33% methanol for 3 h, and then 100% DCM twice for 15 min. Samples were transferred to DiBenzyl Ether (108014, Sigma) and kept until tissues became transparent. After clearing, the images of whole embryos were taken by UltraMicroscope II of LavisionBioTec.

### Light sheet microscopy and 3D reconstruction of mouse embryos with wholemount FISH

A light sheet fluorescence microscope (Ultramicroscope II, Lavision Biotec, Bielefeld, Germany) and the ImspectorTM 347 software were used. The microscope was equipped with an Olympus MVPLAPO ×2/ 0.5 objective lens, an Olympus MVX-10 0.63-6.3x zoom body, a 6.5 mm working distance spherical aberration corrected dipping cap, a sCMOS camera (Andor Neo, pixel size: 2560 × 2160) and Coherent OBIS lasers (488-100 LX, 488 nm; 561-100 LS, 561 nm; 640-100 LX, 640 nm) with appropriate filters. To obtain the required X/Y/Z resolution, homogenous illumination within the entire focal plane and minimal photobleaching, blocks of full E11.5 embryos with MyoD/Col2 whole mount fluorescent in situ hybridization were scanned with the following parameters: multicolor acquisition mode, ×2 objective, 0.8x zoom body and additional magnification of the dipping cap lens (altogether 1.8× effective magnification), 100 ms exposure time, max sheet numerical aperture (0.156), 70% sheet width, 3.0 mm Z-step thickness, no tiling (3.02 mm × 3.02 mm × 3.0 mm voxel size), dynamic horizontal focus (25 steps, contrast adaptive algorithm). The serials of 16 bit

uncompressed tif images were then converted to IMS files using the Imaris File Converter 9.2.1TM program (Bitplane, UK) and the 3D vision of acquisitions was reconstructed in the Imaris 9.2.1TM (Bitplane, UK) software for inspection/quality control and further analysis.

## Tissue contrasting and X-ray computed microtomography measurements

To increase the contrast of soft-tissues for X-ray computed micro-tomography (microCT), the samples were stained by phospho-tungstic acid (PTA). Staining protocol has been adapted from the original protocol developed by Brian Metscher laboratory[93]: After embryo dissection in ice-cold PBS, the samples were fixed in 4% formaldehyde in PBS solution for 24 h at 4 °C with slow rotation. Subsequently, samples were dehydrated in incrementally increasing ethanol concentrations (30%, 50%, 70%), 1 day in each concentration. Samples were transferred into 1.5% PTA in 90% methanol for tissue contrasting. The PTA-methanol solution was changed every 2–3 days. Embryos were stained for 7 weeks. The contrasting procedure was followed by rehydration of the samples by incubation in ethanol series (90%, 70%, 50% and 30%). Then, the rehydrated embryos were embedded in 1% agarose gel (A5304, Sigma-Aldrich) and placed in polypropylene conical tubes to avoid motion artifacts during microCT scanning. The microCT measurements of the embryos were conducted using the system GE phoenix v|tome|x L 240 (GE Sensing and Inspection Technologies GmbH, Germany) with a 180 kV/15 W maximum power nanofocus X-ray tube and flat panel dynamic 41|100: 4000 ×4000 px, with pixel size 100 × 100 μm. The exposure time was 900 ms and 2000 projections were taken over 360°. Three projections in every position were averaged to reduce the noise in the data. The utilized acceleration voltage and current were 60 kV and 200 μA. X-ray spectrum was filtered by 0.2 mm of aluminium plate. The voxel size of the reconstructed data was 6.7 μm and 12 μm, respectively. The tomographic reconstructions were performed using GE phoenix datos|x 2.0 3D computed tomography software (GE Sensing and Inspection Technologies GmbH, Germany). For 3D visualization, the segmentation was done by an operator using a combination of software Avizo (ThermoFisher Scientific, USA) and VG Studio MAX 3.2 software (Volume Graphics GmbH, Germany).

## Bulk RNA sequencing studies

For sequencing experiments, C2C12 cells were seeded on soft (0.5 kPa) and hard (100 kPa) fibronectin-coated polyacrylamide gel, and collected every day until 5 days after seeding. Differentiation was induced 2 days after cell-seeding by changing medium to differentiation medium. For bulk RNA sequencing, samples were collected 0 and 3 days after differentiation. RNeasy micro kit (Qiagen) was used for bulk RNA isolation, and 100 ng starting RNA was used for library preparation. NEBNext Poly(A) mRNA Magnetic Isolation Module (E7490S), NEBNext Ultra II Directional RNA Library Prep Kit for Illumina (E7760), and Multiplex Oligos for Illumina (96 Unique Dual Index Primer Pairs) (E6440) were used for library preparation. The bulk RNA libraries were sequenced for 75 cycles on a NextSeq550 sequencer with reagents from the NextSeq 500/550 High Output Kit v2.5 (20024906). DESeq2 package was used for differential expression analysis of count data. RNA-Sieve package was used for deconvoluting the bulk data using the single-cell transcriptomic dataset as a reference.

## Single-cell transcriptomics and downstream bioinformatic analysis

For single-cell studies, cells were harvested using 0.05% Trypsin/EDTA, fixed in ice-cold methanol, and stored at −80 C prior to single-cell library preparation. A total of 2000 cells per condition were analyzed by single-cell transcriptomics using 10X Genomics Chromium v3 kit, following the manufacturer's instructions, and sequenced on an Illumina HiSeq 2500. Count matrices were generated from the fastq files via CellRanger 3.1.0 using mm10-1.2.0 genome annotation. Ambient and background RNA from the count matrices were first removed using CellBender remove-background tool. Corrected count matrices were cleaned from low-quality cells by removing cells having less than 1000 genes or cells that do not fit the expected detected gene vs molecule count relationship. Batch correction, dimensionality reduction, knn graph (cosine distance, knn=100), clustering (leiden community detection, resolution 1) and pathway overdispersion analysis of the sequencing data from hard and soft conditions was performed using pagoda2 R package[94]. Relevant principal components were selected using an automated elbow curve selection approach, by drawing a line from the first to the last point of the elbow curve. The data point that is the farthest away from that line is considered as the elbow and define the number of PCs to retain. UMAP embedding was subsequently generated from these retained PCs. RNA velocity data was generated using velocyto tool[57], http://velocyto.org/) by counting intronic and exonic reads from output bam files generated by CellRanger. Velocity analysis was performed separately on both datasets using scvelo python package, with a gene filtering step of 20 shared count minimum and retaining only the 2000 most variable genes[59], https://scvelo.org). First-/second-order moments were computed for each cell across 30 of its nearest neighbors, where the neighbor graph is obtained from Euclidean distances in the first 30 principal components. Estimates velocities in a gene-specific manner were generated using the dynamical model of transcriptional dynamics. Velocity graph based on cosine similarities was then computed, and velocity vectors were projected onto the UMAP embedding generated by the pagoda2 pipeline. To visualize cell differentiation trajectories, the CytoTRACE computational framework was used to analyse gene counts and covariant gene expression to predict differentiation states as described in[58], https://cytotrace.stanford.edu/). Cell cycle assignment was performed using standardized scores of mean expression levels of phase marker genes, via integrated function score_genes_cell_cycle from scvelo package.

## Tissue indentation using atomic force microscopy (AFM)

E11.5 mouse embryos were extracted and snap-frozen in O.C.T. compound with liquid nitrogen. Those embryos were then serially cryo-sectioned by a microtome with a thickness of 20 μm. To identify nascent cartilage and muscle regions for AFM measurement, we performed immunofluorescence staining (cartilage: Sox9, muscle: MyoD) and in situ mRNA hybridization (cartilage: Col2a1, muscle: MyoD) for one-before and one-after cryo-sections, respectively. Finally, those staining results for those neighboring cryo-sections were collated with AFM results. For AFM-based tissue indentation experiment, the cryo-section was thawed and the nuclei were immediately stained with Hoechst 33342 in PBS(-). In this study, AFM system (JPK BioAFM NanoWizard 3; Bruker Nano GmbH., IX81; Olympus Co.) was used and AFM cantilevers (TL-CONT; spring constant 0.2 N/m; NANOSENSORS) were modified with glass beads (15 μm-diameter; Polysciences. Inc.). Spring constant of the AFM cantilever was calibrated with a thermal noise method. Piezo displacement speed was set to be 3 μm/s. As a result of tissue indentation, force (F) versus indentation depth (h) curves were obtained and slope (nN/μm) was evaluated for each curve by linear regression. Sample points within the force range of 750 pN ≤ F ≤ 1000 pN were used for the evaluation. After AFM, the sample was immunostained for Sox9 and MyoD.

## Statistics and reproducibility

All micrographs of zebrafish shown are representative of at least $n = 2$ biologically independent samples unless indicated otherwise. Where statistical analysis was performed, details about the test (type of test, sample sizes, definition p-value) are located in the respective figure legends. All values used for statistical analysis can be found in the accompanying integrated datasheet. Box-plot elements are defined as the following: center line, median; box limits, upper and lower quartiles; whiskers, 1.5x interquartile range; points, outliers.

**Reporting summary**

Further information on research design is available in the Nature Portfolio Reporting Summary linked to this article.

## Data availability

The datasets generated during and/or analyzed during the current study are available from the corresponding author on reasonable request. Transcriptomics datasets generated during this study have been deposited to GEO database under accession codes GSE160098 (single cell RNA sequencing) and GSE199093 (bulk RNA sequencing). Source data are provided with this paper.

## Code availability

The code used for analysis of transcriptomics data is available on Github [https://github.com/LouisFaure/muscledirection_paper].

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

## Acknowledgements

We appreciate Dr. Herbert Shea for his idea and discussion about the manuscript. We thank the Single-Cell Genome Information Analysis Core (SignAC) in ASHBi for the Bulk RNA sequencing. I.A. was supported by Bertil Hallsten Research Foundation and Swedish Research Council. M.T., T.Z. and J.K. acknowledge the project CEITEC 2020 (LQ1601) with financial support from the Ministry of Education, Youth and Sports of the Czech Republic under the National Sustainability Programme II and CzechNanoLab Research Infrastructure supported by MEYS CR (LM2018110). M.T. was financially supported by the Brno City Municipality as a Brno Ph.D. Talent Scholarship Holder. A.G.E. was supported by the National Institute of Dental and Craniofacial Research of the National Institutes of Health under award number 1F32DE029662. B.F. and D.K. were supported by DFG grant FA336/12-1. L.F. was supported by the Austrian Science Fund DOC 33-B27. M.K. was supported by the Max Planck Society.

## Author contributions

K.S. and A.G.E. contributed equally to the work. K.F. and I.A. jointly supervised the work. K.S., I.A., A.G.E., B.F., and K.F. conceived and designed the analysis. K.S., A.G.E., D.K., C.A., P.K., S.K., M.K., I.D.E., M.T., T.Z., J.K., and S.E. participated in performing experiments to collect data. K.S., A.G.E., S.E., L.F., M.T., T.Z., T.A., K.M., T.Y., and J.K. provided analysis tools and performed analysis. A.G.E., K.S., and I.A. wrote the paper, and the manuscript was approved by remaining coauthors.

## Funding

## Competing interests

The authors declare no competing interests.
