## [Peer Review File · Nature Communications]

Directionality of developing skeletal muscles is set by mechanical forcesReviewers' Comments:

Reviewer #1:

Remarks to the Author:

The study by Sunadome et al, aims to examine the role of cartilage in regulating the development of striated muscle, specifically examining myocyte orientation and fusion, both of which are tightly regulated processes. Using a combination of model systems including cell culture, zebrafish and mice, they reveal that muscle and cartilage development is synchronized and the mechanical forces generated by growing cartilage dictates the growth of muscle. This is supported by the observation that disrupted cartilage formation, either using genetic approaches or ablation strategies, results in impaired muscle patterning and fusion. The authors further reveal a role of extracellular matrix proteins and the JNK pathway in regulating the orientation of muscle cells. Finally, to identify how the mechanical environment influences muscle development, the authors performed single cell transcriptomics on cells grown on soft and hard substrates. These analyses reveal a role of mechanical forces in influencing the transcriptional state of the muscle fibres, specifically by altering the expression of mechanosensory pathways, although no further functional studies were performed to validate these studies.

Overall, while the question the authors set out to tackle is interesting and has significant implications on ongoing efforts to generate ex vivo grown muscle, the manuscript appears to be several small experiments have been performed independently, and subsequently combined into a single story. This has resulted in a disjointed and non-cohesive piece of work that was extremely difficult to follow, subsequently diluting the overall finds of the paper.

Major comments:

1. While the authors clearly show a role of cartilage development in regulating muscle development, they have not determined the mechanistic basis of this regulation. While it is possible that changes in passive force following growth of cartilage is plays a role in dictating muscle development, the authors have not negated alternative non-autonomous mechanisms. For example, it is possible that deposition of proteins by tenocytes drives muscle development, and without examining such mechanisms it is incorrect to conclude that the "directionality of developing skeletal muscle fibres is set by mechanical forces".
2. The authors have presented single cell RNA seq studies on cell grown in soft and hard matrices, to decipher how mechanical environment influences muscle development. While this data has clearly shown that the mechanical environment can influence the transcriptional state of myocytes, further functional studies are required to validate their findings and support their conclusions in an in vivo setting.
3. To examine how disrupted cartilage development impacts muscle development, the authors heavily rely on data from CRISPR/Cas9 edited larvae. A major criticism of this approach is that not all cells display the same mutation, and that different embryos would show different degrees of mutagenesis. Therefore, a more appropriate approach is to either generate and analyse stable mutants line whereby all cells show the same mutation, or utilize a tissue specific knockout approach, whereby the target gene is mutated specifically in the cartilage. It is also misleading and inaccurate to call these injected embryos mutants.
4. The resolution of some of the data presented is relatively low to draw conclusions. Higher resolution images need to be provided. Relevant datasets include Figure 1c, where the MyHC images are highly saturated making it difficult to clearly see individual muscle fibres and subsequently the misalignment; Figure 3F, specifically the runx1 cells; Figure 4a, specifically the bottom panels which are blurry; Figure 4b.
5. The initial studies in zebrafish reveal that early muscle and cartilage development is synchronized in zebrafish, and that cartilage is required for the development and maintenance of muscle. However, the authors only examined this in the context of facial muscle. Is this also the case in the trunk skeletal muscle, or is the development of these muscle groups independent of cartilage development? It would be beneficial to examine trunk muscle and cartilage patterning at earlier stages when skeletal

muscle is developing, and also perform cartilage loss of function experiments to determine how trunk muscle development and homeostasis is maintained.

Minor comments

1. All figure panels need to be structured better, as currently the individual panels are extremely difficult to decipher. For example, it would be beneficial to add additional sub-labels such as Ai, AII, Aiii to help follow which panels the authors are referring to. Labels also need to be added to detail the different views. For example, what are the 4 different 48 h images in Figure 1b, and which ones are the authors referring to in their discussion?
2. Please use correct terminology for the zebrafish loss of function experiments: Since the authors have analysed gRNA injected embryos, whereby mutagenesis is mosaic, it is incorrect to call them mutants. Please denote gRNA injected embryos as CRISPR/Cas9 edited larvae.
3. Line 119: "By 72 h.p.f., cartilage and facial muscles already establish a mature interconnected pattern (Supplementary Video 1, and Fig. 1a)." The legend of Figure 1a however states that the image is from an 81 hpf larvae. Please correct the discrepancy in stage.
4. Line 131: Suggestion. "Moreover, before cranial cartilaginous elements achieve their shape, newly born individual muscle cells connect to clusters of the coalescing chondrocytes in a mouse embryo (Fig. 2a and 2b, see below)". Discussing mouse data at this point breaks the flow of the zebrafish results. Please omit mouse data results from here, and move to the section titled: "Cartilage development coincides with muscle and controls directional outgrowth in a mouse".
5. Line 151: "although they displayed no or marginal change in the eye size, suggesting a normal progression of general body development". The authors need to include representative images if they would like to make claims on the eye size. Additionally, their analyses clearly shows a significant reduction in the eye size in *runx2b* morphants, saying the statement they have made is misleading. The smaller eyes in the morphants suggest developmental delays and could explain why "functional inactivation of *Runx2b* produced more severe effects".
6. Figure 1C – the graph for the eye diameter is missing a y axis label.
7. Figure 1G – Please provide single channel images (in addition to the merged images).
8. Figure 2H and line 228. Discussion of the fish results is disrupting the flow of the mouse work discussion. A suggestion is to move Figure 2H to Figure 1, where are the remaining fish data is presented.
9. Figure 3A: Please add arrowheads to identify the cells of interest
10. Line 241: "Our live imaging data revealed that the primordium of AM muscle starts to interact with palatoquadrate cartilage as soon as they simultaneously emerge". Since both, cartilage and AM myocytes are already present at first time point examined (44 hours), it cannot be concluded that the interaction begins as soon as the both emerge. Earlier timepoints need to be examined with higher temporal resolution to make this claim.
11. Figure 3bb: Please add details of statistical significance to the changes in polarity in the *sox9* and *runx2* deficient fish.
12. Figure 3e. While in the middle panel it is clear where the ablation is, in the bottom panel the ablation site is not clear. Figure 3F: To examine cell division, please clearly show the myofibril and/or basement membrane.
13. Line 321 and Figure 3a: After cutting, the tip of myocyte moves away from the ablated sites along the axis of the muscle tissue". Rather than an active migration of the myocyte, this appears to be detachment of a cell, due to the loss of attachment site, as seen in muscular dystrophy. Given the former isn't experimentally tested, please amend to present it as a hypothesis rather than a validated finding.
14. Figure 4b: The phalloidin seems to be labelling the sarcomere of muscle fibres, not stress fibres. The disruption of F-Actin staining in the *sox9* and *runx2b* depleted larvae is not surprising given that the authors have already shown that sarcomeric structure is also disrupted in them. These results need to be combined with the Figure 1f, or removed and the relevant sections in the text need to be amended.
15. Additionally, given that phalloidin labels muscle and cartilage, the accumulation on "stress fibers"

at the muscle-cartilage is not clear. Perhaps a heatmap can be used to demonstrate this point 16. The relevance of titin experiments is not clear. Please elaborate why titin loss of function experiments were performed.

Reviewer #2:

Remarks to the Author:

Comments to the author:

“Directionality of developing skeletal muscles is set by mechanical forces”

In this study, the authors investigate how cartilage expansion influences attached myocyte polarization, orientation, and fusion during embryonic development. Using a combination of loss-of-function studies, live and advanced imaging, and C2C12 cell stretching experiments in 2D and 3D culture the authors demonstrate that the directional outgrowth of developing myocytes and myofibrils are regulated by mechanical force. Additionally, the authors attempt to dissect the molecular basis of this phenomenon by functionally targeting members of the WNT and JNK signaling pathway. Using single cell RNA-seq, the authors also examine whether cell polarization is required for muscle cell differentiation by collecting C2C12 cells cultured on hard or soft substrates. Using this approach, the authors analyze the transcriptional dynamics across different stages of skeletal muscle differentiation and across different conditions. Furthermore, gene expression differences between hard and soft cultured cells further support a role for mechanical transduction and signaling in myocyte orientation and polarization.

The manuscript is well-written with the aims, main findings, and methods clearly stated. Moreover, the authors outline the purpose, design, and rationale behind each experiment throughout the results. The findings are strengthened by the use of both zebrafish and mouse animal models in addition to the use of the mouse C2C12 cell line, demonstrating that their findings are reproducible in vivo and in vitro. Overall, this is an interesting manuscript with broad applications, yielding insight into how tissue interactions and mechanical forces mediate development of the musculoskeletal system and will be considerable interest to readers.

The adjoining of muscle to cartilage is mediated by tendon and thus, one would predict tendon too is subject to mechanical tension generated by cartilage expansion. Tendon itself may also produce mechanical tension (see Kalson et al., PNAS 2013) as it forms and grows that is applied to the forming muscle. Others have shown that connective tissue can affect muscle development independent of any defects in skeletal formation (Hasson et al., 2010). While the authors mention tendon in the introduction, it would greatly help the manuscript to have a greater characterization of the tendon cells, particular how the alignment of collagen fibers and/or tendon cells are affected. Early in the manuscript, the authors demonstrate that *runx2b*-deficient embryos have disrupted *xirp2a* expression, but as *xirp2a* does not mark all tendon cells, the authors should examine additional tendon markers such as *scxa*, *col1a2*, or *tnmd* expression in *runx2b* morphants. The authors show *xirp2a* expression pattern is disrupted in *sox9a/b*-deficient embryos and from previous reports, *scxa* expression pattern also is disrupted in *sox9a/b*-deficient embryos, indicating that tendon morphology is equally dependent upon *sox9a/b* and likely cartilage formation. Overall, this raises the question of whether myocyte polarization is directly regulated by cartilage expansion or alternatively, via its effects on tendon development, or perhaps by both tissues. This is not a fundamental issue with the manuscript as the authors demonstrate that mechanical tension is an important regulator of myocyte polarization and orientation directly through their in vitro work. However, I recommend that the authors examine additional markers of tendon in *runx2b* deficient zebrafish and cartilage-ablated mice, and include some discussion of the tendon's mechanical properties and its relation to both cartilage and muscle during development. This would provide greater context to the functional relationship between the different musculoskeletal tissues and their influence on one another.

Here are some minor comments:

- Line 131-132: "...muscle cells connect to clusters of the coalescing..." implies that muscle is directly attached to the cartilage rather than by tendon. This statement also assumes the tissues are attached but evidence of adhesion is not supported without demonstrating the presence of adhesion molecules such as integrins.
- Line 139: tendocytes should be changed to tenocytes
- Line 180- should say "exhibited no xirp2a+ tendon cells" as there still could be scxa+ tendon cells present.
- Line 416-417: Where are *Igtb1a* and *Itgb1b* normally expressed in zebrafish – are they specifically found at the muscle-tendon boundary or tendon-cartilage boundary?
- Figure 7d: there is a typo- "Phase portraits..." rather than "Phase protraits..."
- Figure 7h: Although the colors are representative of different clusters, cluster numbers overlaid as text would be helpful to orient the reader to the co-embedded plots.

Reviewer #3:

Remarks to the Author:

In this study, Sunadome and co-workers have investigated how the directionality of developing skeletal muscles is set by mechanical forces. The authors made use of many different approaches and multiple species in this study. To start with, they showed that muscle and cartilage development is synchronized, and cartilage depletion could lead to a dysregulation of muscle patterning. Using live imaging and laser ablation techniques along with genetic tools, they suggest that developing cartilage influences individual muscle cell polarization at an early stage and the directionality of subsequent fusion events. They also indicate that the non-canonical PCP pathway is not involved in direct control of facial muscle directionality in zebrafish.

The author also described the effects of mechanical stretch on muscle cell orientation in 2D and 3D cultures and concluded that mechanical tension is sufficient for directionality, elongation, and controlled fusion of individual muscle cells. They further demonstrated that the JNK pathway is required for muscle cell polarization. To understand if muscle cells require a well-polarized state as a condition to complete muscle differentiation, single-cell RNA-seq time-course experiments were done on cells that are cultured in a soft or hard mechanical environment. They were unable to conclude that muscle cell polarization is necessary for differentiation.

Overall, this is an elegant study and the authors have used a wide range of techniques to demonstrate that mechanical forces are important for the polarization of muscle cells, and subsequently the directionality of developing skeletal muscle. While the *in vivo* and the *in vitro* data convincingly suggest that the directionality of skeletal muscle requires mechanical forces, the last part of this study regarding the muscle cell polarization on soft and hard substrates is confusing. First, at least in the case of hard substrates, there are many myotubes formed. It is unclear whether the authors have included these cells in their scRNA-seq analysis. Were they sequenced at all, or only mononuclear cells were included? I am also confused with the experimental design and the cell cluster labels. How many time points are collected? Is there a time point before seeding? The authors need to provide more

rationale/explanation for this experiment.

Major points:

1. Fig1g, as Runx2-deficient mutant have distorted myofibrils, is deficient of tenocytes resulting from the loss of cartilages or distorted myofibrils? Do the migration or specification of tenocytes affected, or tenocytes die?
2. Fig4a, can the author provide further explanation on the rationale of using the distance between the tip of the myocyte and the ablated site to indicate a mechanical tension in an individual myocyte?
3. Line354, it is imprecise to claim that the reduced number of nuclei indicate influenced muscle cell fusion. It might also due to reduced migration of myocytes or reduced proliferation.
4. Why the length of myofibrils in Gd3+ treated embryo seems unchanged?
5. Fig7c, why C2 myoblasts have myofibroblast-like signatures? What is this cluster of cells?
6. Fig7, Transcriptome changes may due to the effect of impaired differentiation. A control, i.e., undifferentiated cells cultured on the hard substrate, is needed to strengthen the claim.

Minor points:

1. Fig3a and Sup video 5, the pseudo-color of *fli1*:GFP is purple while *tbx1*:Cre(Dsred) is green, which can be confusing.

Reviewer #4:

Remarks to the Author:

In their paper Directionality of developing skeletal muscles is set by mechanical forces, Sunadome et al provide a very comprehensive and novel study of the factors driving myoblast polarization and myofibers/myofibrils directionality. The authors used several state of the art techniques (live imaging, in situ HCR, single-cell RNAseq, velocity, stretch experiments, microCT) to confirm their hypothesis that skeletal fibers directionality was not a mere function of chemical cues, but also depended on mechanical forces via early cartilage interaction during embryogenesis. They provide convincing approaches and results, combining experiments in C2C12 cells, zebrafish embryos, and murine embryos. They used control, morphant, and mutant embryos, as well as pharmacological inhibitors to prove their points. They added compelling evidence using time-lapse videos, showing the importance of the synchronization between early muscle and early cartilage development. In particular experiments involving disruption of cartilage development either using CRISPR/Cas9 or morpholinos of several key cartilage markers highlighted the downstream effects on myofiber polarization and fusion. They ruled out a direct control of the WNT/PCP pathway in the control of muscle directionality, and confirmed the importance of mechanical tension/stretching using experiments in embryos and cell cultures. They also showed the role played by the extracellular matrix in providing directionality cues. Overall the strength of this paper is the cutting-edge techniques used throughout the study to validate their points. This is a great very well written piece of work. The authors provided many movies that are really helpful for the community. The only drawback is the single cell experiment (Fig. 7) which does not appear as convincing as presented. The authors should properly address the following concerns.

Major comments:

1- Missing sections. The authors should provide a Statistics section as specified in the authors' guidelines, and be more specific on the exact n values used for some of the experiments (lines 987-988 'of more than 5 embryos'; lines 1067-1068 'Values are means of more than 4 embryos'). The authors also should add an Authors' contribution section.

2- For the C2C12 2D/3D experiments, what is the relation between 20% continuous stretch (fig. 5) and the soft/hard surfaces (fig. 7a). To how many kPa does 20% corresponds to?

3- Fig. 2d. Why was a single control animal used for the quantification? It is hard to compare with the mutant ones, assess significance, and impossible to evaluate inter-animal variability.

4- Fig. 3b: The *runx2b* morpholino really seems to affect the polarization and development of the muscle cells. However for the *sox9a+9b* CRISPR mutant, it is difficult to say if the process is only delayed in time, or if it will not progress further? Did the author look at a later time point for comparison, for example ~72hpf to see if the mutant caught up with the control?

5- Fig. 3g: The *jamb* morpholino clearly affected fusion of the muscle cells. However, could the authors explain/comment on what are all the small tiny fluorescent dots in the jam MO vs control? It seems that myofiber length is not affected per se (definitely not statistically significant), but that there is a structural issue in these embryos.

6- Fig. 4e: the legend is unclear and the labeling on the panel is not clear either. Not sure which panel corresponds to control vs titin CRISPR mutant. This should be modified and clarified in the figure legend.

7- Fig. 5: This figure has many sub-panels, specifically for panel d and e. It would help if the authors could label the sub-panel, maybe as i, ii, iii, iv, etc... It would also simplify the description in the figure legend.

Panel d (right part) of this figure is unclear. The authors specified that they show the 2 pillars case without stretch due to fragility of the tissues. It seems however from their schematics that the cell orientation is not affected by the absence of stretch? Or would they expect an even better alignment with stretch? It seems the zoomed boxes from this panel were not chosen randomly. What would a zoom of the middle region show (like done in window 2 from the 3-pillar below)? Here having larger pictures would really benefit.

8- Fig. 6: Can the authors develop on the potential effects of SP600125 on cell fusion? It seems clear that with blebbistatin fusion is affected (also shown in fig. 4e). But it is unclear for SP600125. Is the effect only on polarization or also on fusion?

Panel d of this figure does not seem to add much. Also the figure legend is missing the description of panel d.

9- For the scRNAseq experiments, the authors collected C2C12 cells over 2 day prior and over 3 days after differentiation. Although myoblasts (cells prior differentiation) are 'regular' cells to be analyzed by single cell assays (shape, mononuclear), how did the authors proceed for differentiated myotubes that can be really long and polynucleated, especially from the hard substrate condition and may not have passed through filtering step or through the microfluidic? It seems from the methods that the authors performed single cell and not single nuclei assays. How were the cells/myotubes selected or filtered to go through the 10x Genomics chip and protocol? The chip size let cells with a size less than 40um go through. Bigger sizes would lead to clogging of the microfluidics. If filtering, isn't there a great risk of bias toward undifferentiated and thus smaller cells? Indeed the differences between the hard and soft surfaces would have been expected greater, especially based on Fig. 7a, where the 0.5kPa treated cells look very different from the mostly all differentiated 100kPa myotubes. A few smaller myotubes/myocytes may have gone through the analysis as shown from panel 7h (bottom left corner). I think this graph comparison panel in 7h is the important part to highlight as it shows more

than some of the other panels in Fig. 7.

10- Related to point #9 above, it would be useful for the interpretation of the results if the authors could show the individual 5 data points corresponding to the individual days of cell collection, as well as the individual UMI counts for all samples, rather than a pooled UMI UMAP, at least as supplementary figures.

11- For fig.7: It seems unclear if all the cells from individual days were pooled together before running the 10x assay, or at the analysis stage. This should be added in the method, as it is difficult otherwise to interpret the results. If the cells were pooled at the analysis stage (after running individual 10x Genomics assays on individual samples collected each day), then it should be easy to show the individual stages and differences between the soft/hard substrates. If the pooling occurred before running the scRNAseq assay, the samples may have been filtered and there may be a bias on which cells were indeed analyzed (potentially mostly undifferentiated cells). Cells from day 3 in the hard vs soft conditions would be expected to show very different profiles. It seems very strange that only a small proportion of the cells (mainly from cluster 4) is differentiating the hard vs soft surfaces grown cells, as well as an additional myoblast cluster in the hard substrate whereas the general profile is very similar. Another difference is for bridge 2 where there is high velocity on the soft substrate, compared to almost none in the hard substrate condition. From the UMI plots in supplementary fig. 6, it does not appear like some cells have much more UMI and thus are larger which would have corresponded to myotubes (longer + multinucleated) or that the proliferating cells (highlighted on the adjacent supplementary panel) have much more UMI. This is surprising and should be better explained by the authors.

12- Fig. 7: panel d is unclear and not well explained in the legend. Better when looking at the supplementary figures, but still should be better shown/explained in the main text and main figures. I may be missing something, but I am not sure what this panel 7d is adding. There does not seem to be much differences between the 2 conditions, which again point toward a bias in the cells that went through the 10x Genomics chip, probably selected toward the smaller ones. Panels f-h could be combined. I think panel f (left) and panel h (left) should be shown together. Panels f (middle and right) are not adding anything not show already in panel f (left). The legend from panel g is really unclear and should be better explained. It says GO terms relative to cells in cluster 4, but this is showing all clusters, and the enrichment are not all for cluster 4 (only 3 out of 6 sub-panels).

13- Do the hard vs soft substrates both eventually lead to the cell fusion, or does fusion never occur on the soft substrate, even with longer times in culture?

14- Suppl. Fig.6: there are no a, b, c, or d in the figures, although there are in the legend.

Minor comments:

1- Some sections of the main text could go in the Methods. For example, lines 384-387, or lines 487-491.

2- Line 589: the authors should be more specific than 'over many hours'.

3- A few typos, missing bold, language errors, in particular of the Figures legend section.

Some examples, but the authors should review the figure legends:

Line 1028: A should not be capitalized

Line 1040: remove 'were' from 'embryos were served as control'
Line 1044: Instead of '... was ablated by laser pulse in the embryo from the outcross...', prefer 'was ablated by a laser pulse from an outcross of Tg... and Tg... embryo'.
Line 1073: change 'The device of 2D cell-stretcher', prefer 'The 2D cell-stretcher device'.
Line 1082: change 'the area of dashed box', prefer 'the dashed box area'
Line 1092: remove total from 'About total 200 myotubes'
Line 1097: change 'measured for total 200 myotubes' to 'measured for a total of 200 myotubes'.
Line 1110: C should not be capitalized
Lines 1114 /1120: choose between single-cell or single cell, but keep consistency in the manuscript
Line 1142: 'showing expression of example genes', maybe replace with 'showing some examples of expression of key genes'.
Line 1146: may want to change 'we would like to appreciate Dr... for his idea' with 'We appreciate the contribution of Dr...'

In the legend of Suppl. Fig. 6: c and d should be bolded.

4- Fig.2b, the authors should add a box of the zoomed area as it is unclear from the legend which area is actually shown as a blow-up of the top panel. There are 2 dark blue arrows on the top panel, and so which one corresponds to the area shown on the bottom panel?

5- Fig. 2a, fig. 2b, and fig. 2g: the authors should use colors that distinguish better the arrowheads. Also they should specify what the different colors mean in the figure legend as this is missing.

6- For many of the zoomed figures and some other figures, the resolution is too low and should be increased. It is hard to see (figs 3f, 4b, 5c, 6c). Some panels look very tiny and would benefit from being larger with a larger area shown, see for examples fig. 1f,g; fig. 5.

7- Supplementary figs 5-9 are not labeled as such. Suppl fig. 8 is on 2 pages.

8- The authors may want to follow the standard zebrafish orientation on figures, with the head on the left, tail on the right, dorsal on top, ventral on the bottom. Some of the pictures do not respect the convention.

Dear Referees, we took your comments very seriously and run a more than a year-long revision to address all your questions. We hope the additional work and improved results will satisfy all concerns and encourage you to be more positive towards publishing our study.

Reviewer #1 (Remarks to the Author):

The study by Sunadome et al, aims to examine the role of cartilage in regulating the development of striated muscle, specifically examining myocyte orientation and fusion, both of which are tightly regulated processes. Using a combination of model systems including cell culture, zebrafish and mice, they reveal that muscle and cartilage development is synchronized and the mechanical forces generated by growing cartilage dictates the growth of muscle. This is supported by the observation that disrupted cartilage formation, either using genetic approaches or ablation strategies, results in impaired muscle patterning and fusion. The authors further reveal a role of extracellular matrix proteins and the JNK pathway in regulating the orientation of muscle cells.

Finally, to identify how the mechanical environment influences muscle development, the authors performed single cell transcriptomics on cells grown on soft and hard substrates. These analyses reveal a role of mechanical forces in influencing the transcriptional state of the muscle fibres, specifically by altering the expression of mechanosensory pathways, although no further functional studies were performed to validate these studies. Overall, while the question the authors set out to tackle is interesting and has significant implications on ongoing efforts to generate *ex vivo* grown muscle, the manuscript appears to be several small experiments have been performed independently, and subsequently combined into a single story. This has resulted in a disjointed and non-cohesive piece of work that was extremely difficult to follow, subsequently diluting the overall finds of the paper.

We are grateful for all advices and apologize for the non-cohesive structure of the earlier draft of the manuscript. We thank the reviewer for putting the effort into understanding it. During the revision, we have re-written the paper, re-arranged figures, added new clarifying panels and new experiments, and we hope the reviewer finds it improved and better connected.

Major comments:

1. While the authors clearly show a role of cartilage development in regulating muscle development, they have not determined the mechanistic basis of this regulation. While it is possible that changes in passive force following growth of cartilage is plays a role in dictating muscle development, the authors have not negated alternative non-autonomous mechanisms.

The major aim of this study was to answer which developmental interactions prime future muscle orientation. We found a key guiding role of the early embryonic skeleton in this process. Next, we attempted to address the mechanistic and molecular basis of this interaction, which is an extremely challenging question. We believe that during this revision we made a better progress towards this aim, which is outlined in responses below.

For example, it is possible that deposition of proteins by tenocytes drives muscle development, and without examining such mechanisms it is incorrect to conclude that the “directionality of developing skeletal muscle fibres is set by mechanical forces”.

We understand and agree with the reasoning of the reviewer. Indeed, tenocytes might potentially play a role in muscle development. To address this hypothesis during the revision, we performed a number of additional experiments listed below. According to our new data, tenocytes emerge only after the initial interactions between polarizing myoblasts and cartilaginous condensations have begun. The new staining shows that in mouse embryos, cartilage condensations are present from as early as E10.5-E11.5 in the face. Scleraxis, one indicator of tenocyte differentiation, is also expressed in the pre-cartilage condensations, often prior to tendon formation (<https://pubmed.ncbi.nlm.nih.gov/7743923/>), whereas Tenomodulin, a gene more specifically expressed in mature tenocytes, only shows up after E12.5-E13.5 after tenocytes consolidate their fate. In our updated Figure 3, we now show the synchronized development of MyoD- and MyHC3-expressing muscle cells with Col2a1+ cartilage elements at E11.5 in the embryonic mouse face with whole-mount HCR *in situ*. Using this technology during the revision, we did not detect Scx in the face at this same time point (E11.5), although we identified clear expression of Scx at E12.5 (updated Supplementary Figure 2b).

Similarly, in zebrafish embryos, we find that Scxa is not expressed at 44hpf (Supplementary Figure 2a), and we detect weak expression of Scxa at 48 hpf (Supplementary Figure 3a), already after the polarization of muscle fibers begins, according to new live imaging data (updated Fig. 4a). However, expression of Scxa indicates committed tenocyte progenitors, but does not necessarily indicate functional maturation of the tenocyte fate. Scxa expression could be found in all cartilage-perturbed embryos, showing that tenocytes were still specified in conditions where muscle polarity is greatly disrupted. This blunts the observation that tenocyte specification occurs during myocyte elongation, and begs the question of whether functional markers of mature tenocytes (Tnmd, Col1a1) might also coincidentally arise. To check this during the revision, we found tnmd expression at 60 hpf, hours after the initial lengthening of developing myofibrils along the direction between cartilage attachment points (Supplementary Fig 2c).

Consistently, one recent report (<https://dev.biologists.org/content/141/10/2035>) shows expression of scxa in fish as early as 56 hpf, and one image with very weak staining is shown to represent the expression of scxa at 48 hpf. The same paper did not find tnmd expression before 72 hpf. Thus, it is unlikely that the initial stages of polarization are driven by tenocytes expressing tenocyte markers, but tenocytes play an important role in later developmental stages.

Also, during the revision we found that Col1a2 was expressed throughout the process of myocyte elongation. However, the established tendon matrix does not appear to affect the early polarity or organization of the muscle fibers. To prove this, we performed a new Crispr/Cas9 knockout of multiple tendon matrix proteins simultaneously, using gRNA library targeting the TSP4b, Col12a1a, Col12a1b, Col22a1a genes, and a separate experiment using a gRNA library targeting Cola1a, Cola1b, Cola2 genes. This experiment can also be found in updated Supplementary Figure 2d. After the knockdown of these tendon matrix gene sets, the muscle fibers polarity, myocyte shape, and muscle patterning, were unaffected by the perturbation, whereas other features in the embryo were perturbed.

On the other hand, muscle appears to play an important role in the development and specification of tendon and ligament progenitor cells, as knockdown of *myoD* or *myf5* inhibits specification of *xirp2a* in the face and trunk, and inhibits the maintenance of *scxa* in the face (<https://dev.biologists.org/content/141/10/2035>).

Overall, we added new text in the Results section to outline our observations during the revision and to discuss how the tenocytes and early myocyte polarization are related to each other:

“Because tendons produce the anchoring point that attaches muscle to cartilage in the adult, we assessed the role of tenocytes in the early stages of muscle polarization. We explored the expression of Xirp2a, a previously reported marker of tendon cells in zebrafish (Chen and Galloway, 2014). Xirp2a mRNA accumulates both in the tip of the muscle (Xirp2a+, MyHC+) and the tendon (Xirp2a+, MyHC-) as organized units adjoining to cartilage (Fig. 2e) at 72 hpf. Sox9-deficient embryos showed a slightly more diffuse and disorganized expression of Xirp2a, although the general pattern across the entire face was preserved. Also, the Xirp2a+/MyHC+ muscle tips detached from Xirp2a+/MyHC- tendon cells, corresponding to the formation of reduced and fractured cartilage in this mutant. A similar result of muscle and tenocyte disorganization was also obtained by examining the expression of Col2a1, Tnmd and MyHC at 72 hpf in a stable mutant line Sox9atw37/tw37 (Supplementary Fig. 1d). Runx2b-deficient embryos exhibited no Xirp2a+ tendon cells with scattered muscle tips (Fig. 2e), and the shortage in tendon formation was confirmed by a loss of the Tnmd expression pattern at 72 hpf (Supplementary Fig. 1e). Thus, severity of cartilage perturbations correlates with phenotype severity of both myocytes and tenocytes, but a hierarchy of how each cell population contributed to the phenotype was still unclear.

It was previously reported that cranial tenocytes and myocytes are specified independently in zebrafish but that muscles are required for tendon maintenance and maturation (Chen and Galloway, 2014). Consistent with this idea, expression of Scxa, a transcriptional regulator of tenocyte cell identity, was absent at 44 hpf but visible at 56 hpf by whole mount ISH (Supplementary Fig. 2a), after muscles already begin elongating (Fig. 1b). That the induction of Scx+ tenocytes occurs later than that of Myod+ myoblasts was also confirmed in mouse (Supplementary Fig. 2b). Time-course experiments revealed that Tnmd expression was visible by 60 hpf but not at 48 hpf, suggesting muscles elongate significantly before tenocytes mature (Supplementary Fig. 2c). Finally, we performed CRISPR/Cas9-based genetic screening with gRNA pools targeted at the tenocyte matrix, and found minimal effect on musculoskeletal morphologies (Supplementary Fig. 2d and 2e). This data suggests that the mechanism establishing myofibril polarization depends primarily on cartilage growth, and does not necessarily require mature tenocytes. Further, many components of the tendon-associated extracellular matrix appear dispensable for this mechanism to remain functional.

*Focusing on the AM muscle specifically, we further confirmed the developmental timelines are offset between muscle and tendon by assessing gene expression over time using a panel of tendon markers including *scxa*, *tnmd*, and *colla2* (Supplementary Fig. 3a). Further, labeling tenocytes in cartilage-less embryos revealed that although *tnmd* expression is lost in *Runx2b* MO-injected embryos, *scxa*+ tenocyte progenitor cells and *colla2*+ cells could be found in the face in all conditions (Supplementary Fig. 3b). These results further indicate that the muscular phenotype resulting from cartilage perturbation is not likely driven by effects on tenocytes.*

Taken together, these results reveal that cartilage growth regulates facial muscular development. Depending on its severity, cartilage growth defects cause disruptions in myoblast cytoskeletal organization and polarity, myofiber elongation and fusion, and the resulting muscle morphology.”

2. The authors have presented single cell RNA seq studies on cell grown in soft and hard matrices, to decipher how mechanical environment influences muscle development. While this data has clearly shown that the mechanical environment can influence the transcriptional state of myocytes, further functional studies are required to validate their findings and support their conclusions in an *in vivo* setting.

We agree and understand the reviewer’s concern, and thus, before diving into further validating experiments, we decided to improve the single cell data analysis and also to cross-compare it with deeper sequenced bulk batches of cells collected at specific timepoints at hard or soft matrixes. The advanced analysis of bulk RNA sequencing and single cell transcriptomic analysis of these samples agreed that transcriptional differences between hard and soft matrix-cultured cells were rather quantitative (not qualitative). Thus, we acknowledge that there is likely feedback between cell polarity, cell morphology, cytoskeletal organization, and resulting gene expression status. However, we made the finding that 1. temporal variation accounted for 98% of gene expression differences in bulk RNA data (whereas hard/soft condition only made up 2% of the gene expression across samples when analyzed in a PCA), combined with our finding that 2. Overrepresentation of hard/soft cultured cells was minor across most cell clusters, particularly the most mature myocyte clusters (there was no cluster or fate with an identity specific to the hard condition as we expected in both bulk and single cell analysis). We interpret this result with caution, and as an evidence that the effect of mechanical forces on muscle cells might rely less on differences in gene expression to drive cell polarity. One possible explanation for this result is that the effect of mechanical forces on cell polarity and morphology might occur mostly at the protein and cell architecture levels.

Although we agree that the transcriptomics part would be stronger with subsequent validations *in vivo*, the validations of weak and quantitative (not qualitative) predictions (no clear cut transcriptional mediators of mechanically-driven changes in cell morphology) is more difficult than validating a positive result (if we would have found specific gene expression signatures with expressed vs not expressed genes driving this change). We did, however, perform morpholino experiments during the revision with the goal of testing the role of a few genes found to be upregulated in hard/soft cultured conditions (injection of Ppp1r14b MO and SLC25a4 MO at the single cell stage), and did not find any phenotype in tendons, cartilage, or muscle *in vivo* at 72hpf. We did not put these negative data into a new figure but would be happy to share these data on request.

3. To examine how disrupted cartilage development impacts muscle development, the authors heavily rely on data from CRISPR/Cas9 edited larvae. A major criticism of this approach is that not all cells display the same mutation, and that different embryos would show different degrees of mutagenesis. Therefore, a more appropriate approach is to either generate and analyse stable mutants line whereby all cells show the same mutation, or utilize a tissue specific knockout approach, whereby the target gene is mutated specifically in the cartilage. It is also misleading and inaccurate to call these injected embryos mutants.

We agree that analysis of CRISPR/Cas9-edited larvae is tricky, and we should be cautious in our interpretation and reporting of results. Our method for CRISPR/Cas9 gene editing involves injection of gRNA and Cas9 at the single-cell stage. Because of this fact, within a given injected embryo, all cells in the embryo should display the same mutation, because the editing should happen prior to the first cell division. We agree that with some targets, it is possible that different embryos could show different degrees of mutagenesis after injection. However, the quantifications for muscle length and fusion demonstrate an agreement across many injected fish per target in the study and demonstrate the specificity of phenotypes observed via gene editing. Additionally, we have performed analysis of the musculoskeletal development in stable fish lines, fish with point mutations resulting in truncations of the proteins Sox9a, Itgb1a, and Runx2b. During the revision, we found that stable genetic disruption of the Sox9a gene (using the Sox9a^{tw37} mutation) results in the same phenotype in the muscle and cartilage that we observed via CRISPR/Cas9 editing Sox9a/b – that is to say, cartilage was under-developed and muscle was disorganized (Supplementary Figure 1). In our analysis of Runx2b and Itgb1a stable lines (also point mutations resulting in early truncations) we did not observe the expected phenotype in the cartilage or muscle of embryos that were verified to be mutant by sanger sequencing, and allele-specific PCR amplification, separately (data not shown, can be provided upon request). We attribute the different phenotypes between the stable lines and the CRISPR/Cas9 or morpholino studies to be a matter of compensation, as often phenotypes from morpholino experiments are much more severe than analogous phenotypes observed in experiments involving stable mutation lines. Nonetheless, the tw37 fish recapitulate the phenotype of editing via Sox9a/Sox9b gRNA, which we hope the reviewer will agree helps to validate our findings. In addition, according to the reviewer's advice, we have changed all the terminology to be correct when referring to the genetic perturbations of the embryo.

4. The resolution of some of the data presented is relatively low to draw conclusions. Higher resolution images need to be provided. Relevant datasets include Figure 1c, where the MyHC images are highly saturated making it difficult to clearly see individual muscle fibres and subsequently the misalignment; Figure 3F, specifically the runx1 cells; Figure 4a, specifically the bottom panels which are blurry; Figure 4b.

We thank the reviewer for pointing out which images need improvement in our manuscript, and we have improved the quality/resolution of the aforementioned figures. The apparent saturation of the images Figure 1C is a symptom of being a maximum intensity projection, making individual fibers difficult to see. Individual fibers are readily apparent in thin orthogonal projections, however, because the curvature of the face, capturing sufficient depth to see the whole muscle in a single image required a thicker projection. To provide additional resolution in the Z-axis, during the

revision, we included separate Supplementary Videos 3-5 which shows sliding 4um orthogonal projection through each fish embryo in 1C. We hope the reviewer agrees the video helps manifest the disorganization of individual muscle fibers.

5. The initial studies in zebrafish reveal that early muscle and cartilage development is synchronized in zebrafish, and that cartilage is required for the development and maintenance of muscle. However, the authors only examined this in the context of facial muscle. Is this also the case in the trunk skeletal muscle, or is the development of these muscle groups independent of cartilage development? It would be beneficial to examine trunk muscle and cartilage patterning at earlier stages when skeletal muscle is developing, and also perform cartilage loss of function experiments to determine how trunk muscle development and homeostasis is maintained.

We have also examined the myofiber formation in the trunk muscles during this revision, where myogenin-expressing cells arrange into myosepta along the body axis. We found that these muscles don't appear to be attached to pre-skeletal elements during their polarization. The mechanism of myocyte polarization in the trunk may also rely on mechanical forces, but does not appear to require muscle-cartilage attachment points. The new video is attached as Supplementary Video 18. Indeed, the attachment of trunk muscle is mediated by myosepta, which is a non-skeletal element. Our obtained time-lapse images show that early unpolarized myocytes attach to the myosepta (dashed line) and become polarized, and then stretched according to the growth of the trunk. Thus, the principle of muscle polarization shown in the manuscript (attachment and oriented stretch) is expected to work here as well, even though the structure that supports muscle attachment is different from cartilage.

Minor comments

1. All figure panels need to be structured better, as currently the individual panels are extremely difficult to decipher. For example, it would be beneficial to add additional sub-labels such as Ai, AII, Aiii to help follow which panels the authors are referring to. Labels also need to be added to detail the different views. For example, what are the 4 different 48 h images in Figure 1b, and which ones are the authors referring to in their discussion?

We agree and we tried to improve the figures since the last submission. Regarding the Figure 1b we improved clarity of the text to indicate the reason for the multiple panels: each example was showing the same process happening in different facial muscles. We hope it is clearer now.

2. Please use correct terminology for the zebrafish loss of function experiments: Since the authors have analysed gRNA injected embryos, whereby mutagenesis is mosaic, it is incorrect to call them mutants. Please denote gRNA injected embryos as CRISPR/Cas9 edited larvae.

We have changed the terminology, and we thank the reviewer for the good suggestion.

3. Line 119: “By 72 h.p.f., cartilage and facial muscles already establish a mature interconnected pattern (Supplementary Video 1, and Fig. 1a).” The legend of Figure 1a however states that the image is from an 81 hpf larvae. Please correct the discrepancy in stage.

We have corrected this error, and we apologize for the misleading labels.

4. Line 131: Suggestion. “Moreover, before cranial cartilaginous elements achieve their shape, newly born individual muscle cells connect to clusters of the coalescing chondrocytes in a mouse embryo (Fig. 2a and 2b, see below)”. Discussing mouse data at this point breaks the flow of the zebrafish results. Please omit mouse data results from here, and move to the section titled: “Cartilage development coincides with muscle and controls directional outgrowth in a mouse”.

We have mostly reorganized the discussion of different model systems to reduce any confusion. Thank you for the suggestion.

5. Line 151: “although they displayed no or marginal change in the eye size, suggesting a normal progression of general body development”. The authors need to include representative images if they would like to make claims on the eye size. Additionally, their analyses clearly shows a significant reduction in the eye size in *runx2b* morphants, saying the statement they have made is misleading. The smaller eyes in the morphants suggest developmental delays and could explain why “functional inactivation of *Runx2b* produced more severe effects”.

We have to thank the reviewer for causing us to think more deeply about the caveats in our work. We have quantified eye diameter in our data and found a small but significant difference in eye size in *Runx2* morphants. Despite this, we observe variation of similar magnitude among different control conditions, and eye size variation is greater in *Runx2* MO compared to other conditions (that is, some individual *Runx2* MO eyes match some controls). In stark contrast to the eye size variation, the skeletal phenotype in *Runx2* MO is orders of magnitude more severe than normal variation among controls (that is, *Runx2* MO practically eliminates the cartilage, an effect that would be virtually impossible to mistake for a control embryo). Therefore we would like to insist that the changes in eye size can be accurately described as comparatively marginal, and it is unlikely that developmental delays indicated by differences in eye size could reasonably explain the drastic phenotype in the musculoskeletal system. For instance, we feel it is important to note the muscle cells still differentiate in these embryos, but fail to attach to cartilage. However, we updated the text to admit that we cannot completely rule out the possibility of nonspecific developmental delays affecting the phenotype of the *Runx2* MO.

“Although we cannot completely rule out that developmental delays contribute to the phenotype caused by Runx2b deficiency, these results suggest that correct morphogenesis of some zebrafish facial muscles depends on correct development of the skeletal elements to which they attach.”

6. Figure 1C – the graph for the eye diameter is missing a y axis label.

The y-axis label has been added in figure 1C.

7. Figure 1G – Please provide single channel images (in addition to the merged images).

Now, the single channel images as well as the merged images are provided (these panels have been reorganized and now the data is in a new panel (Figure 2e) about the effects cartilage ablation had on the cytoskeletal organization in muscle.

8. Figure 2H and line 228. Discussion of the fish results is disrupting the flow of the mouse work discussion. A suggestion is to move Figure 2H to Figure 1, where are the remaining fish data is presented.

We agree with the reviewer that it is better to keep the various model systems separate. We have moved this to Figure 1c, and we think the flow should be improved now.

9. Figure 3A: Please add arrowheads to identify the cells of interest

We have now added arrowheads (now Figure 4a) to indicate the tips of the growing muscle primordium.

10. Line 241: “Our live imaging data revealed that the primordium of AM muscle starts to interact with palatoquadrate cartilage as soon as they simultaneously emerge”. Since both, cartilage and AM myocytes are already present at first time point examined (44 hours), it cannot be concluded that the interaction begins as soon as the both emerge. Earlier timepoints need to be examined with higher temporal resolution to make this claim.

We acknowledge the tricky nature of the initial claims, and we have altered the text in the manuscript to reflect the nuance of developmental timing and cell identity. What constitutes “cartilage emergence” is a difficult question in its own right and subject to debate regarding which characteristics would be most relevant identifiers of so-called “cartilage”, and might be partly a matter of semantics. As the pre-cartilaginous condensations are growing and taking shape at the gross anatomical level, the cells are also differentiating by adopting a round cell morphology and increasing expression levels of Collagen II.

Although the cell morphology of cells inside the pre-skeletal element palatoquadrate may resemble chondrocytes by 44h, examination by HCR at 48hpf shows very little expression of col2a1 compared to 60hpf, indicating that chondrocyte differentiation hasn't fully completed even by that

time. Also, Col2-mCherry label is only barely apparent several hours after 48hpf. However, Col2a1 gene expression can precede the existence of a morphologically distinct skeletal element, which might be another reasonable sense in which a cartilage could “emerge”.

We would like to point out that there is precedent in the literature for the claim that craniofacial cartilages have not yet emerged by 44hpf (Schilling TF et al, Development, 1997 and seven older publications cited by Schilling et al on this point). The earliest to emerge is the trabecula (45hpf), but this cartilage doesn't interact with the AM muscle. The palatoquadrate cartilage was reported in that publication to appear at 53hpf. The AM muscle, in the same publication, is cited to appear at 53hpf.

The discussion about the timeline regarding interactions between these anatomical structures and their developmental precursors has been updated:

“Our live imaging data revealed that the primordium of AM muscle starts to interact with palatoquadrate cartilage condensations as soon as they simultaneously emerge between 48 and 53 hpf”

11. Figure 3bb: Please add details of statistical significance to the changes in polarity in the sox9 and runx2 deficient fish.

done

12. Figure 3e. While in the middle panel it is clear where the ablation is, in the bottom panel the ablation site is not clear. Figure 3F: To examine cell division, please clearly show the myofibril and/or basement membrane.

done

13. Line 321 and Figure 3a: After cutting, the tip of myocyte moves away from the ablated sites along the axis of the muscle tissue”. Rather than an active migration of the myocyte, this appears to be detachment of a cell, due to the loss of attachment site, as seen in muscular dystrophy. Given the former isn't experimentally tested, please amend to present it as a hypothesis rather than a validated finding.

This is a good point brought up by the reviewer. We have re-worded the text to carefully weigh the alternative explanations for the possible movement of the myocyte. However, we would like to point out that the word “move” does not imply a passive nor an active process – further, we did not make use of the term “migration” to interpret the results of this particular experiment. We have

made the text clearer in our interpretation that the loss of attachment sites caused passive movement of the myocyte, which is consistent with the presence of mechanical tension placed on the muscle by cartilage, transmitted via the attachment site.

Updated text: *“We ablated the attachment point between myocytes and cartilage (Fig. 4d, the region encompassed with bracket) with 2-photon laser, and observed the response with live-imaging. After cutting, the tip of myocyte appears to detach and drift away from the ablated sites along the axis of the muscle tissue (Fig. 4d, Supplementary Video 11). We measured the distance between the tip of the myocyte and the ablated site, and found that the recoil velocity decreased over time (Fig. 4d, right panel), indicating a viscoelastic response that depends on the tension and friction-like resistance in muscle primordia. Importantly, this model assumes no active cell migration, but rather that external forces cause the cells to move after they have been detached from the cartilage by laser ablation.”*

14. Figure 4b: The phalloidin seems to be labelling the sarcomere of muscle fibres, not stress fibres. The disruption of F-Actin staining in the *sox9* and *runbx2b* depleted larvae is not surprising given that the authors have already shown that sarcomeric structure is also disrupted in them. These results need to be combined with the Figure 1f, or removed and the relevant sections in the text need to be amended.

We agree that the organization of the previous draft was lacking and thank the reviewer for helping us to consolidate redundant findings. The results have been combined in Figure 2. Also, stress fibers are re-worded as *“actin fibers”*.

15. Additionally, given that phalloidin labels muscle and cartilage, the accumulation on “stress fibers” at the muscle-cartilage is not clear. Perhaps a heatmap can be used to demonstrate this point

We used stress fiber to describe (presumably contractile) actin bundles found to be at the musculoskeletal interface, because this feature of actin cytoskeletal organization in muscle was disrupted in cartilage-perturbed conditions. We have re-worded this to be less confusing, because “stress fibers” are not supposed to be in muscles. The accumulating actin fibers are shown as a heat map in a new panel, shown in Figure 2c.

The new text reads:

“Given that tensile strain can influence cell shape via actin fibers, we tested whether cartilage disruptions could cause alterations in actin cytoskeletal organization in early developing myocytes. In control embryos between 48 and 81 hpf, phalloidin staining revealed accumulation of actin fibers at the interface between muscle and cartilage. Here, filamentous actin fibers start and run along the longitudinal direction of the elongating muscle (Fig. 2c). Polarized fibers were abrogated in cartilage-reduced mutants at 52 hpf (Fig. 2d), consistent with a model in which these embryonic structures failed to transmit force to differentiating myoblasts.

16. The relevance of titin experiments is not clear. Please elaborate why titin loss of function experiments were performed.

The previous experiments suggested a potential molecular mechanism in muscle to sense mechanical tension mediated by muscle attachment to cartilage. Because Titin has a known function as a force sensor, and has known roles in stabilizing the sarcomeric structure in mature muscle, we hypothesized that Titin could potentially be involved in the initial polarization steps of the muscle. It turns out we were wrong, but this is the reason why the experiment was performed. In addition, Titin was found to have drastically higher levels of expression in soft- versus hard-cultured myocytes *in vitro* (which can be found in the new Supplementary table 2).

Reviewer #2 (Remarks to the Author):

Comments to the author: “Directionality of developing skeletal muscles is set by mechanical forces”

In this study, the authors investigate how cartilage expansion influences attached myocyte polarization, orientation, and fusion during embryonic development. Using a combination of loss-of-function studies, live and advanced imaging, and C2C12 cell stretching experiments in 2D and 3D culture the authors demonstrate that the directional outgrowth of developing myocytes and myofibrils are regulated by mechanical force. Additionally, the authors attempt to dissect the molecular basis of this phenomenon by functionally targeting members of the WNT and JNK signaling pathway. Using single cell RNA-seq, the authors also examine whether cell polarization is required for muscle cell differentiation by collecting C2C12 cells cultured on hard or soft substrates. Using this approach, the authors analyze the transcriptional dynamics across different stages of skeletal muscle differentiation and across different conditions. Furthermore, gene expression differences between hard and soft cultured cells further support a role for mechanical transduction and signaling in myocyte orientation and polarization.

The manuscript is well-written with the aims, main findings, and methods clearly stated. Moreover, the authors outline the purpose, design, and rationale behind each experiment throughout the results. The findings are strengthened by the use of both zebrafish and mouse animal models in addition to the use of the mouse C2C12 cell line, demonstrating that their findings are reproducible *in vivo* and *in vitro*. Overall, this is an interesting manuscript with broad applications, yielding insight into how tissue interactions and mechanical forces mediate development of the musculoskeletal system and will be of considerable interest to readers.

The adjoining of muscle to cartilage is mediated by tendon and thus, one would predict tendon too is subject to mechanical tension generated by cartilage expansion. Tendon itself may also produce mechanical tension (see Kalson et al., PNAS 2013) as it forms and grows that is applied to the forming muscle. Others have shown that connective tissue can affect muscle development independent of any defects in skeletal formation (Hasson et al., 2010). While the authors mention tendon in the introduction, it would greatly help the manuscript to have a greater characterization

of the tendon cells, particular how the alignment of collagen fibers and/or tendon cells are affected. Early in the manuscript, the authors demonstrate that *runx2b*-deficient embryos have disrupted *xirp2a* expression, but as *xirp2a* does not mark all tendon cells, the authors should examine additional tendon markers such as *scxa*, *colla2*, or *tnmd* expression in *runx2b* morphants. The authors show *xirp2a* expression pattern is disrupted in *sox9a/b*-deficient embryos and from previous reports, *scxa* expression pattern also is disrupted in *sox9a/b*-deficient embryos, indicating that tendon morphology is equally dependent upon *sox9a/b* and likely cartilage formation.

Overall, this raises the question of whether myocyte polarization is directly regulated by cartilage expansion or alternatively, via its effects on tendon development, or perhaps by both tissues. This is not a fundamental issue with the manuscript as the authors demonstrate that mechanical tension is an important regulator of myocyte polarization and orientation directly through their *in vitro* work. However, I recommend that the authors examine additional markers of tendon in *runx2b* deficient zebrafish and cartilage-ablated mice, and include some discussion of the tendon's mechanical properties and its relation to both cartilage and muscle during development. This would provide greater context to the functional relationship between the different musculoskeletal tissues and their influence on one another.

Thank you for your comments. We improved the discussion and added much more data on the role of tendons (please see the related new Supplementary figures 2 and 3 about the role of tendons).

Indeed, tenocytes might potentially play a role in early muscle polarization. To address this hypothesis during the revision, we performed additional experiments. According to our new data, tenocytes emerge only after the initial interactions between polarizing myoblasts and cartilaginous condensations have begun. It was shown that in mouse embryos, cartilage condensations are present from E10.5-E11.5 in the face. Scleraxis, the marker of tenocyte fate, is also expressed in the pre-cartilage condensations, often prior to tendon formation (<https://pubmed.ncbi.nlm.nih.gov/7743923/>), whereas Tenomodulin, a gene expressed in more mature tenocytes, only shows up after E12.5-E13.5 after tenocytes consolidate their fate. In our Figure 2, we show the synchronized development of MyoD- and MyHC3-expressing muscle cells with Col2a1+ cartilage elements at E11.5 in the embryonic mouse face. At this same time point E11.5, we did not detect Scx in the face, although we saw clear expression of Scx at E12.5 (Supplementary Figure 2b). Similarly, in zebrafish embryos, we found during this revision that *Scxa* is not expressed at 44hpf by whole mount *in situ* hybridization (the new Supplementary Figure 2a), and we detected weak expression of *Scxa* only later at 48 hpf by HCR (the new Supplementary Figure 3a). Thus, it seems that tendons are being specified around a time at which muscles are elongating.

Consistently, one recent report (<https://dev.biologists.org/content/141/10/2035>) shows expression of *scxa* in fish as early as 56 hpf, and one image with very weak staining is shown to represent the expression of *scxa* at 48 hpf. The same paper did not find *tnmd* expression before 72 hpf. We found *tnmd* expression around 60hpf but not before (the new Supplementary Figures 2 and 3). Thus, it is unlikely that the initial stages of polarization are driven by mature tenocytes. One could make the argument that immature *scxa*⁺ tenocytes are instead the cause; however, *Runx2b* MO-injected and *Sox9a+b* gRNA injected embryos both show normal specification of tenocytes (*scxa* expression), so the effects of cartilage ablation on muscle aren't apparently mediated by tenocyte progenitor cells.

Secondly, the established tendon matrix does not appear to affect the early polarity or organization of the muscle fibers. To test this, we performed Crispr/Cas9 knockout of multiple tendon matrix proteins simultaneously, using gRNA library targeting the TSP4b, Col12a1a, Col12a1b, Col22a1a genes, and a separate experiment using a gRNA library targeting Cola1a, Cola1b, Cola2 genes. This experiment can also be found in the new Supplementary Figure 2d and 2e. After the knockdown of these tendon matrix gene sets, the muscle fibers polarity, myocyte shape, and muscle patterning, were unaffected by the perturbation, whereas other features in the embryo were perturbed. On the other hand, muscle appears to play an important role in the development and specification of tendon and ligament progenitor cells, as knockdown of myoD or myf5 inhibits specification of xirp2a in the face and trunk, and inhibits the specification and maintenance of scxa in the face (<https://dev.biologists.org/content/141/10/2035>).

Overall, we added a new text in Results and Discussion section to outline our observations and to discuss how the tenocytes and early myocyte polarization are related to each other:

“Because tendons produce the anchoring point that attaches muscle to cartilage in the adult, we assessed the role of tenocytes in the early stages of muscle polarization. We explored the expression of Xirp2a, a previously reported marker of tendon cells in zebrafish (Chen and Galloway, 2014). Xirp2a mRNA accumulates both in the tip of the muscle (Xirp2a+, MyHC+) and the tendon (Xirp2a+, MyHC-) as organized units adjoining to cartilage (Fig. 2e) at 72 hpf. Sox9-deficient embryos showed a slightly more diffuse and disorganized expression of Xirp2a, although the general pattern across the entire face was preserved. Also, the Xirp2a+/MyHC+ muscle tips detached from Xirp2a+/MyHC- tendon cells, corresponding to the formation of reduced and fractured cartilage in this mutant. A similar result of muscle and tenocyte disorganization was also obtained by examining the expression of Col2a1, Tnmd and MyHC at 72 hpf in a stable mutant line Sox9atw37/tw37 (Supplementary Fig. 1d). Runx2b-deficient embryos exhibited no Xirp2a+ tendon cells with scattered muscle tips (Fig. 2e), and the shortage in tendon formation was confirmed by a loss of the Tnmd expression pattern at 72 hpf (Supplementary Fig. 1e). Thus, severity of cartilage perturbations correlates with phenotype severity of both myocytes and tenocytes, but a hierarchy of how each cell population contributed to the phenotype was still unclear.

It was previously reported that cranial tenocytes and myocytes are specified independently in zebrafish but that muscles are required for tendon maintenance and maturation (Chen and Galloway, 2014). Consistent with this idea, expression of Scxa, a transcriptional regulator of tenocyte cell identity, was absent at 44 hpf but visible at 56 hpf by whole mount ISH (Supplementary Fig. 2a), after muscles already begin elongating (Fig. 1b). That the induction of Scx+ tenocytes occurs later than that of Myod+ myoblasts was also confirmed in mouse (Supplementary Fig. 2b). Time-course experiments revealed that Tnmd expression was visible by 60 hpf but not at 48 hpf, suggesting muscles elongate significantly before tenocytes mature (Supplementary Fig. 2c). Finally, we performed CRISPR/Cas9-based genetic screening with gRNA pools targeted at the tenocyte matrix, and found minimal effect on musculoskeletal morphologies (Supplementary Fig. 2d and 2e). This data suggests that the mechanism establishing myofibril polarization depends primarily on cartilage growth, and does not necessarily require mature tenocytes. Further, many components of the tendon-associated extracellular matrix appear dispensable for this mechanism to remain functional.

*Focusing on the AM muscle specifically, we further confirmed the developmental timelines are offset between muscle and tendon by assessing gene expression over time using a panel of tendon markers including *scxa*, *tnmd*, and *coll1a2* (Supplementary Fig. 3a). Further, labeling tenocytes in cartilage-less embryos revealed that although *tnmd* expression is lost in *Runx2b* MO-injected embryos, *scxa*⁺ tenocyte progenitor cells and *coll1a2*⁺ cells could be found in the face in all conditions (Supplementary Fig. 3b). These results further indicate that the muscular phenotype resulting from cartilage perturbation is not likely driven by effects on tenocytes.*

Taken together, these results reveal that cartilage growth regulates facial muscular development. Depending on its severity, cartilage growth defects cause disruptions in myoblast cytoskeletal organization and polarity, myofiber elongation and fusion, and the resulting muscle morphology.”

Here are some minor comments:

- Line 131-132: “...muscle cells connect to clusters of the coalescing...” implies that muscle is directly attached to the cartilage rather than by tendon. This statement also assumes the tissues are attached but evidence of adhesion is not supported without demonstrating the presence of adhesion molecules such as integrins.

This is a good point. Please see our updated Supplementary figure 7 showing a new experiment to report localization of ITGB1 expression in zebrafish face.

Many adhesion proteins including ITGA5 and ITGB1 are expressed at the attachment point of cartilage, tendon, and muscle (LaMonica et al Genesis 2015 <https://pubmed.ncbi.nlm.nih.gov/25810090/>) because there is widespread of adhesion proteins in the pharyngeal arch mesenchymal derivatives generally speaking. We would like to direct attention to our laser ablation of the attachment points, where muscle and cartilage tissues became disconnected thereafter distance increased between them, which suggests that these tissues were in fact attached either directly or indirectly. We want to be very clear about the point that we are not claiming that muscle directly attaches to cartilage without a tendon intermediate (even though this may be true).

Our new data on the tendon, and with further reflection on the nuanced situation regarding developmental timelines and developmental identities makes it very difficult to prove or disprove such a particular claim (about direct attachment), because tendons may arise from cells at the junction between pre-cartilaginous condensation and pre-muscle. Whether or not the attachment involves an intermediate tissue in the earliest stages of muscle polarization, the conclusion is that the muscle and cartilage are attached, and that mechanical force transmitted by this attachment point is important for muscle development.

- Line 139: tendocytes should be changed to tenocytes

We apologize for that mistake and we fixed the nomenclature, as requested.

- Line 180- should say “exhibited no *xirp2a*+ tendon cells” as there still could be *scxa*+ tendon cells present.

That is a good caveat to clarify. To cross-examine the fate of tenocytes in the *Runx2* knockdown, we have performed an additional in situ hybridization of the *Runx2* morphants and confirmed that in addition to *xirp2a* loss, we also saw that *tnmd*+ tendon cells are also not present (Supplementary Figure 1e and 3b), however *Sxca*+ cells were indeed present (updated Supplementary Figure 3b). Please find the relevant data in updated Supplementary Fig 1e. According to your recommendation we changed the text:

“Runx2b-deficient embryos exhibited no Xirp2a+ tendon cells with scattered muscle tips (Fig. 2e), and the shortage in tendon formation was confirmed by a loss of the Tnmd expression pattern at 72 hpf (Supplementary Fig. 1e). Thus, severity of cartilage perturbations correlates with phenotype severity of both myocytes and tenocytes, but a hierarchy of how each cell population contributed to the phenotype was still unclear.”

- Line 416-417: Where are *Igtb1a* and *Itgb1b* normally expressed in zebrafish – are they specifically found at the muscle-tendon boundary or tendon-cartilage boundary?

ITGB1 integrins are expressed broadly in the craniofacial tissues in the developing zebrafish, on the side of muscles, cartilage, and tendon (Wang et al 2014, Gene Expression Patterns, <https://doi.org/10.1016/j.gexp.2014.10.001>). We have fixed the text with this clarification.

Also, our new stainings in Supplementary figure 7a shows the expression pattern now, at least for the AM muscle.

- Figure 7d: there is a typo- “Phase portraits...” rather than “Phase protraits...”

This has been resolved, thanks for the keen eye!

- Figure 7h: Although the colors are representative of different clusters, cluster numbers overlaid as text would be helpful to orient the reader to the co-embedded plots.

This has also been improved in the latest Figure 8. Now it should be clearer what cluster is what. Thanks for the good idea!

Reviewer #3 (Remarks to the Author):

In this study, Sunadome and co-workers have investigated how the directionality of developing skeletal muscles is set by mechanical forces. The authors made use of many different approaches and multiple species in this study. To start with, they showed that muscle and cartilage development is synchronized, and cartilage depletion could lead to a dysregulation of muscle patterning. Using live imaging and laser ablation techniques along with genetic tools, they suggest that developing cartilage influences individual muscle cell polarization at an early stage and the directionality of subsequent fusion events. They also indicate that the non-canonical PCP pathway is not involved in direct control of facial muscle directionality in zebrafish.

The author also described the effects of mechanical stretch on muscle cell orientation in 2D and 3D cultures and concluded that mechanical tension is sufficient for directionality, elongation, and controlled fusion of individual muscle cells. They further demonstrated that the JNK pathway is required for muscle cell polarization. To understand if muscle cells require a well-polarized state as a condition to complete muscle differentiation, single-cell RNA-seq time-course experiments were done on cells that are cultured in a soft or hard mechanical environment. They were unable to conclude that muscle cell polarization is necessary for differentiation.

Overall, this is an elegant study and the authors have used a wide range of techniques to demonstrate that mechanical forces are important for the polarization of muscle cells, and subsequently the directionality of developing skeletal muscle. While the *in vivo* and the *in vitro* data convincingly suggest that the directionality of skeletal muscle requires mechanical forces, the last part of this study regarding the muscle cell polarization on soft and hard substrates is confusing. First, at least in the case of hard substrates, there are many myotubes formed. It is unclear whether the authors have included these cells in their scRNA-seq analysis. Were they sequenced at all, or only mononuclear cells were included? I am also confused with the experimental design and the cell cluster labels. How many time points are collected? Is there a time point before seeding? The authors need to provide more rationale/explanation for this experiment.

Major points:

1. Fig1g, as Runx2-deficient mutant have distorted myofibrils, is deficient of tenocytes resulting from the loss of cartilages or distorted myofibrils? Do the migration or specification of tenocytes affected, or tenocytes die?

Please see the new data about the effect of Runx2-ablation on tenocyte marker expression in fish, we have added this to Supplementary Figure 3b. Scxa expression is maintained in the Runx2-deficient embryos, suggesting that specification of tenocytes proceeds normally. Tnmd is not expressed, suggesting that further differentiation is somehow halted. This is reminiscent of a different study in which cartilage and muscle were shown to both play an important role in the development of tendon and ligament progenitor cells, as knockdown of myoD or myf5 inhibited

specification of *xirp2a* in the face and trunk, and inhibited the maintenance of *scxa* in the face (<https://dev.biologists.org/content/141/10/2035>).

Because of the seriously interconnected nature of these developmental systems, and the severe phenotype of the *Runx2*, it is difficult to understand exactly why the tenocytes are disrupted in the *Runx2* fish. However the admittedly less severe phenotype in *Sox9a* (*tw37*) mutant embryos all show relatively normal tenocyte patterning (as shown by HCR in situ hybridization for *Tnmd*) despite substantial cartilage ablation, shown in Supplementary figure 1. The overall patterning of the muscle in these *Sox9a tw37* mutants appears not disrupted although the growth, polarity, and organization of myofibrils is affected. Thus distorted myofibrils is not likely to explain a loss of tenocytes in *Runx2* morphant zebrafish embryos, rather the loss of cartilage might explain it.

2. Fig4a, can the author provide further explanation on the rationale of using the distance between the tip of the myocyte and the ablated site to indicate a mechanical tension in an individual myocyte?

We have updated the discussion to reflect our rationale in this experiment. See below:

“One way to reveal the existence of tension in biological systems is through laser ablation. Stable attachments may resist external forces, creating tension in a material. When tension is applied to viscoelastic materials, they undergo deformation, meaning that the material would deform according to the applied force, but tend to recoil to its original shape when relieved of the force. Laser ablation studies are used with the assumption that by disconnecting the tissue from either the applied force or the normal force, the tissue is at once in a state of disequilibrium where the deformation is no longer actively maintained. If the material immediately moves away from the ablation site, it can be reasonably concluded that the system was under tension.”

We ablated the attachment point between myocytes and cartilage (Fig. 4d, the region encompassed with bracket) with 2-photon laser, and observed the response with live-imaging. After cutting, the tip of myocyte appears to detach and drift away from the ablated sites along the axis of the muscle tissue (Fig. 4d, Supplementary Video 11). We measured the distance between the tip of the myocyte and the ablated site, and found that the recoil velocity decreased over time (Fig. 4d, right panel), indicating a viscoelastic response that depends on the tension and friction-like resistance in muscle primordia. Importantly, this model assumes no active cell migration, but rather that external forces cause the cells to move after they have been detached from the cartilage by laser ablation.

3. Line354, it is imprecise to claim that the reduced number of nuclei indicate influenced muscle cell fusion. It might also due to reduced migration of myocytes or reduced proliferation.

Duly noted. We have changed the text accordingly, see below.

“Importantly, both blebbistatin-treated and Titin-crispant embryos had reduced numbers of nuclei per myofiber, suggesting that these molecules influence myocyte fusion, either directly or indirectly via defects in cell proliferation/migration.”

4. Why the length of myofibrils in Gd3+ treated embryo seems unchanged?

We wrote this explanation in the first version of the manuscript:

“Application of Gd3+ to developing fish larvae or to C2C12 cells grown on stretchable membrane did not affect muscle orientation per se, but rather influenced the number of nuclei within the resulting myofibrils. This suggests that Gd3+- sensitive ion channels participate in myocyte fusion events and further oriented growth of myofibrils, but not in the initial polarization.”

We acknowledge that this explanation doesn't explain everything about the role of Gd3+ because specifically inhibiting myocyte fusion events did affect cytoskeletal organization of the myocytes. However, that experiment with Jamb MO had much fewer nuclei than did the Gd3+ embryos offering a possible explanation that the relatively more severe fusion effect was above the critical threshold to see an interaction between fusion and myocyte length whereas a less severe fusion phenotype was sub-threshold.

As to why Gd3+ treated embryos don't have shorter muscles, there may be different ion channels (not Piezo or TRP) that participate in mechanotransduction relevant to myocyte polarity. Unfortunately we were not able to uncover the entire mechanism of the Gd3+ treatment, as it would require a completely different project out of the current scope.

5. Fig7c, why C2 myoblasts have myofibroblast-like signatures? What is this cluster of cells?

We agree with this question and we added additional marker genes to the main figure (see the new annotations in Figure 8e) supplementary figures to help identify these cell types and explain their identity (see the new markers in Supplementary Figure 10a). Since the first submission we found some papers that characterized our cell population by single cell transcriptomics and found relevant markers to get a more accurate picture of the different clusters. Actually, the myofibroblasts should express PDGFRA/PDGFRB and these cells are found on cluster 6/7. The cells that we previously labeled myofibroblast-like cells are in fact more similar to the mononuclear cell population found in previous transcriptomics studies (De Micheli et al., 2020; McKellar et al., 2021; Zeng et al., 2016). We feel that the current annotations are more robust.

6. Fig7, Transcriptome changes may due to the effect of impaired differentiation. A control, i.e., undifferentiated cells cultured on the hard substrate, is needed to strengthen the claim.

To respond to this comment, we greatly improved our transcriptomic analysis of the hard/soft cultured cells to increase the sensitivity and robustness. Unexpectedly to us, we found that hard/soft substrates seem to have little effect on gene expression during cell differentiation.

The new analysis of bulk RNA sequencing and single cell transcriptomic analysis of these samples agreed that transcriptional differences between hard and soft matrix-cultured cells were rather quantitative (not qualitative). Thus, we acknowledge that there is likely feedback between cell polarity, cell morphology, cytoskeletal organization, and resulting gene expression status.

However, during the revision we made the finding in Figure 8k-m that 1. temporal variation accounted for 96% of gene expression differences in bulk RNA data (whereas hard/soft condition only made up 2% of the gene expression across samples when analyzed in a PCA), combined with our finding that 2. Overrepresentation of hard/soft cultured cells was minor across most cell clusters, particularly the most mature myocyte clusters (there was no cluster or fate with an identity specific to the hard condition as we expected in both bulk and single cell analysis). We interpret this result with caution, and as an evidence that the effect of mechanical forces on muscle cells might rely less on differences in gene expression to drive cell polarity. One possible explanation for this result is that the effect of mechanical forces on cell polarity and morphology might occur mostly at the protein and cell architecture levels. The section regarding transcriptomic analyses has been thoroughly modified to include these new sentiments.

Minor points:

1. Fig3a and Sup video 5, the pseudo-color of *fli1*:GFP is purple while *tbx1*:Cre(Dsred) is green, which can be confusing.

We apologize for this potentially confusing aspect of the presentation. However, we feel that because the main focus is on the muscle morphogenesis, understanding the data would actually be easier for the reader if the color is brighter for the muscle reporter than compared with the cartilage reporter. Thus, we chose this color scheme for most the paper, where the green muscle appears to be foreground and purple cartilage appears more in the background. If the reviewer insists, we would be happy to change it.

Reviewer #4 (Remarks to the Author):

In their paper Directionality of developing skeletal muscles is set by mechanical forces, Sunadome et al provide a very comprehensive and novel study of the factors driving myoblast polarization and myofibers/myofibrils directionality. The authors used several state of the art techniques (live imaging, in situ HCR, single-cell RNAseq, velocity, stretch experiments, microCT) to confirm their hypothesis that skeletal fibers directionality was not a mere function of chemical cues, but also depended on mechanical forces via early cartilage interaction during embryogenesis. They provide convincing approaches and results, combining experiments in C2C12 cells, zebrafish embryos, and murine embryos. They used control, morphant, and mutant embryos, as well as pharmacological inhibitors to prove their points. They added compelling evidence using time-lapse videos, showing the importance of the synchronization between early muscle and early cartilage development.

In particular experiments involving disruption of cartilage development either using CRISPR/Cas9 or morpholinos of several key cartilage markers highlighted the downstream effects on myofiber polarization and fusion. They ruled out a direct control of the WNT/PCP pathway in the control of muscle directionality, and confirmed the importance of mechanical tension/stretching using experiments in embryos and cell cultures. They also showed the role played by the extracellular matrix in providing directionality cues. Overall the strength of this paper is the cutting-edge techniques used throughout the study to validate their points. This is a great very well written piece of work. The authors provided many movies that are really helpful for the community. The only drawback is the single cell experiment (Fig. 7) which does not appear as convincing as presented. The authors should properly address the following concerns.

Major comments:

1- Missing sections. The authors should provide a Statistics section as specified in the authors' guidelines, and be more specific on the exact n values used for some of the experiments (lines 987-988 'of more than 5 embryos'; lines 1067-1068 'Values are means of more than 4 embryos'). The authors also should add an Authors' contribution section.

We added the statistics section and according to new guidelines we have included excel spreadsheet with all quantitative information necessary to recreate the charts.

2- For the C2C12 2D/3D experiments, what is the relation between 20% continuous stretch (fig. 5) and the soft/hard surfaces (fig. 7a). To how many kPa does 20% corresponds to?

We are sorry that we cannot expand much on the relationship between these different experimental setups because tension coming from a single pair of poles can be measured in units of force (Newtons) rather than in units of pressure (kPa). Secondly, the pressure unit used to describe the hard/soft substrates (kPa) is not an active force being applied to the cells, rather a description of the material properties of the substrate's elastic deformation in response to externally applied

forces, for instance via cell-generated tension. A second difference that makes these almost incomparable is the vector is in a different direction: whereas 20% stretch is directed between the poles, hard/soft resistance to cell-derived forces is dependent on the direction of those forces, which are not defined.

3- Fig. 2d. Why was a single control animal used for the quantification? It is hard to compare with the mutant ones, assess significance, and impossible to evaluate inter-animal variability.

We agree that a single control was too low. In the revised manuscript, we have used more control animals for the quantification and we found that in fact, volume is not significantly decreased in muscle, (although a paired comparison might appear this way) but rather that the length of the muscle is definitely significantly decreased, which suggests a lack of polarization in the muscles of Collagen II CreERT2 R26 DTA, cartilage ablated embryos. Please see the updated data in Figure 3.

4- Fig. 3b: The runx2b morpholino really seems to affect the polarization and development of the muscle cells. However, for the sox9a+9b CRISPR mutant, it is difficult to say if the process is only delayed in time, or if it will not progress further? Did the author look at a later time point for comparison, for example ~72hpf to see if the mutant caught up with the control?

By 72hpf, the mutant does not catch up with the control. During the revision, we checked at 72 hours and found a similarly severe phenotype in the Sox9a mutant stable line, one that very closely resembles the phenotype of the Sox9a/Sox9b crispant embryos at the earlier timepoints (Supplementary figure 1d). Apparently, there are lasting effects of Sox9a ablation on the cartilage in the face. Also, we checked the Sox9a+9b crispant embryos at 120 hpf during the revision to confirm the long-lasting nature of the phenotype. The new 120 hpf data is included in Supplementary figure 1c.

5- Fig. 3g: The jamb morpholino clearly affected fusion of the muscle cells. However, could the authors explain/comment on what are all the small tiny fluorescent dots in the jam MO vs control? It seems that myofiber length is not affected per se (definitely not statistically significant), but that there is a structural issue in these embryos.

We are not sure what caused the punctate signal in these embryos, but we included more data in Figure 5d (Phalloidin staining and Col2a1:mCherry) to show that no structural issue exists in the jamb MO embryos at 72hpf.

6- Fig. 4e: the legend is unclear and the labeling on the panel is not clear either. Not sure which panel corresponds to control vs titin CRISPR mutant. This should be modified and clarified in the figure legend.

Done

7- Fig. 5: This figure has many sub-panels, specifically for panel d and e. It would help if the authors could label the sub-panel, maybe as i, ii, iii, iv, etc... It would also simplify the description in the figure legend.

We agree and we think this is a good idea. We further subdivided labels in this figure, thanks for the suggestion!

Panel d (right part) of this figure is unclear. The authors specified that they show the 2 pillars case without stretch due to fragility of the tissues. It seems however from their schematics that the cell orientation is not affected by the absence of stretch? Or would they expect an even better alignment with stretch? It seems the zoomed boxes from this panel were not chosen randomly. **What would a zoom of the middle region show (like done in window 2 from the 3-pillar below)? Here having larger pictures would really benefit.**

We tried to increase the size of pictures in this picture for the benefit of the reader's eyes.

Given the two attachment points, cells in the 2-pillar case are able to generate their own tension via cell contractility, mimicking the effects of stretch. That is why the tissue is polarized even without external stretch. However, tissue width is slightly reduced by increased tension due to ectopic applied stretch, perhaps via a small increase in cell orientation.

Regarding the comparison of 2- and 3- pillar: The zoomed boxes in the panel are not chosen randomly, this is true. We first chose ROI close to the pillar to compare with ROI relatively far from the pillar. Our original premise was that in the 2-pillar case, orientation is uniform throughout the tissue, whereas in 3- and 4- pillar samples, the cell-generated tension has multiple directional vectors, resulting in a less polarized tissue in the center, whereas ROI close to the pillar are still polarized. But, this reviewer made us realize that distance from the tissue edge also seems to play a role, as in all cases, the cells near the edge are polarized while in the center they are not. The difference is that the 3-pillar and 4-pillar tissues are much thicker, thereby have more central, unpolarized region, whereas the 2-pillar is thin, making it so that most central cells are also near the tissue edge. We reason that the directionality of cell-generated tension, combined with the tissue shape, together are responsible for this phenomenon. We updated the text to present this more nuanced view, and we thank the reviewer for helping us improve the conclusion.

“Similar to the in vivo situation with natural muscle attachment points, the pillar positions dictated the directionality of coalescing muscle tissue. We compared artificial tissues seeded with single (Fig. 6d), dual (Fig. 6e), and multiple attachment points (Fig. 6f). When we provided two attachment points, the individual myocytes were aligned along the pillar-to-pillar axis, and they formed a polarized and highly aligned muscle tissue, expressing alpha-actinin, a muscle-specific structural protein (Fig. 6g, boxes 1 and 2). However, cell polarity was more uniform along the edges than in the very center, suggesting that the tissue edge might also influence cell polarity (Fig. 6g, compare boxes 2 and 3). When myoblasts were seeded without a second attachment point, cells attached to the single pillar also expressed alpha-actinin (Fig. 6h), suggesting that attachment to a stiff object can induce differentiation. However, individual myotubes did not show directional consistency, and the overall tissue structure ended up in un-polarized aggregates (Fig

6h). This result confirms that mechanical tension in a muscle cluster between two attachment points is sufficient to guide cell orientation.

Next, we looked into the cases of 3 or 4 attachment points (Fig. 6i and j). In these cases, myotubes near the pillar align following the axis pointing to the nearest pillar (Fig. 6i see windows 1 and 3, and Fig. 6j, see window 1). Where most cells in the 2-pillar condition are either near a pillar or the tissue edge, 3-pillar and 4-pillar conditions feature large central regions of cells, distant from any directional input. Many cells positioned in these central areas of the tissue displayed random orientation, suggesting that increasing distance from attachment points, or distance from the tissue edges, reduces the effectiveness directional cues for coherent cell orientation (Fig. 6i and 6j, inset windows 2). In these artificial 3D muscle tissues, polarization and orientation of individual muscle cells is entirely self-organized and only guided by the attachment points and the ensuing cell-generated mechanical tension that appears to primarily build along the tissue edges. These results show that mechanical tension is sufficient to drive directionality, elongation, and controlled fusion of individual muscle cells, but that this can be potentiated by muscle tissue shape.”

8- Fig. 6: Can the authors develop on the potential effects of SP600125 on cell fusion? It seems clear that with blebbistatin fusion is affected (also shown in fig. 4e). But it is unclear for SP600125. Is the effect only on polarization or also on fusion? Panel d of this figure does not seem to add much. Also the figure legend is missing the description of panel d.

JNK inhibition with SP600125 does affect fusion in vitro, giving rise to mostly single-nuclei cells but also a few highly fused cell clusters, similar to the distribution of fusion in the soft-cultured cells. Distribution of cell fusion in a control sample is shown in Fig 8b, showing that normal cultures of myocytes fuse extensively, usually resulting in at least 5-20 nuclei per tube, whereas from the JNK figure (now Figure 7), it seems most treated cells have only 1-3 nuclei.

We fixed the issue with the legend.

9- For the scRNAseq experiments, the authors collected C2C12 cells over 2 day prior and over 3 days after differentiation. Although myoblasts (cells prior differentiation) are ‘regular’ cells to be analyzed by single cell assays (shape, mononuclear), how did the authors proceed for differentiated myotubes that can be really long and polynucleated, especially from the hard substrate condition and may not have passed through filtering step or through the microfluidic? It seems from the methods that the authors performed single cell and not single nuclei assays. How were the cells/myotubes selected or filtered to go through the 10x Genomics chip and protocol? The chip size let cells with a size less than 40um go through. Bigger sizes would lead to clogging of the microfluidics. If filtering, isn’t there a great risk of bias toward undifferentiated and thus smaller cells? Indeed the differences between the hard and soft surfaces would have been expected greater, especially based on Fig. 7a, where the 0.5kPa treated cells look very different from the mostly all differentiated 100kPa myotubes. A few smaller myotubes/myocytes may have gone through the analysis as shown from panel 7h (bottom left corner). I think this graph comparison panel in 7h is the important part to highlight as it shows more than some of the other panels in Fig. 7.

After receiving your comments about a potential bias due to cell filtering, we decided to run more tests to learn how much bias we might be introducing. We compared our single cell analysis to a replicate experiment analyzed instead by bulk RNA analysis (which should capture all myotubes), and the differences between hard/soft were similarly subtle in both cases. We also compared our single cell analysis and bulk analysis to previous single-cell transcriptomic studies and single-nucleus studies that also examine myocyte differentiation, some of which use the C2C12 cell line, and found the gene expression signatures in our data is comparable to both single-cell and single-nucleus sequencing of these cell lines, therefore we think that our cell dissociation protocol probably includes myotubes.

De Micheli et al., 2020, Cell Reports 30, 3583–3595

McKellar, D.W., Walter, L.D., Song, L.T. *et al. Commun Biol* 4, 1280 (2021).

Zeng W, Jiang S, Kong X, et al. *Nucleic Acids Res.* 2016;44(21):e158.

We were equally puzzled by the apparently small difference between hard and soft-cultured cells, which is why we were trying to add as much computational analysis as possible to reveal any subtle difference was there. But after looking at the new results with bulk RNA sequencing it is clear that the differences between hard and soft-cultured cells isn't strongly manifested by the transcriptome despite a morphological transformation. See text from the revised manuscript below:

“Because hard-cultured cells form myotubes that may hinder creating single cell suspension for transcriptomics, we also sequenced bulk RNA from hard-cultured versus soft-cultured cells after zero or three days of induced differentiation and compared it to our single cell transcriptomics data. To this end we used two different approaches: DESeq, for understanding the major causes of variability across the four types of samples (D0/D3 x Hard/Soft), and RNA-sieve, a tool for finding representation of single cell clusters within bulk RNA datasets. DESeq analysis revealed that many genes differed when comparing hard versus soft conditions (Fig. 8l, Supplementary Table 2), including 1267 and 1516 genes upregulated in soft and hard in bulk, respectively. Of these bulk differentially expressed genes, 548 and 475 of these genes were also differentially regulated specifically in clusters overrepresented by soft and hard cultured cells, respectively. However, despite these measurable differences in gene expression, hard/soft culture conditions accounted for less than 2% of variability across samples, whereas 96% of variability could be explained by time in culture. In other words, samples at day 3 were transcriptionally very similar, regardless of whether they were cultured on hard versus soft substrates (despite radically different cell morphology). We analyzed genes correlated with time in culture (Figure 8m, Supplementary Table 2) of which the significantly differentially expressed genes could be used to create aggregate gene modules associated with initial or final timepoints. When cells from the single cell dataset were plotted, comparing RNA velocity latent time against the activity of time-associated gene modules, we could confirm the general prediction that over time, myocytes were in fact differentiating irrespective of culture condition (Fig. 8m). Further, RNA-sieve deconvolution of hard/soft cluster representation in Bulk RNA basically matched that of our single-cell cluster composition analysis, and showed similarly increased myocyte (C9) associated gene expression over time in both hard/soft samples (Fig. 8n), arguing against artefactual dropouts of differentiated cells in our single cell datasets. This is in agreement with studies of single-nuclei transcriptomics in the same cell line.”

10- Related to point #9 above, it would be useful for the interpretation of the results if the authors

could show the individual 5 data points corresponding to the individual days of cell collection, as well as the individual UMI counts for all samples, rather than a pooled UMI UMAP, at least as supplementary figures.

The single cell analysis used cells that were pooled from the different days, but our bulk RNA analysis was separated by both time and condition. This allows us to show which days represent which clusters by comparing the marker genes present. **See the above blurb of text from the revised manuscript discussing the temporal effects on gene expression.**

11- For fig.7: It seems unclear if all the cells from individual days were pooled together before running the 10x assay, or at the analysis stage. This should be added in the method, as it is difficult otherwise to interpret the results. If the cells were pooled at the analysis stage (after running individual 10x Genomics assays on individual samples collected each day), then it should be easy to show the individual stages and differences between the soft/hard substrates. If the pooling occurred before running the scRNAseq assay, the samples may have been filtered and there may be a bias on which cells were indeed analyzed (potentially mostly undifferentiated cells). Cells from day 3 in the hard vs soft conditions would be expected to show very different profiles. It seems very strange that only a small proportion of the cells (mainly from cluster 4) is differentiating the hard vs soft surfaces grown cells, as well as an additional myoblast cluster in the hard substrate whereas the general profile is very similar. Another difference is for bridge 2 where there is high velocity on the soft substrate, compared to almost none in the hard substrate condition. From the UMI plots in supplementary fig. 6, it does not appear like some cells have much more UMI and thus are larger which would have corresponded to myotubes (longer + multinucleated) or that the proliferating cells (highlighted on the adjacent supplementary panel) have much more UMI. This is surprising and should be better explained by the authors.

The cells from different days were pooled together before running the 10X assay. We added this to the description in the text. The scenario hypothesized by the reviewer implies that there should be striking transcriptional differences between hard and soft/cultured cells. We also performed the experiment under this paradigm and were shocked to find that when analyzing bulk RNA sequencing, hard/soft culture conditions only accounted for 2% of all transcriptional variability, whereas time accounted for 96% of transcriptional variability across the samples analyzed (Day 0 hard, Day 0 soft, Day 3 hard, Day 3 soft). Furthermore, the genes upregulated in Day 3 for both samples were found to be enriched in the most differentiated cluster (what we are calling myotubes), whereas the undifferentiated myoblast clusters express genes found on Day 0. The intermediate clusters, not accounted for by Day 0 or Day 3, must represent the intermediate days. To us it suggests that the hard and soft conditions are transcriptionally very similar, and that our single cell analysis contains all the appropriate cell types including actual myotubes. There is a small probability that myotubes are actually excluded from the analysis, but if it is true, at least cells with a highly similar transcriptional profile got analyzed.

12- Fig. 7: panel d is unclear and not well explained in the legend. Better when looking at the supplementary figures, but still should be better shown/explained in the main text and main figures. I may be missing something, but I am not sure what this panel 7d is adding. There does not seem

to be much differences between the 2 conditions, which again point toward a bias in the cells that went through the 10x Genomics chip, probably selected toward the smaller ones. Panels f-h could be combined. I think panel f (left) and panel h (left) should be shown together. Panels f (middle and right) are not adding anything not show already in panel f (left). The legend from panel g is really unclear and should be better explained. It says GO terms relative to cells in cluster 4, but this is showing all clusters, and the enrichment are not all for cluster 4 (only 3 out of 6 sub-panels).

We agree the velocity profiles don't add much and it has been moved to supplementary information.

13- Do the hard vs soft substrates both eventually lead to the cell fusion, or does fusion never occur on the soft substrate, even with longer times in culture?

Fusion occurs to a limited degree in the soft culture, but nowhere near as much as in the hard culture. After 5 days in soft culture the cells appeared broadly unfused (see new analysis in Fig. 8b).

14- Suppl. Fig.6: there are no a, b, c, or d in the figures, although there are in the legend.

- Thanks, we fixed this.

Minor comments:

1- Some sections of the main text could go in the Methods. For example, lines 384-387, or lines 487-491.

We placed the sentence from lines 487-491 in Methods. Thanks for the suggestion. As for the other sentence, we tried moving it but felt the narrative became a bit choppy as a result, so we removed some of the technical details from the narrative and kept the full sentence in Methods.

2- Line 589: the authors should be more specific than 'over many hours'.

Done

3- A few typos, missing bold, language errors, in particular of the Figures legend section. Some examples, but the authors should review the figure legends:

Line 1028: A should not be capitalized.

Done

Line 1040: remove 'were' from 'embryos were served as control'

Done

Line 1044: Instead of '... was ablated by laser pulse in the embryo from the outcross...', prefer 'was ablated by a laser pulse from an outcross of Tg... and Tg... embryo'.

Done

Line 1073: change 'The device of 2D cell-stretcher', prefer 'The 2D cell-stretcher device'.

Done

Line 1082: change 'the area of dashed box', prefer 'the dashed box area'

Done

Line 1092: remove total from 'About total 200 myotubes'

Done

Line 1097: change 'measured for total 200 myotubes' to 'measured for a total of 200 myotubes'.

Done

Line 1110: C should not be capitalized

Fixed

Lines 1114 /1120: choose between single-cell or single cell, but keep consistency in the manuscript

Fixed, we chose the hyphen.

Line 1142: 'showing expression of example genes', maybe replace with 'showing some examples of expression of key genes'.

Fixed

Line 1146: may want to change 'we would like to appreciate Dr... for his idea' with 'We appreciate the contribution of Dr...'

Fixed

In the legend of Suppl. Fig. 6: c and d should be bolded.

4- Fig.2b, the authors should add a box of the zoomed area as it is unclear from the legend which area is actually shown as a blow-up of the top panel. There are 2 dark blue arrows on the top panel, and so which one corresponds to the area shown on the bottom panel?

We changed the colors of the arrows for better clarity. Dark blue arrow now definitely corresponds to the image in the zoomed area in 2b.

5- Fig. 2a, fig. 2b, and fig. 2g: the authors should use colors that distinguish better the arrowheads. Also they should specify what the different colors mean in the figure legend as this is missing.

Done

6- For many of the zoomed figures and some other figures, the resolution is too low and should be increased. It is hard to see (figs 3f, 4b, 5c, 6c). Some panels look very tiny and would benefit from being larger with a larger area shown, see for examples fig. 1f,g; fig. 5.

We are sorry that some of our images had low resolution, but for some pictures that we were imaging in whole mount, it was impossible for us to increase due to technical limitations. In case of Figure 5 (now Figure 6) we increased the size of the panels.

7- Supplementary figs 5-9 are not labeled as such. Suppl fig. 8 is on 2 pages.

We were previously unable to fit Supplementary fig. 8 on 1 page while keeping the panels readable. We would like to include it as a supplementary data file instead, but this time we added it as a very large figure.

8- The authors may want to follow the standard zebrafish orientation on figures, with the head on the left, tail on the right, dorsal on top, ventral on the bottom. Some of the pictures do not respect the convention.

This issue pained us during the original data gathering stages of the project because during live imaging experiments, to see the muscle of interest required us to use an unorthodox angle (45 degrees between ventral and side view). As for the analysis of musculature, especially muscles which are present at an angle if viewed from the conventional perspective, it was more efficient to present them as vertical to save space in figures which are already full of data. We hope that with the additional information in the revised panels, the images that break convention do not confuse other readers from the zebrafish community.

Reviewers' Comments:

Reviewer #1:

Remarks to the Author:

We would like to commend the authors for their excellent efforts in addressing the concerns raised in the previous round. The paper is reasonably well written now and clear to follow. The additional experiments performed have significantly improved the quality of the paper, and strengthened the arguments made. We only had minor comments that we think need to be addressed before this is accepted for publications:

Panel numbering within each figure needs to be improved. It is recommended to give a different letter to each panel/graph so that the reader can clearly refer to which image is being referred to in the text. For example, in Figure 2a, there are 4 panels and when reading the text, it is not very clear which one is being referred to. These could easily be broken down into 2A, B, C and D. Please confer this for all figures.

"Both Sox9 and Runx2b perturbation resulted in fewer numbers of nuclei per myofiber, suggesting that the cell-cell fusion was partly inhibited". Which is the DsRed and Hoechst staining on a select few cells (Figure 2A)?

Figure 2A - Please include n number for the graphs.

"This observation is consistent with the fact that the muscle of Sox9-deficient mutant has a higher number of thin myofibrils as compared to controls". Where is the evidence for this? None of those graphs represent the muscle cell number.

Throughout the manuscript please carefully correct the words "mutant" and "crispants". For example, "Lamc1-mutant embryo" should be Lamc1 CRISPR edited larvae (or crispants). Given that these were CRISPR injected larvae, you should be referring to them as Crispants and mutants.

Reviewer #2:

Remarks to the Author:

The authors have provided a greatly revised manuscript describing the role of cartilage growth on muscle cell polarization and morphology. Although much of the data supports the role for cartilage and mechanical forces on muscle phenotypes, there are some concerns about specific conclusions that are made in the manuscript as outlined below.

The authors show new data on tendon development analyzing the timing of Scx and Tnmd expression, knockdown of tendon-associated matrix genes, and analysis of these markers in runx2b MO-injected or sox9a/b CRISPR gRNA-injected mutants. Based on their new data, the authors conclude that "Tendons are not required for myocyte polarization". This broad interpretation of the data is problematic for the reasons mentioned below and the authors should consider being more specific and instead stating that "mature tendons are not required" as is mentioned in some places of the text.

One issue is technical in that Scx expression has been reported at earlier stages in both zebrafish and mouse in contrast to the results presented in this paper. For instance, Pryce et al., 2007 show Scx transcripts by in situ hybridization and Scx-AP/Scx-GFP expression in the branchial arches at E9.5 and E10.5 in the mouse (Fig 2). Chen and Galloway report scxa transcripts in the pharyngeal arches at 40 hpf. Although staining methods can vary to some extent, it is surprising that expression is not detected at similar stages, especially when there is a discrepancy of 3 days in the mouse (between the authors' reported expression at E12.5 and the previous publications at E9.5). The zebrafish embryo

shown for *scxa* expression looks significantly delayed for 48 hours with very small eye size.

The main conceptual issue is the interpretation that the presence of *scxa*⁺ tendon cells in the cartilage mutants indicates that they have no role in the muscle phenotype. The authors show that *scxa* and *col1a2*⁺ cells are present in *runx2b* MO-injected embryos and *Sox9* gRNA-injected mutants. Based on this, they conclude that the muscular phenotype results from cartilage perturbations rather than effects on tenocytes. It is clear that *scxa*⁺ tendon cells are present in the mutant/morphant embryos, but the pattern of *scxa* expression is severely distorted (broadened and shortened in some regions). Furthermore, *col1a2* in the *runx2b* deficient embryos does not appear specific to tendon and looks diffusely staining the head so this gene should not be used in this mutant to signify the presence of tendon cells. It seems possible that the abnormal pattern of *scxa* expressing cells could result in abnormalities in how the muscle attaches. In addition, the cell ablation experiments could in theory be targeting tendon cells that are attaching to muscle (Fig 4d). In support of this interpretation, previous studies have implicated neural crest-derived connective tissue in muscle attachment phenotypes. For example, Woronowicz et al., 2018 through chick-duck transplantation experiments show that muscle insertion patterning is dictated by the amount and spatial distribution of Neural Crest Mesenchyme-derived connective tissues. This seems to leave open the possibility that the altered pattern of *scxa*⁺ cells could have some contribution to abnormalities in the muscle. Particularly this altered geometry may influence how muscle may be experiencing force from the cartilage (size/pattern of the attachment).

Another issue deals with the experiments to show that there is no muscle phenotype upon gRNA-mediated loss of tendon-matrix genes. Loss of the target genes is not shown for these experiments so the extent of protein loss in the Crispr/gRNA-injected embryos is unclear. Without understanding the extent of knock-down efficiency, it is difficult to conclude that these genes and thereby tendon matrix is not required. It is also possible that the interactions between early forming tendon and muscle are different (in that they utilize different molecules) than the matrix genes involved in making a mature tendon.

However, it is an agreed point that the timing of expression of the 'mature' tendon genes (*tnmd* for example) occurs after the muscle phenotype is observed. Therefore, it seems plausible that mature tendons are not required for the muscle phenotype (as stated in some parts of the section), but that the title of the section should be changed to reflect this more modest interpretation to "Mature tendons are not required for myocyte polarization".

The authors conclude that Wnt is not involved in facial muscle directionality. The authors use gRNAs to disrupt *Wnt11*, *Wnt5a*, and *Wnt5b* expression but do not show any efficacy of knockdown (other than the cartilage defects). Moreover, it should be noted that Wnt regulation of muscle formation is different between cranial and axial regions (Tzahor et al., 2003). Therefore, later muscle polarization events may also be regulated differently as the Gros et al., 2009 study focused on axial muscle. Also, it is unclear how Wnt ligands from the expressing cells reach the C2C12 cells (Sup Fig 5d). Since the Wnt producing and receiving cells do not appear to be directly touching each other and Wnt can act as a juxtacrine signal, is Wnt diffusing throughout the culture dish? Is a gradient formed? It seems this experimental set-up is very different from the transplantation of *Wnt11* expressing cells done in embryos in Gros et al., 2009.

Could the authors comments on why *Scx* and *Acta2a* are enriched in the myocyte populations in the single cell RNA-seq experiments?

Sup fig 13 is difficult to read and appears very pixelated.

Reviewer #3:

Remarks to the Author:

In this study, Sunadome and co-workers used various approaches, including genetics, chemical perturbations, and imaging, to show that cartilage supports directional muscle growth. They provide evidence indicating that this orientated muscle growth is related to the tension and mediated at least from the ECM rather than fusion. They have beautiful live-cell imaging showing the developing myocytes' nuclei migration and division. The stretch experiment is elegant in showing that myoblast alignment and oriented muscle growth independent of tension but also fusion. Though it is still difficult to attribute the disoriented muscle growth to tension or myoblasts number in some assays (fig5ca, fig5c), it generally provides strong evidence to illustrate the importance of cartilage structure and tension in orientating muscle growth.

Major points:

1. It is great to show multiple approaches to genetics and drug treatment in the paper, and I respect the huge effort that is put in, yet either in situ or staining is necessary to show that the targeted genes/proteins are perturbed.
2. Please address why sox9 and runx2 mutant have different severity.
3. Where is fig4b 39h image? Why is the time point of the control different from mutants? Why are the images' time points (left) and quantifications (right) mis-matched?
4. In Fig 4e, very interesting results and a beautiful assay. How do myocytes sense the new cartilage attachment point? Is the ECM affected after laser ablation? Any cytokines changes?
5. In Fig 5a, there is a need to show that it is not due to the change in myocyte number. Also, it would be good to use other myosin inhibitors to show drug specificity.
6. Fig 5c, are myocytes number reduced in drug-treated fish and crisprant? Without proper control, it is hard to conclude that the myofiber elongation depends on cytoskeletal organization

Minor points:

1. The development time point is missing in Fig1c, 1d, and others.
2. Fig 3d middle graph – where is the t-test result for the volumes?

Reviewer #4:

Remarks to the Author:

This is a revised version of a manuscript that investigate the role of cartilage in regulating the development of striated muscles in zebrafish. The authors answered all my questions in great details and really improved their manuscript. The responses are thoughtful and thorough. I have no concerns.

Reviewer #1 (Remarks to the Author):

We would like to commend the authors for their excellent efforts in addressing the concerns raised in the previous round. The paper is reasonably well written now and clear to follow. The additional experiments performed have significantly improved the quality of the paper, and strengthened the arguments made. We only had minor comments that we think need to be addressed before this is accepted for publications:

Panel numbering within each figure needs to be improved. It is recommended to give a different letter to each panel/graph so that the reader can clearly refer to which image is being referred to in the text. For example, in Figure 2a, there are 4 panels and when reading the text, it is not very clear which one is being referred to. These could easily be broken down into 2A, B, C and D. Please consider this for all figures.

Thank you for the kind words. We agree with the suggestion about labels and we have changed labeling for all main figures and majority of supplementary figures where it was possible to disambiguate them without overcrowding (or causing confusion) throughout the entire manuscript.

"Both Sox9 and Runx2b perturbation resulted in fewer numbers of nuclei per myofiber, suggesting that the cell-cell fusion was partly inhibited". Which is the DsRed and Hoechst staining on a select few cells (Figure 2A)?

We thank the reviewer for this comment. This discussion is related to the revised Figure 2d and 2e. The labeling for DsRed and Hoescht is clarified (legend for staining is based on the color of the text below the image). The quoted claim is based on quantification of 5 embryos per condition, around 5 myofibers per embryo on average, with anywhere from 1 to 6 nuclei being counted per myofiber. We believe that the amount of investigated samples is sufficient for supporting the statement.

The updated text in the figure legend 2e states this information:

*"Muscle cells at 81 hpf were classified according to the number of nuclei and the percentage of muscle cells is shown. On average, 5 different myofibers were counted for nuclei per embryo. Values are means of n=5 embryos \pm SD. *, $P < 0.01$, Fisher's exact test."*

Figure 2A - Please include n number for the graphs.

*Thanks for spotting this. The n values for all quantifications are now included in the accompanying figure legends for these panels (Corresponding to former 2a) . For muscle length and nuclei counts in 2A, the n values are now reported as such in the Figure 2b legend: "The length of individual muscle cells is quantified and shown in box plot. *, P<0.01, unpaired two-tailed Student's t test (n=27, 31, and 24 for ctrl, sox9, and runx2b, respectively)."*

Figure 2c legend:

*"Number of myofibers per muscle were quantified from cross-sectional images of am muscle at 81 hpf and shown in the box plot. *, P<0.01, unpaired two-tailed Student's t test (n=7, 10, and 7 for ctrl, sox9, and runx2b, respectively)."*

Figure 2e legend:

*"Muscle cells at 81 hpf were classified according to the number of nuclei and the percentage of muscle cells is shown. On average, 5 different myofibers were counted for nuclei per embryo. Values are means of n=5 embryos \pm SD. *, P<0.01, Fisher's exact test."*

Raw counts are also provided for each graph as part of the data reporting summary that we submitted alongside the manuscript. Thanks for suggesting a good way to report our findings more transparently.

"This observation is consistent with the fact that the muscle of Sox9-deficient mutant has a higher number of thin myofibrils as compared to controls". Where is the evidence for this? None of those graphs represent the muscle cell number.

In the revised manuscript we returned to the original dataset and performed new quantifications to understand the distribution of myofibers per muscle in the control, Sox9-crispant, and Runx2-MO conditions (Figure 2C). We hope the reviewer agrees that the additional data substantiates the claim.

Throughout the manuscript please carefully correct the words "mutant" and "crispants". For example, "Lamc1-mutant embryo" should be Lamc1 CRISPR edited larvae (or crispants). Given that these were CRISPR injected larvae, you should be referring to them as Crispants and mutants.

We are sorry for not catching all instances in the previous revision and we think it is 100%

fixed now.

Reviewer #2 (Remarks to the Author):

The authors have provided a greatly revised manuscript describing the role of cartilage growth on muscle cell polarization and morphology. Although much of the data supports the role for cartilage and mechanical forces on muscle phenotypes, there are some concerns about specific conclusions that are made in the manuscript as outlined below. The authors show new data on tendon development analyzing the timing of Scx and Tnmd expression, knockdown of tendon-associated matrix genes, and analysis of these markers in runx2b MO-injected or sox9a/b CRISPR gRNA-injected mutants. Based on their new data, the authors conclude that Tendons are not required for myocyte polarization. This broad interpretation of the data is problematic for the reasons mentioned below and the authors should consider being more specific and instead stating that Mature tendons are not required, as is mentioned in some places of the text.

We are thankful for the insights into the potential roles of tenocyte and tenocyte precursors during muscle polarization. We revised the manuscript carefully according to your important comments. The result of the revision yielded more nuanced conclusions and suggested that although mature tenocytes are not required, the role of tenocyte progenitor cells in the process of muscle polarization cannot be ruled out. We clearly state this in the manuscript text now and provide detailed response to all exact comments below.

One issue is technical in that Scx expression has been reported at earlier stages in both zebrafish and mouse in contrast to the results presented in this paper. For instance, Pryce et al., 2007 show Scx transcripts by in situ hybridization and Scx-AP/Scx-GFP expression in the branchial arches at E9.5 and E10.5 in the mouse (Fig 2). Chen and Galloway report scxa transcripts in the pharyngeal arches at 40 hpf. Although staining methods can vary to some extent, it is surprising that expression is not detected at similar stages, especially when there is a discrepancy of 3 days in the mouse (between the authors reported expression at E12.5 and the previous publications at E9.5). The zebrafish embryo shown for scxa expression looks significantly delayed for 48 hours with very small eye size.

Thank you for this comment; based on this comment we got concerned about the sensitivity of our Scx stainings. Therefore we decided to perform additional systematic experiments in a wide range of conditions (fixation, probe concentration, whole mount versus sections, mode

of ISH) to rule out technical issues preventing us from detecting early expression of Scx.

The successful condition that enabled early detection of Scx in E11.5 mouse embryonic face was based on HCR on fresh frozen, unfixed tissue sections and higher probe concentration than the previous supplementary figure 2b. In the revised manuscript (updated Supplementary Fig. 3t-v), we report Scx expression in facial Col2a1+ population and its neighborhood. This observation is consistent with previous report showing Scx is expressed in condensing cartilage of the face (Cserjesi P. et al., 1995 <https://journals.biologists.com/dev/article/121/4/1099/38583/Scleraxis-a-basic-helix-loop-helix-protein-that>, see Fig. 5).

Also, we revisited the HCR experiment in multiple zebrafish embryos at 48 hpf (previous supplementary fig. 3a, right) by making thinner optical section to find faint signal. We found that scxa is somewhat broadly expressed in mesenchymal populations including col2a1+ populations (revised Supplementary Fig. 3c-d).

Therefore, our revised data show that Scx/scxa have broader expression pattern in mesenchyme at early stage in contrast to later stage when their expression is more restricted to mature tendon. We corrected our results in the revised manuscript. It is an open question if/how early Scx/scxa positive cells are involved in muscle polarization, and will be our future work.

The main conceptual issue is the interpretation that the presence of scxa+ tendon cells in the cartilage mutants indicates that they have no role in the muscle phenotype. The authors show that scxa and col1a2+ cells are present in runx2b MO-injected embryos and Sox9 gRNA-injected mutants. Based on this, they conclude that the muscular phenotype results from cartilage perturbations rather than effects on tenocytes. It is clear that scxa+ tendon cells are present in the mutant/morphant embryos, but the pattern of scxa expression is severely distorted (broadened and shortened in some regions). Furthermore, col1a2 in the runx2b deficient embryos does not appear specific to tendon and looks diffusely staining the head so this gene should not be used in this mutant to signify the presence of tendon cells. It seems possible that the abnormal pattern of scxa expressing cells could result in abnormalities in how the muscle attaches. In addition, the cell ablation experiments could in theory be targeting tendon cells that are attaching to muscle (Fig 4d). In support of this interpretation, previous studies have implicated neural crest-derived connective tissue in muscle attachment phenotypes. For example, Woronowicz et al., 2018 through chick-duck

transplantation experiments show that muscle insertion patterning is dictated by the amount and spatial distribution of Neural Crest Mesenchyme-derived connective tissues. This seems to leave open the possibility that the altered pattern of *scxa*⁺ cells could have some contribution to abnormalities in the muscle. Particularly this altered geometry may influence how muscle may be experiencing force from the cartilage (size/pattern of the attachment).

*We agree that *scxa*⁺ cells may affect the attachment between early chondrogenic condensations and myoblasts. In that case, the tendon precursor cells might help coordinate the biomechanical landscape driving polarization. Furthermore, it would therefore be impossible, given the current set of evidences, to rule out a role for these cells in conditions where the gene expression distribution of *scxa* is altered. We modified the text of results as quoted below:*

*“We examined whether tenocyte precursors (expressing *scxa*, a transcriptional regulator of tenocyte cell identity) were present during facial muscle polarization. *Scxa* was absent at 44 hpf (Supplementary Fig. 3a) and visible at 56 hpf (Supplementary Fig. 3b) when using traditional whole mount in situ hybridization. Using RNA hybridization chain reaction technology (HCR, Supplementary Fig. 3c) *scxa* can be found in facial mesenchyme as early as 48 hpf (Supplementary Fig. 3d). We found *MyoD* expression (visible at E11.5) preceded *Scx* expression (visible by E12.5) in mouse embryonic lower jaw when using standard HCR in whole mount (Supplementary Fig 3e-j). When increasing the HCR probe concentration, *Scx* expression domains were obvious in the E11.5 ribs and limb but not in jaw (Supplementary Fig. 3k-s). However, HCR of unfixed, fresh frozen sections revealed low levels of *Scx* expression in the facial mesenchyme (Supplementary Fig 3t-v). Thus, tenocyte precursor cells might be present during muscle polarization in zebrafish and mouse, even if *Scx* might not be exclusively expressed in these cells at early stages.”*

*We mapped the developmental timeline of tendons near the am muscle by assessing gene expression of *scxa*, *tnmd*, and *colla2* (Supplementary Fig. 4a) and which also suggests that tenocyte precursors, but not mature tenocytes, could be present during am muscle polarization. Labeling tenocytes in cartilage-less embryos revealed that both *tnmd* and *scxa* expression patterns were disrupted alongside cartilage growth defects (Supplementary Fig. 4b). It remains unknown to what degree the positioning of tenocyte precursors influences myoblast attachment to chondrogenic condensations, and this open question should be pursued in future work.”*

Another issue deals with the experiments to show that there is no muscle phenotype upon gRNA-mediated loss of tendon-matrix genes. Loss of the target genes is not shown for these experiments so the extent of protein loss in the Crispr/gRNA-injected embryos is unclear. Without understanding the extent of knock-down efficiency, it is difficult to conclude that these genes and thereby tendon matrix is not required. It is also possible that the interactions between early forming tendon and muscle are different (in that they utilize different molecules) than the matrix genes involved in making a mature tendon.

We agree with the reviewer, and we sought to validate the efficiency of experiments that had negative results.

Therefore, in the revised manuscript, to examine the mutagenesis efficiency in tenocyte matrix-targeted crispants, we randomly selected the gRNA-injected embryos and sequenced them. We found that all of the gRNAs used for this analysis are highly effective (Supplementary Data File); all the sequenced embryos were shown to have heavy mutations on the target region, where we could not find a stretch of the wild type sequence. Crispant assay in our study is based on the method reported by Shah et al. (<https://www.ncbi.nlm.nih.gov/pmc/articles/PMC4667794/>) as a knockdown/screening tool. This method depends on gRNA-targeted double-strand DNA breaks and subsequent non-homologous end joining, which often introduces insertions or deletions that can disrupt gene functions. Sha et al. showed that the introduced mutations are generally deleterious for protein and allowed for phenocopy of known mutants across many phenotypes. We described these results in the revised manuscript as a supporting evidence for our assay:

“Finally, we performed CRISPR/Cas9-based genetic screening with gRNA pools targeted at the tenocyte matrix, and found minimal effect on musculoskeletal morphologies (Supplementary Fig. 2b and 2c, deleterious mutations were verified by Sanger sequencing shown in Supplementary Data File).”

In methods:

“Crispant assay in our study is based on the method reported by Shah et al. (<https://www.ncbi.nlm.nih.gov/pmc/articles/PMC4667794/>) as a knockdown/screening tool. This method depends on gRNA-targeted double-strand DNA breaks and subsequent non-homologous end joining, which often introduces insertions or deletions that can disrupt gene functions. We found that all of the gRNAs used for this analysis are highly effective

(Supplementary Data File); all the sequenced embryos were shown to have heavy mutations on the target region, where we could not find a stretch of the wild type sequence.”

Finally, we totally agree with the reviewer in the point that our results do not exclude the role of early tendon cells in the muscle polarization process. We made this point clearer in the revised manuscript, as mentioned in the responses above.

However, it is an agreed point that the timing of expression of the Mature tendon genes (tnmd for example) occurs after the muscle phenotype is observed. Therefore, it seems plausible that mature tendons are not required for the muscle phenotype (as stated in some parts of the section), but that the title of the section should be changed to reflect this more modest interpretation to Mature tendons are not required for myocyte polarization).

We changed the title to: “Mature tendons are not required for myocyte polarization”.

The authors conclude that Wnt is not involved in facial muscle directionality. The authors use gRNAs to disrupt Wnt11, Wnt5a, and Wnt5b expression but do not show any efficacy of knockdown (other than the cartilage defects).

We agree that this is a valid concern and in the revised manuscript we have addressed this in the following ways.

(1) to validate the mutagenesis efficiency of wnt crispants, wnt5a, wnt5b, or wnt11 gRNA-injected embryos were randomly selected and sequenced (Supplementary Data File). We found that all the sequenced embryos were successfully introduced biallelic mutations with some diversity: In case of wnt5b gRNA, while ex1 or ex5 have homogenous frameshift deletion or large insertion, the other embryos showed mosaicism, which will include both frameshift and in-frame mutations.

(2) Regardless of the differences of mutational states, however, wnt5b gRNA-injected embryos consistently showed shorter body length (revised Supplementary Figure 7a), similarly as wnt5b morphant and mutant (Robu et al., (2007) <https://journals.plos.org/plosgenetics/article?id=10.1371/journal.pgen.0030078>). In addition, wnt11 mutant is reported to have shorter eye distance (Heisenberg et al., (2000) <https://pubmed.ncbi.nlm.nih.gov/10811221/>), and this phenotype was well recapitulated in wnt11 gRNA-injected embryo (revised Supplementary Figure 7b). Taken together, we

conclude that our assay is effective for Wnt knockdown.

(3) Although mutated mRNAs are not always undergoing nonsense mediated decay (~20% of mRNA types are expected to undergo this process after introduction of nonsense mutations), after additional experiments we could see downregulation of Wnt5b mutated mRNA after CRISPR editing (Supplementary Data File).

Moreover, it should be noted that Wnt regulation of muscle formation is different between cranial and axial regions (Tzahor et al., 2003). Therefore, later muscle polarization events may also be regulated differently as the Gros et al., 2009 study focused on axial muscle.

We agree with the reviewer and we made this note and referenced the Tzahor 2003 paper: "We cannot fully rule out a direct role of WNT/PCP in controlling facial of myocyte polarization, partly because WNT regulation of muscle formation differs along the body axis (Tzahor et al., 2003), and different WNTs may be involved. However, our results did not yet support such a direct role, and rather support an indirect role via effects on skeletal growth. Future experiments using tissue-specific and inducible transgenes will be needed to test this idea."

Also, it is unclear how Wnt ligands from the expressing cells reach the C2C12 cells (Sup Fig 5d). Since the Wnt producing and receiving cells do not appear to be directly touching each other and Wnt can act as a juxtacrine signal, is Wnt diffusing throughout the culture dish? Is a gradient formed? It seems this experimental set-up is very different from the transplantation of Wnt11 expressing cells done in embryos in Gros et al., 2009.

To address this comment, we performed a new experiment in which Wnt gradients was created in a specialized cell culture chip (80326, ibidi) and the directionality of myotubes was examined. As a result, we still did not find specific pattern of myotubes toward Wnt gradients (Supplementary Fig. 7h). However, we cannot rule out a multitude of potential caveats related to the strength of the Wnt gradient, distance of the Wnt gradient, competence of cells in vitro, and many more. Therefore, after mentioning this experiment in the updated text we write that

"We cannot fully rule out a direct role of WNT/PCP in controlling facial of myocyte polarization, partly because WNT regulation of muscle formation differs along the body axis (Tzahor et al., 2003), and different WNTs may be involved. However, our results did not yet support such a direct role, and rather support an indirect role via effects on skeletal growth. Future

experiments using tissue-specific and inducible transgenes will be needed to test this idea.”

Could the authors comments on why Scx and Acta2a are enriched in the myocyte populations in the single cell RNA-seq experiments?

We agree that despite expressing many markers of terminal skeletal myocyte differentiation, these myocytes are not exactly the same type of cells found in skeletal muscle in vivo and might express some markers found in other cell types (such as Scx found in tenocyte precursor cells, as well as Acta2a found in smooth muscle)

We added a word of caution to the manuscript regarding interpretation of the single cell dataset. “Scx and Acta2 were expressed in C9, hinting the cultured myotubes might be immature compared to those observed in vivo, or may retain some properties of smooth muscle and/or general connective tissue (Supplementary Fig. 12a).”

Sup fig 13 is difficult to read and appears very pixelated.

We have moved Supplementary Figure 13 to a new multi-page Supplementary Data File, because it is probably too big to fit inside a single page. We hope the reviewers agree this is a good solution – we want to show many markers characterizing the various relevant biomechanical processes for this dataset. Plus, not every reader will take the effort to download our dataset and check these markers for themselves, so we believe this Data File will be a handy resource for researchers studying muscle cell differentiation.

Reviewer #3 (Remarks to the Author):

In this study, Sunadome and co-workers used various approaches, including genetics, chemical perturbations, and imaging, to show that cartilage supports directional muscle growth. They provide evidence indicating that this orientated muscle growth is related to the tension and mediated at least from the ECM rather than fusion. They have beautiful live-cell imaging showing the developing myocytes' nuclei migration and division. The stretch experiment is elegant in showing that myoblast alignment and oriented muscle growth independent of tension but also fusion. Though it is still difficult to attribute the disoriented muscle growth to tension or myoblasts number in some assays (fig5ca, fig5c), it generally

provides strong evidence to illustrate the importance of cartilage structure and tension in orientating muscle growth.

Major points:

1. It is great to show multiple approaches to genetics and drug treatment in the paper, and I respect the huge effort that is put in, yet either in situ or staining is necessary to show that the targeted genes/proteins are perturbed.

We agree with the reviewer that we should be very careful with all gene perturbation studies as the amount of chimerism and residual gene expression could affect the interpretation.

As the reviewer suggested, it would be perfect if we could directly confirm the knock down efficiency with antibody staining. Unfortunately, during this revision this strategy didn't work. This happened because we targeted so many genes that we didn't have enough money and time to optimize and select antibodies that would unambiguously work from the multitude of available antibodies on the market. The other option would be in situ hybridization. However this approach is plagued by another problem – only 20% of the types of mRNA undergo nonsense mediated decay, whereas the other mRNAs containing edited nonsense mutations would be present at the same level of control even if no functional protein was generated. Still, we took this approach according to the reviewers advice as we hoped to see the downregulation of some edited mRNAs. According to our expectations we observed the downregulation of some mRNA's with nonsense mutations (Supplementary Data File). Because these first two approaches were not systematic enough to allow us to analyze all crispant embryos, we opted for a universal method which is the Sanger sequencing of CRISPR edited embryos where we could observe the presence of mutated versus non-mutated alleles.

Crispant assay in our study is based on the method reported by Shah et al. (<https://www.ncbi.nlm.nih.gov/pmc/articles/PMC4667794/>) as a knockdown/screening tool. This method depends on gRNA-targeted double-strand DNA breaks and subsequent non-homologous end joining, which often introduces insertions or deletions that can disrupt gene functions. To examine the mutagenesis efficiency of our method, we injected gRNAs used in our study, randomly selected the embryos, and sequenced them. During this revision, we sequenced all types of crispant embryos and found that all of the gRNAs used in our study are highly effective (Fig. rev3-1); all the sequenced embryos were shown to have heavy mutations on the target region, where we could not find a stretch of the wild type sequence.

There is a diversity of mutational states: In case of wnt5b (Fig. rev3-1b) and lamc1 (Fig. rev3-1f), while wnt5b-ex1, wnt5b-ex5, lamc1-ex4, and lacmc1-ex5 have homogenous frameshift deletion or large insertion, the other embryos showed mosaicism, which will include both frameshift and in-frame mutations.

For instance, regardless of the differences of mutational states, wnt5b gRNA- and lamc1 gRNA-injected embryos consistently showed shorter body length (Fig. rev3-2), as expected from the analyses of morphants or mutants (Robu et al., (2007) <https://journals.plos.org/plosgenetics/article?id=10.1371/journal.pgen.0030078>, Goody et al., (2012) <https://pubmed.ncbi.nlm.nih.gov/23109907/>). These results show our method is effective for gene knock down assay.

Finally, most of our conclusions are based on strong phenotypes caused by the CRISPR editing, even if we admit in principle that the embryos might be chimeric to some extent. For example, with respect to Sox9, we are using the CRISPR system as a molecular tool to impair cartilage development, in order to study the role of cartilage growth in muscle polarization (as we are not studying the role of Sox9 in cartilage induction). As we successfully achieved the impairment of cartilage development, the theoretically possible residual chimerism is not a point of concern. This logic applies to other results with strong positive phenotypes. However, the absence of phenotypes in some types of crisprant embryos becomes another case which interpretation is truly sensitive to residual chimerism. As we performed additional control experiments (including HCR, sequencing, phenotype matching), and also new functional tests in vitro for Wnt genes), we increased our level of confidence in the knockout efficiency. Additionally, we do not draw strong crucial conclusions from the negative experiments, as we admit those can be confounded by residual chimerism and other possible technical issues. Instead, in the revised manuscript, we conclude on such experiments in a suggestive way, rather than an authoritative way.

2. Please address why sox9 and runx2 mutant have different severity.

The comment addresses the experiments performed using Sox9a/b gRNAs compared to morpholino-based knockdown of Runx2.

We explain this in the revised manuscript as such:

“This can be explained by differential impacts on cartilage development, which in turn might be explained by previous reports that morpholino knockdowns are generally more potent than CRISPR editing.”

We were pleased when we found that perturbation of Sox9 and Runx2 gave us an instrumental way to impair cartilage development to different extents. This difference in the strength of the phenotype, whether it is caused by biological or technical factors, was helpful and enabled us to observe the behavior of myocytes as a function of the extent of cartilage induction.

3. Where is fig4b 39h image? Why is the time point of the control different from mutants? Why are the images' time points (left) and quantifications (right) mis-matched?

Crispant and morphant zebrafish embryos displayed variation in temporal landmarks of muscle development when it comes to the live imaging, and so even if the exact hpf was different we were trying match the morphological structures (cartilage, eyes, muscle shape). In the revised manuscript, we have obtained the better matching time points for a strict comparison across the conditions. These are now matched with quantifications in revised Fig 4c and 4d.

4. In Fig 4e, very interesting results and a beautiful assay. How do myocytes sense the new cartilage attachment point?

We thank the reviewer for the kind words.

This topic is complicated as there are many matrix molecules that might affect the adhesion of myocytes to cartilage, and the sensation mechanism might be altogether independent from adhesion and include some form of signalling gradient. Therefore, we have tried to address this question by knocking out the integrin and laminin family genes, which products are covering the forming cartilage.

From the revised text:

In order to determine the composition of the laminin matrix, we examined the gene expression pattern of Laminin α 2 (Lama2), Laminin α 1a (Lamb1a), Laminin α 1b (Lamb1b), and Laminin γ 1 (Lamc1), which are reported to be required for the attachment and integrity of muscle in the somite of zebrafish (Hall et al., 2007; Snow et al., 2008). At 56 hpf, we found specific and strong expression of Lamc1 in AM muscle, Lamb1a in Meckel's and palatoquadrate cartilage, and Lamb1b both in the muscle and cartilage (Supplementary Fig.

10c). We inhibited Laminin function using CRISPR/Cas9-mediated mutagenesis and verified editing by Sanger sequencing (Supplementary Data File). *Lamc1*-crispant embryos had short body length, as expected in cases of *Lamc1* loss of function (Supplementary Fig. 10d). Myofibrils of *Lam* crispant embryos appeared misaligned (although generally oriented towards incorrectly positioned cartilage) (Supplementary Fig. 10e). Live-imaging analysis shows individual myoblasts of the *Lamc1*-mutant embryo were able to contact the cartilage but subsequently failed to elongate (Supplementary Video 16 and Supplementary Fig. 10f). Additional loss of function assays on extracellular matrices highly expressed in cartilage or tendon either affected the organization and orientation of facial muscles (such as type II collagen, aggrecan, and comp, primary structural proteins of cartilage, see ECM in Supplementary Fig. 10e), or did not cause any obvious abnormality in the pattern of muscle structure (such as *Tsp4b*, a gene with functions at the muscle-tendon junction (Subramanian and Schilling, 2014), Supplementary Fig. 2b). Together, these data suggest that many different matrix molecules are involved in establishing and maintaining the physical contact between muscle and cartilage.

Also, from the revised manuscript:

“We performed CRISPR/Cas9 editing of *Itgb1a* and *Itgb1b*, two zebrafish paralogues of the *Itgb1* gene, and verified editing by Sanger sequencing (Supplementary Data File). The crispant embryos showed profound abnormalities in muscle, affecting polarization (Supplementary Fig. 9b), nuclear positioning (Supplementary Fig. 9c), myofiber length, (Supplementary Fig. 9d) and fusion (Supplementary Fig. 9e). The extent of abnormalities of muscle and cartilage in *Itgb1* crispant embryos appeared similarly severe to that of *Runx2b*-deficient embryos, in which cartilage development is severely impaired. Although we cannot rule out whether inactivation of *Itgb1* might cause muscle-autonomous effects, these findings are consistent with a model in which mechanical sensing via integrins guides oriented muscle development *in vivo*.”

Here we would like to speculate that how myocyte sense the new attachment point is likely to depend on ECM adhesion via integrins or a similar cell-adhesion protein or glycoprotein (as integrin adhesion does seem to play a role in attachment if you take a look at Supp Fig 9), and we also know from our phalloidin staining of C2C12 myocytes seeded on hard/soft substrates that myocytes on ECM-coated material can grow filamentous actin, which helps polarization. In conjunction with this, the increased stiffness of the nascent cartilage compared to muscle (judging from our AFM data in supplementary figure 15) might be one

part of the sensing mechanism.

Is the ECM affected after laser ablation? Any cytokines changes?

Aside from ECM being demolished by the laser locally, we were not able to assess how far the ECM damage spreads from the laser ablation, however it must not have been too sprawling of a lesion because of how the muscle was able to rapidly re-orient and attach to a different surface. We hope to continue our investigation along the lines of muscle regeneration and reattachment in a follow up project, and this will take a long time and require a variety of new technical approaches/zebrafish strains, and therefore we believe this is beyond the scope of the current specific manuscript. We included the caveat that these parameters were not checked in the current study.

“We did not assess downstream alterations in the nearby ECM or signaling environment following laser ablation.”

5. In Fig 5a, there is a need to show that it is not due to the change in myocyte number. Also, it would be good to use other myosin inhibitors to show drug specificity.

We agree with the reviewer and we performed several new experiments to address these comments (major results can be found in the revised Supplementary Figure 6).

Since it is technically difficult to segment all of myocytes and count them (see our stochastic labelling using CreERT2), we counted myofiber in the cross-section, which is indicative of myocyte number, in embryos treated with drugs as in previous Fig. 5a. In this assay, we incorporated (+)-Blebbistatin, which is the inactive enantiomer of (-)-Blebbistatin, for the confirmation of the drug specificity. As expected, in contrast to (-)-Blebbistatin, (+)-Blebbistatin does not affect the myofiber alignment (revised Supplementary Fig. 6f).

Myofiber number in cross-section (revised Supplementary Fig. 6i-j dotted line in left panels show the position where the optical slices were produced) was slightly increased in (-)-

Blebbistatin-treated embryo compared to two control samples (revised Supplementary Fig. 6l), which may be due to the delay of fusion. We also counted the number of nuclei in region segmented by MyHC staining, and we did not find difference between samples (revised Supplementary Fig 6h). Overall, mis-polarization caused by (-)-Blebbistatin is not likely due to the reduction of myocyte number.

We also took the reviewers advice and used other myosin inhibitors to demonstrate drug specificity. We used two alternative myosin inhibitors to check if a similar effect on the zebrafish craniofacial muscles would occur with a similar treatment regime (55-74hpf). First we tried a low-affinity, non-competitive inhibitor of skeletal muscle myosin II called 2,3-butanedione monoxime (BDM, revised Supplementary Figure 6c-d), which is well characterized to inhibit the contraction of vertebrate muscle. We also used a highly specific myosin II inhibitor called N-Benzyl-p-toluenesulphonamide (BTS, revised Supplementary Figure 6a-b), which blocks actin-stimulated ATPase activity in skeletal muscle myosin II but has no effect on platelet myosin II. In both cases we found a pattern of fibril disorganization of the craniofacial musculature compared to controls, similar to the blebbistatin treatment.

6. Fig 5c, are myocytes number reduced in drug-treated fish and crispant? Without proper control, it is hard to conclude that the myofiber elongation depends on cytoskeletal organization

Firstly, in the revised manuscript we counted myofibers in cross-sections of muscle of previous Fig 5c. The result shows no reduction in the number of multinucleated myofibers in any of the conditions. On the opposite, (-)-Blebbistatin-treated embryos showed significantly increased number of shorter myofibers (revised Supplementary Fig. 6t), suggesting a possible fusion defect. In ttn-gRNA injected embryos, a difference was not observed with statistical significance (Supplementary Fig. 6t).

Since myofibrils in ttn gRNA-injected and (-)-Blebbistatin-treated embryos are shorter than control and do not elongate from end to end of the muscle (see revised Fig. 5d and 5f), the actual myocyte number would be underestimated by simply counting myofibers in cross-sections.

To cope with this challenge, in the revised manuscript we counted the number of nuclei in 3D regions segmented by MyHC staining (revised Supplementary Fig. 6m-o and quantified in Supplementary Fig. 6p), and found that there was a slight decrease in nuclei count, with

statistical significance in case of ttn-gRNA injected embryos ($p = 0.049$, unpaired T-test), but not blebbistatin treated embryos. Having less nuclei per muscle as well as per myofiber, this could indicate a cell proliferation defect in titin-perturbed embryos.

Despite these effects on nuclei count or myofibril count in blebbistatin/titin conditions, the magnitude doesn't look sufficient to account for the severe shortage of myofiber length which suggests that the elongation defect were not secondary only to proliferation/fusion defect. This results actually suggests that cytoskeleton in myocytes is important for myofiber elongation.

Minor points:

1. The development time point is missing in Fig1c, 1d, and others.

We have added missing time points to either the figures or corresponding figure legend.

2. Fig 3d middle graph _ where is the t-test result for the volumes?

The cartilage volumes are significantly different between control and DTA mice (Student's t-test, $p < 0.05$), whereas the muscle volumes are not significantly different between control and DTA mice (Student's t-test, $p > 0.05$). The cartilage is significantly more voluminous than the muscle (Student's t-test, $p < 0.05$). We added it to the figure legend (revised Figure 3e and f).

Reviewer #4 (Remarks to the Author):

This is a revised version of a manuscript that investigate the role of cartilage in regulating the development of striated muscles in zebrafish. The authors answered all my questions in great details and really improved their manuscript. The responses are thoughtful and thorough. I have no concerns.

Thank you!

Reviewers' Comments:

Reviewer #1:

Remarks to the Author:

The authors have addressed all our residual concerns and I am happy to recommend publication. I look forward to seeing it in print.

Reviewer #2:

Remarks to the Author:

The effort to address all of my previous questions is greatly appreciated and I no longer have any concerns about their manuscript. Their work nicely describes the regulation of myocyte orientation using fish and mouse systems.

Reviewer #3:

Remarks to the Author:

In this study, Sunadome and co-workers used various approaches, including genetics, chemical perturbations, and imaging, to show that cartilage supports directional muscle growth. They provided evidence indicating that this orientated muscle growth is related to the tension and mediated at least from the ECM rather than fusion. They had beautiful live-cell imaging showing the developing myocytes' nuclei migration and division. The stretch experiment is elegant in showing that myoblast alignment and oriented muscle growth independent of tension but also fusion. All questions are addressed in the revised manuscript. The revised manuscript provides strong evidence to illustrate the importance of cartilage structure and tension in orientating muscle growth. I will recommend acceptance.